

**OLYMPUS v1.0: Development of an integrated air pollutant and GHG urban emissions model - Methodology and calibration over the greater Paris**

Arthur Elessa Etuman[1], Isabelle Coll[1]

[1]Laboratoire Interuniversitaire des Systèmes Atmosphériques (LISA), UMR CNRS 7583, Université Paris Est Créteil et Université Paris Diderot, Institut Pierre Simon Laplace (IPSL), Créteil, France

*Correspondence to*: Arthur Elessa Etuman (Arthur.elessa-etuman@lisa.u-pec.fr)

**Abstract.** Air pollutants and greenhouse gases have many effects on health, the economy, urban climate and atmospheric environment. At the city level, the transport and heating sectors contribute significantly to air
pollution. In order to quantify the impact of urban policies on anthropogenic air pollutants, the main processes leading to emissions need to be understood: they principally include mobility for work and leisure as well as household behavior, themselves impacted by a variety of social parameters.

In this context, the Olympus modeling platform has been designed for environmental decision support. It generates a synthetic population of individuals and defines the mobility of each individual in the city through an activity-
based approach of travel demand. The model then spatializes road traffic by taking into account congestion on the road network. It also includes a module that estimates the energy demand of the territory by calculating the unit energy consumption of households and the tertiary sector. Finally, the emissions associated with all the modeled activities are calculated using the COPERT emission factors for the traffic, and the European Environmental Agency (EEA) methodology for heating-related combustions. The comparison of emissions with AIRPARIF's
regional inventory shows discrepancies that are consistent with differences in assumptions and input data, mainly in the sense of underestimation. The methodological choices, as well as the potential ways of improvement, including the refinement of traffic congestion modeling and of the transport of goods, are discussed.

## 1 Introduction

As the world population is rising, the share of the population living in urban areas is increasing (United Nations,
2014). These areas can be described as activity clusters supporting a substantial density of individuals, buildings, transport networks and job centers. All of the human activities inherent in these metropolises induce a major local consumption of fossil energy and natural resources, favoring the concentration of a great variety of nuisances (noise, stress, pollution). Among the most emitting activities induced by the city, one can find - according to the nomenclature of the (IPCC, 1996) - energy consumption, industrial processes, solvent use and agriculture. But at
the city scale, anthropogenic emissions result mainly from fuel combustion related to road transportation, as well as residential, commercial and institutional heating and boiling, that represent more than half of the total urban emissions (International Energy Agency, 2016). In Europe in particular, the fact that many cities are heavily dependent on cars (and sometimes on diesel fuel) adds to the air pollutant emissions balance. Thus, road transportation, production of electricity and heat represent more than 60% of the anthropogenic emitted nitrogen
oxides (NOx), fine particles under $2.5\mu$m (PM$_{2.5}$) and non-methane volatile organic compounds (NMVOC) (International Energy Agency, 2016). Quantitatively, although sulfur oxide (SOx) emissions have been decreasing since the 1990s, NOx and particulate matter (PM) emissions are still increasing in Asia and do not show a clear downward trend in Europe (Amann et al., 2013; Klimont, 2017; Miyazaki et al., 2016). As a result, even if the exposure to short-time duration peak decreases, the population's exposure to chronic pollution is still high in
European urban areas (EEA, 2015), and 94% of exceedances of the short-term limit value for PM$_{10}$ have been observed in urban or suburban areas (EEA, 2016). The health consequences are major. Recent estimates confirm the considerable burden of disease associated with urban air pollution, which is expressed through the occurrence of pulmonary and cardiovascular diseases, cancer, but also certain types of diabetes in adults, or through an attack on the neuronal development of very young populations. From an economic point of view, this represents a high
cost of health care and a significant drop in productivity for businesses. In parallel, the societal issue related to the degradation of air quality is growing. According to a survey carried out between 2007 and 2015 on behalf of the European Commission (European Commission, 2010), there are 9 European Union capitals among the 20 cities with the lowest rate of people satisfied with the quality of urban air, with the biggest decrease of the satisfaction index being observed in Greater Paris. To deal with these issues, it becomes necessary to characterize the link
between city, individuals, energy consumption and pollutant emissions, in order to understand how urban



5 structures guide residents' behavior. This turns out to be a necessary step in understanding the levers for improving air quality in large cities.

The literature now recognizes the determining role of city configuration and building structure (Borrego et al., 2006), as well as that of individual mobility in the exposure of individuals to pollution of the agglomeration. In

10 particular, the relationship between the individual and urban space is known to be at the origin of a highly differentiated exposure, discriminating places of residence, lifestyles and social categories to be modified (O'Neill et al., 2003).

Emission inventories for a current situation can be obtained through a top-down approach (using national

15 aggregated information and spatial keys to distribute emissions on the territory), or through a bottom-up approach (collecting local information from specific sectors - e.g. road traffic count data - to generate a high-resolution inventory). Conventionally, regulatory coefficients are applied to current emissions to produce prospective emission inventories in order to stand for technological developments and the effects of a constant re-evaluation of emission standards. However, in view of the elements mentioned above, emission models used for prospective

20 scenarios must be able to take into account the effects of urban planning in reducing the use of personal vehicles, as well as the role of individual practices in energy consumption. Thus, prospective emission calculations need to be rethought on the basis of the relationship between urban organization, individuals and activity. Only in that way can we help to identify the levers of urban air quality and urban sustainability.

There are still a few research projects in literature that have incorporated a large number of urban components into

25 the forcing of air quality models (Manins, 1995; Marquez and Smith, 1999; Martins, 2012; De Ridder et al., 2008). However, recent model developments have allowed the articulation between city and environmental mobility. Thus, over the last decade, social components have progressively been integrated in urban emissions models, such as TASHA-MATSIM-MOBILE6.2C (Hao et al., 2010) or TRANUS-TREM (Bandeira, Coelho, S, Tavares, & Borrego, 2011), which are now able to quantify the impact of urban policies on road traffic emissions by taking

30 into account a variety of parameters such as car-pooling, transportation fleet technology and modal choice for individuals. The strength of these models lies in the choice of a microsimulation approach based on population choice, which depend on economics parameters. However, as seen in the works of (Hatzopoulou et al., 2008; Hülsmann et al., 2014), most of the resulting studies focused on road traffic emissions only. Yet, in the current environmental context which places great emphasis on the emerging concept of sustainable cities, it is necessary

35 to consider all the emissions of air pollutants that are related to energy consumption, as they interact with climate change. In particular, there is a need to also take into account small combustion emissions (both residential and commercial) and their related policies to go further in the realism of the urban scenarios, and to treat the levers of future air quality more holistically.

40 The OLYMPUS modeling platform was developed with the aim of taking into account the connections between the types of urban organizations, regulatory constraints, energy consumption behaviors and pollutant emissions. The objective is to build present, progressive or hypothetical urban scenarios in which we heighten the role of the built environment, and that of political and economic forcing, in the exposure of people to atmospheric pollutants. OLYMPUS is an emission modelling platform based on the simulation of the behavior (mobility, energy

45 consumption) of individuals within an urban space. That is, it considers every single individual and the impact of their daily choices for activities, mobility and energy consumption practices on combustion emissions. In this paper, an overview is made of the characteristic features of the model. The main modules composing OLYMPUS will be presented individually. An application on the Greater Paris will be shown in the last section. The results and uncertainties of the model are discussed.

50 **2 OLYMPUS model overview**

The objective of this model is to estimate the pollutant emissions linked with energy-consuming urban activities. In this frame a first necessary step is the simulation of a synthetic population and its distribution in the considered urban structure, in order to quantify and to spatialize the targeted activities (road transport and building/domestic heating). Then, the model uses emission coefficients based on activity to produce a spatialized inventory of volatile

55 organic compounds, nitrogen and carbon oxides and primary particulate matter. The OLYMPUS platform was designed to grasp information on city-specific parameters such as morphology, distribution of populations and employment centers, road transport networks and public transport, energy consumption units and climatic variables that influence the emissions.



### 2.1 Main characteristics

In its current version, OLYMPUS models the pollutant emissions due to road transport and combustion processes from the residential and tertiary sector. It is composed of 6 calculation modules, supporting 4 main tasks (see Figure 1).

10 One first specificity of OLYMPUS is that it relies on a **synthetic population** for the calculation of emissions and constitutes the first task.
(a) This synthetic population is designed to be representative of the population living on the territory considered, and is characterized by the agents' age, gender, principal activity and situation in a household. The creation of such a synthetic population is based on the reconstitution of surveys in the GAIA module.

The second task of OLYMPUS is the provision of a road transportation database, calculated by considering individuals' lifestyle. This database is obtained from successive diagnoses about individual trip generation - based on zonal attractions, travel distribution, modal split and route choice. In the OLYMPUS modelling process, this task relies on 3 modules.
20 (b) A first module (THEMIS) defines the **accessibility** and **attractiveness** of the different administrative units of the city, as well as the average time travels between them.
(c) The **Activity-Based Travel Demand** (ABTD) module called MOIRAI computes the daily activity patterns of all agents. It also describes their temporal and spatial daily mobility.
(d) The **assignment** module HERMES provides spatialized daily trips by computing the shortest path between 25 the origin and the destination of a trip (OD matrices).

In parallel, OLYMPUS tackles the third task by computing the **building energy demand.**
(e) For this purpose, the HESTIA module computes the average household energy consumption per square meter, the dwelling size of the household and the energy mix of the city in order to produce a spatialized building 30 energy demand, including a climate correction specific to the simulated period.

In the fourth task, OLYMPUS generates **emissions** from both road transport and small combustion heating systems.
(f) Those emissions are calculated using reference methodologies. The computation of pollutant emissions is 35 carried out by the VULCAN module.

All the OLYMPUS running scripts are based on a shell interface, programmed in Python 2.7 and compiled in C for faster execution speed. For graph networks and spatialized data analysis, NetworkX (Hagberg et al., 2008), GeoPandas and NetCDF's libraries have been included to the Python interpreter. Because of the significant number 40 of calculation loops, the model applies a data parallelism which consists in a partitioning of data with a Multithreading approach.

### 3 The synthetic population generator (GAIA)

The synthetic population generator GAIA is the first OLYMPUS module to be run. It allows the generation of a synthetic population that is representative of a given urban area. The synthetic population generator mainly uses 45 urban-level census data to attribute age, gender and main activity – as well as socio-economics parameters such as the possession of a driver's license - to each agent in this population. The module spatializes this synthetic population through an urban zoning based on household densities in the urban area, an exogenous variable provided to the model. In the end, a synthetic population based on data census or demographic scenarios is obtained, with an individual description of its agents.

50

There are several techniques to estimate a population in a small area based on statistical approaches, as listed in (Rahman, 2017). The most common method is the Iterative Proportional Fitting (IFP) procedure (Deming & Stephan, 1940, Baggerly, & McKay, 1996; Müller K., & Axhausen, K.W. 2010), which generates an adjusted matrix of the survey data used to constrain the global synthetic population patterns, based on the minimization of 55 chi2, a method for estimating unobserved quantities from marginal numbers. The algorithm must be fed with the total population data and subtotals by property type using both aggregated and disaggregated data. Conditional probabilities are also one of the methodologies for creating a synthetic population. This approach is based on Bayesian statistics and it relies on a representative sample of population, in which the discrete conditional probabilities governing every characteristic (age for example) are identified. Then, a unique value of this



characteristic is assigned to each agent of the population using a random distribution that follows the identified probability law. The approach makes it possible to create (on the basis of a representative sample) a database that distinguishes each individual (disaggregated data) as well as each household and dwelling, by attributing them to their own characteristics. These two methods differ in how to generate the dummy population but the results are admittedly identical. In terms of outputs, one of the major interests of the IPF procedure is its ability to generate

greater variability in populations than conditional probabilities. However, the use of this approach in a region such as Ile-de-France requires a significant amount of work to structure the input data. For this reason, we decided to use the conditional probability approach, which has been widely used in the fields of populations' transport demand modelling (Antoni et al., 2010; Banos et al., 2010; Mathis et al., 2008). In order to mitigate a possible lack of variability, we proceeded to the implementation of a spatial component in the distribution of the socio-economic

characteristics of the agents.

The implementation of GAIA takes place in 2 main stages and is referenced in Figure 2.
- The determination of the urban structure, based on an urban density index (**UDI**) and divided into 3 classes: the urban pole (CENTER), urban areas (URBAN) and suburbs (SUBURBAN). (Figure 2.b)

- For each household in the urban area, the module generates the synthetic population by defining the household size and the agent properties using conditional probabilities. (Figure 2)

### 3.1 Urban structure

The prerequisites of the population generation are the domain definition and the classification of urban areas in

based on an urban density index (**UDI**). GAIA discretizes the type of urban zone on a scale from 1 to 3 (SUBURBAN – URBAN – CENTER) depending on population density. Figure 2.b represents a schematic representation of UDI with population-specific attributes in each urban area based on the UDI. Such a discernment is needed for realistic purposes because population density affects the urban landscape (buildings and houses), the localization of activities and the structure of households. Here, the assumption made is that household family type

distribution is different between urban pole, urban areas and suburb areas, and that building and house distribution will vary between the different zones. In parallel, the structure of households varies according to the distance from the urban center, following (Hulchanski, 2010). These hypotheses provide a larger spatial distribution variability of agents than a simple conditional probability distribution would.

The UDI is defined in Eq. (1) This is the result of the classification of the dataset following 3 large sets by applying

a linear cut from sparse to very dense areas. It is based on population density and the population is digitized as a function of the population density logarithm.

$$
UDI(z) = \begin{cases}
\textit{CENTER}, \; if \; \log\left(1 + \frac{n_{hh}}{A}\right) > \propto_2 \\
\textit{URBAN}, \; if \; \propto_1 < \log\left(1 + \frac{n_{hh}}{A}\right) < \propto_2, \\
\textit{SUBURB}, \; if \; \log\left(1 + \frac{n_{hh}}{A}\right) < \propto_1
\end{cases} \quad (1)
$$

Where $\propto_1$ and $\propto_2$ are key classification values depending on the logarithm of households density, $n_{hh}$ the number of households, z a specific area of the domain and A is the surface area.

### 3.2 Generation of synthetic population

The generation of the population depends on Probability Mass Functions (PMF) that rely on census data as referred to in figure 2.d which represent the PMF of the age of an agent living alone. In each zone and for each household,

GAIA uses a discrete probability distribution to:

(a) Define the number of agents in the household
(b) Characterize the type of family
(c) Define agents gender, age, principal activity


Eq. (2) predicts the number of agents in the household, depending on the type of zone (CENTER, URBAN, SUBURBAN) that differ in terms of urban structuration. The probability to have *n* agents in the household is based on conditional probabilities, and defined by a truncated Poisson distribution:





$$P_{hhs}(n|\lambda, UDI) = e^{-\lambda}\frac{\lambda^n}{n!} \qquad (2)$$

where $\lambda$ is the average household size, $n$ is the number of agent in household $\in A$, (with A = [1,7] and A $\in \mathbb{N}$). Figure 2.c is an example of the household size probability distribution based on a truncated Poisson's law.

Eq. (3) is used to define the type of household among the 4 family classes which are Alone (male/female); Couple - No children; Couple with children; Single parent family (male/female). The selection of the family type is also based on conditional probabilities ($P_{FAM}$) and follows

$$P_{FAM}(n) = \begin{cases} \text{"Alone"}, & if\ n = 1 \\ P_{1F}(n), & if\ n = 2\ and\ \in R_{1F} , \\ P_{2F}(n), & if\ n \geq 3\ and\ \in R_{2F} \end{cases} \qquad (3)$$

where $R_{at}$ is family type set for n individuals in the household

$R_{1F} = \{\text{"Couple"}, "single\ parent\ family"\}$

$R_{2F} = \{\text{"Family"}, "single\ parent\ family"\}$
and $P_{1F}(n), P_{2F}(n)$ correspond to weighted functions based on survey data.

Eq. (4) allows estimating the agents' attributes (age, gender, principal activity). The gender of every agent is defined by a conditional probability (while the gender of its partner is opposite) such as:

$$P_{gdr}(\eta) = \begin{cases} P_{sex}(), & if\ householder\ and\ \in R_{gdr} \\ P_{sex}(), & if\ Child\ and\ \in R_{gdr} , \\ P_{sex2}(), & if\ Partner\ and\ \in R_{gdr} \end{cases} \qquad (4)$$

$\eta$ represent the agent situation in the household, $R_{am}$ is the sample space and consist of 2 elements {"male", "female"}, $P_{sex1}()$ correspond to a weighted function based on census data and $P_{sex2}()$ is conditioned by the householder sex.

The age of an agent depends on the type of household – still based on conditional probabilities – which is linked to specific sample spaces: for householders, for couples (age difference less than 20 years), and for children. There are 20 age classes with a 5-year division.

$$P_{AGE}(\eta) = \begin{cases} P_{A\_1}(n), & if\ householder\ and\ \in R_{householder} \\ P_{A\_2}(n), & if\ Child\ \quad and\ \in R_{child} , \\ P_{A\_3}(n), & if\ Partner\ and\ \in R_{householder} \end{cases} \qquad (5)$$

Where $P_{A\_1}(n), P_{A\_2}(n)$ and $P_{A\_3}(n)$ are probability mass functions based on census data. $R_{householder}$ , $R_{child}$ and $R_{partner}$ are age type set for n individuals in the household with
$R_{head} \quad = [20;70]$ and $R_{head} \in \mathbb{N}$
$R_{child} \quad = [0;20]$ and $R_{child} \in \mathbb{N}$
$R_{partner} = [20;70]$ and $R_{partner} \in \mathbb{N}$

The principal activity of the agent depends on its age and on the unemployment rate. Agents under 18 are educated and agents over 65 are retired. The other agents may be either employed, unemployed or follow studies.

$$P_{ACT}(age) = \begin{cases} P_{ACT1}(age), & if\ 20 < age < 30\ and\ \in R_{act1} \\ P_{ACT2}(age), & if\ 30 < age < 65\ and\ \in R_{act2} , \\ \text{"School"}, & if\ f\ age < 18 \\ \text{"Retired"}, & if\ age > 65 \end{cases} \qquad (6)$$

Where $P_{ACT1}(age)$ and $P_{ACT2}(age)$ represent the probability mass functions to have as principal activity one of the activities set $R_{act1}$ or $R_{act2}$
$R_{act1} = \{\text{"Active"}, "School"\}$





$R_{act2} = \{"Active", "Unemployed"\}$

**4 Road transportation generation**

To simulate the transport demand of the population based on activities, OLYMPUS needs to be fed with numerous external data. The spatial distribution of employment centers is one first key parameter: OLYMPUS is filled in with a spatialized file containing information on the number of employment per zone. The data are usually
formatted under a Geographic Information System (GIS) due to their spatial pattern. To simulate the ABTD, the model requires socio-economic data at a disaggregate level for each zone, which are provided by the synthetic population generator GAIA. Then, transport networks are required as another external parameter. The road network includes the city highways and main traffic lanes with information on free-flow speeds. The transit network includes all the transit stations and the format of the data is also GIS. All these data will be analyzed at a
local level called travel analysis zones (**TAZ**) which may be districts, sub-districts, municipalities or any other city division. The degree of precision of the zones imposes the refinement of population mobility.

Urban road transport modelling is organized in 3 steps:

(a) Determination of the attractiveness and accessibility of the different zones that constitute the domain.
(b) Restitution of the agent movements based on the realization of the programmed activities.
(c) Assignment of motorized displacements on the road network.

**4.1 TAZ accessibility and attractiveness (THEMIS)**

The operating diagram of this module is presented in figure 3.a.
The main steps are:
-    Definition of accessibility
-    Computation of attractiveness

One of the main data driving the ABTD model (ABTDM) is the accessibility of the TAZ, which helps to estimate the mobility choices. Accessibility refers to the pool of activities considered as useful within a given radius. Its value accounts for the potential of access to this zone by public transportation, which makes it a crucial parameter for the choice of the agent transportation mode. For this purpose, THEMIS analyses the TAZ population density,
road network and public transportation network. This results in a 5-level index of public and private transport accessibility to the area which is called the Urban Transport Accessibility Index (**UTAI**). The hypothesis we make is that density of household and public transportation station both influence the transportation mode. As shown in Figure 3.b, a zone with $UTAI_{MIN}$ will only be served by WALK and a zone with $UTAI_{MAX}$ will be well connected with a large choice of transport infrastructures. The definition of the 5 UTAI classes depends on the value of $\mu$,
defined by the following equation:

$$UTAI(zone) = \begin{cases} 1, if\ \mu < \propto_1 \\ 2, if\ \propto_1 < \mu < \propto_2 \\ 3, if\ \propto_1 < \mu < \propto_3, \\ 4, if\ \propto_1 < \mu < \propto_4 \\ 5, if\ \mu > \propto_4 \end{cases} \quad (7)$$

with

$$\mu = \log\left(1 + \frac{n_{hh} \times n_{st}}{A}\right),$$

Where $\propto_1$, $\propto_2$, $\propto_3$ and $\propto_4$ are key classification values depending on the logarithm of household density and public transport density. $n_{hh}$ is the number of households per TAZ, $n_{st}$ is the number of public transportation stations in the TAZ and A is the area of the TAZ.
The UTAI index (see the correspondence in Figure 3.b) thus helps to design the city use from its transport
infrastructure and to define realistic public transport travel time such as Paris public transport isochronous curve (Figure 3.c).



The attractiveness of activities is an important parameter that shapes the agenda of population agents. It can be defined as its ability to attract agents in order to perform a given activity and based on the average distance of a trip as shown in Figure 3.c. The model assumes that there are 2 types of activities: WORK and OTHER. The complete list of the OTHER activities taken into account in the ABTD model are:

- HOME
- SCHOOL
- SHOPPING
- SECONDARY
- ACCOMPAGNYING
- VISIT
- LEISURE

The major parameter that differentiates between our 2 types of activities is the average trip length that varies widely, the average work-based trip distance being longer than the other activities-based trips. The distance to the TAZ is a substantial variable in estimating its potential attractiveness. The attraction potential of WORK depends on the number of jobs in the TAZ. Concerning OTHER activities, their attraction is a function of the population density per TAZ. However, some activities like visiting a friend or going on vacation remain underestimated by the ABTD model.

The activity attractiveness calculation is based on the gravity theory model from (Huff, 1964). It relies on the definition of an activity weight, and it works by analogy with Newton's law of gravity. The probability of conducting any activity at a specific location is thus defined as follows:

$$P_{\Omega}(i,j) = \frac{\Omega(i,j,d)}{\sum_j \Omega(i,j,d)} \qquad (8)$$

where the attractiveness $\Omega$ is defined by:

$$\Omega(i,j,d) = \sum_i Act \times \sum_j Act \times \left( \frac{1}{\sigma\sqrt{2\pi}} e^{-\frac{1}{2}\left(\frac{|x-\overline{d}|}{\sigma}\right)} \right), \quad (9)$$

and $\sigma = \frac{\overline{d}}{2}$,

In this equation, $d_{mean}$ represents the mean distance to reach the *Act* activity, while *i* and *j* respectively are the indexes of the origin and destination zones.

In the end, the attractiveness parameter is highly dependent on the city structure and inhabitant travel uses. (Kwan, 2003) found that few peoples are acting to minimize their journey to work by relocating either their home or workplace. Considering this, mobility surveys provided by some countries may be used to match realistic mean travel distances.

## 4.2 Activity-based travel demand (MOIRAI)

The MOIRAI module simulates the mobility choices for each agent of the synthetic population during the day. One essential challenge of the module is to represent the mobility in the most realistic way possible, by taking into account the social constraints of each agent in space and time. Several ABTDM exist in literature. (Malayath and Verma, 2013) propose a review of existing models and their uses. Based on this review, we have decided to use the theory of random utility to simulate the choice of individuals in MOIRAI. In this theory, a stochastic approach allows to take into account rationality in the agent decisions. That is to say, the decision is described as the choice to do what is most useful, depending on possibilities. In this process, utility is generally expressed according to 2 components, one describing the observed practices and another one describing the random component.

In the theory of random utility, the main hypothesis is the maximization of utility, influencing the decisions of the agent. MOIRAI is based on the use of the MultiNomial Logit (MNL) model (McFadden, 1973), which considers that the random components of the utility are Independent and Identically Distributed (IID) and that the distribution is Gumbel type:

$$P_{mode,i} = \frac{e^{-Utility_{mode,i}}}{\sum_{i=0}^{n} e^{-Utility_{mode,i}}} \qquad . \qquad (10)$$



where $Utility_{mode,i}$ represent the utility function of a transportation mode i.

MOIRAI is implemented in 3 main stages common to many ABTD models (Castiglione et al., 2015):

(a) Generating daily activities of the agent.
(b) Managing the time schedule of the agent.
(c) Identifying type of transport used for each trip.
These steps are described hereafter (see figure 4.a).

### 4.2.1 Generation of daily activities

One important step when modeling the time schedule of an individual is to estimate the number of trips this agent
makes per day as shown in figure 4.b. As this number is a function of priorities, the first step in generating an
agent's agenda is thus to define his priorities. The obligation to carry out an activity, such as going to work or
accompanying kids to the nursery, defines the priority of an agent. Once these priorities are defined, an agent can
perform optional activities such as shopping, visiting a friend, or going to the movies. There are 3 priorities in the
model. **Work – School - Accompanying** (bring a child under 10 years to school). And the model can combine
WORK and ACCOMPANYING priorities.
The number of daily trips ($p$) made by an agent depends on his priorities ($x$) and is based on discrete probabilities
distribution as follows:

$$P_{trips}(x) = \begin{cases} P_{trips\,act1}(x), if\ priorities = 1 \\ P_{trips\,act2}(x), if\ priorities = 2 \\ P_{trips\,other}(x), if\ priorities = 0 \\ P_{trips\,retired}(x), if\ agent\ age > 65 \end{cases}, \qquad (11)$$

where $P_{trips\,act1}(x), P_{trips\,act2}(x), P_{trips\,other}(x), P_{trips\,retired}(x)$ are the probability to make $p$ daily trips based
on $x$ agent priorities and with $x, p \in \mathbb{N}$.


The probability $P_{trips}(x)$ is derived from a specification of a daily trip number based on age and priorities. This
daily number of trips varies from one country to another. This parameter can be provided from local surveys, or
estimated from an aggregated survey database. We used the information provided by household travel surveys,
indicating that the mobility of children and old people is lower than the average population (20 - 60 years).


After determining the number of activities of the agent and prioritizing them, MOIRAI defines a tour sequencing
to estimate the number of activities and their order. The sequences may be a Home-Based Tour (HBT), a Multiple
Home-Based Tour (MHBT), or a Non-Home-Based Tour (NHABT) tour. Figure 5 presents the different types of
sequencing modelled in OLYMPUS.
Depending on the number of trips, it is possible to conduct a home-centered tour or several tours, one centered on
the place of residence and another one centered on other activities.
The model takes into account HBT, MHBT, escort tour, HBT with activities based on sub-tour. The probability to
make a single or several tours depends on the number of daily trips.


After having generated the agent's time schedule, the module locates each activity. According to the TAZ in which
the agent is located, the model estimates the place where the agent will have the greatest probability to carry out
the planned activities. This is done according to the attractiveness of the TAZ calculated by THEMIS, and using
the Huff random probability approach for the activity location choice. For the location of the WORK activity, we
use $P_{\Omega}(i,j)$ the probability of attractiveness for jobs center. For OTHER activities, the $P_{\Omega}(i,j)$ probability of
attractiveness is based on population density.

The last step of the generation of daily activities is the implementation of the time to provide a duration for all
activities as shown in figure 4.c. MOIRAI computes this parameter using conditionals probabilities with a time



5    step of 1 hour. The module affects a random value to the start time, depending on the agent priorities. If the sum of activities exceeds 24 hours there is a re-start of the simulation.
For the first activity, the start time is calculated by:

$$ACT_1^{st}(Act) = \begin{cases} st_{escort}(x),\ if\ Act = "escort" \\ st_{school}(x),\ if\ Act = "School" \\ st_{work}(x),\ if\ Act = "Work" \\ st_{other}(x),\ if\ Act = "Other" \end{cases}, \qquad (12)$$

Where $st_{escort}(x)$, $st_{school}(x)$, $st_{work}(x)$ and $st_{other}(x)$ represent the start time of the first activity of the day. The start time is based on a normal distribution as shown in Figure 6.

$$ACT_{Other}^{st}(Act) = itime + Dur(Act) \qquad (13)$$

Where:
$$itime(i) = \sum_0^{i-1} Dur_{Act}(j) + ACT_1^{st}(Act) \qquad (14)$$

20    And *Dur* represent the duration of activities, defined as a random variable in a truncated interval over a range of time. The distribution of activities start time and duration in OLYMPUS is presented in Figure 6.

### 4.2.2 Modal split

The modal choice is clearly a critical parameter for calculating pollutant emissions from urban mobility. In OLYMPUS, the set of simulated modes of travel includes the WALK (walking and cycling), PC (passenger car 25   and 2-wheels) and PT (including underground, bus, tram, and train) modes. The objective is to define the probability of the use of a specific mode of transport according to the utilities of the modes. The modal choice is obtained from the expression of the utility function for each transportation mode ($U_{mod,i}$). The utility value arises from the generalized cost of transport, (including time budget ($t_{budget}$), the perception $a$ of mode $i$ and from the monetary cost ($m_{cost}$)). In this calculation, the travel time is expressed as a weighting of transport time and 30   penalties (tolls, parking, traffic jams ...) named $P_{mod}$, that adds up to the total:

$$U_{mod,i} = m_{cost} + a * t_{budget} + P_{mod}, \qquad (15)$$

The utility function of the WALK transport mode mainly depends on the time cost of the travel. The WALK mode 35   average speed, $Speed_{WALK\ mean}$, is defined to be 3.6 km/h. Thus:

$$U_{mod,WALK} = \frac{OD_{distance}}{Speed_{WALK\ mean}} + P_{WALK}, \qquad (16)$$

Where $P_{WALK}$ represent walk penalties and the distance between the origin and the destination activity, $OD_{distance}$, 40   is based on the Great-circle distance calculation,

$$OD_d = Arc\ cos\ (\sin \varphi_A \times \sin \varphi_B + \cos \varphi_A \times \cos \varphi_B \times d\lambda), \quad (17)$$

where A and B respectively designate the origin and destinations points, $\varphi_A$, $\varphi_B$, $\lambda_a$ and $\lambda_a$ represent their latitudes 45   and longitudes, and d$\lambda$ = $\lambda_a$ − $\lambda_a$.

For the individual passenger car mode(PC), the utility function is defined as follows:

$$U_{mod,PC} = \frac{OD_{distance}}{Speed_{PC\ mean}} + Cost_{CAR} + P_{PC}, \qquad (18)$$

50

By default, the PC average speed, $Speed_{PC\ mean}$, in urban areas is defined to be 22.6 km/h. This value is based on (Hickman et al., 1999) and represents the average driving speed in urban areas as recorded during the MEET project. The $OD_{distance}$ is also based on the Great-circle distance equation (Eq. (17)). The $Cost_{CAR}$ variable accounts for the mean kilometric cost of the car use. The penalties are coded as an additional monetary cost such 55   as tolls, parking tickets, penalty for short distance trips, congestions and taxes, which can be summed to the




5 calculation of the PC utility. The time cost is calculated by computing the shortest path at the city scale. There, the computational time is considerably increased during this process due to the important number of agents.

As for the public transport (PT) utility function from one TAZ to another, we use the following equation:

10 $$U_{mod,PT} = t_{PT} + Cost_{PT} + P_{PT} \quad (19)$$

In this equation, the trip duration using the PT mode is $Time_{PT}$. It is a function of both the accessibility to the destination area and the average distance between the origin and destination points. The average transport time from one TAZ to another includes walking, waiting and travel duration. Its calculation is done by a linear
15 regression based on indications of transport times per zone and is thus based on realistic travel time.

$$t_{PT}(d,z) = \alpha_{i,j} \times a \times d + b + W_{Time}(\text{UTAI}), \quad (20)$$

where $\alpha_{i,j}$ represents the mean time spent in public transportation between 2 zones and a and b are the linear
20 regression coefficients. And $W_{Time}$ is the waiting time depending on UTAI.

The $t_{PT}$ parameter is usually calculated using General Transit Feed Specification (GTFS) data - if available for the city, and computed using the Connection Scan Algorithm (Dibbelt et al., 2013) or the RAPTOR algorithm (Delling et al., 2012). The limitation of these methods is the huge computational time required. As a consequence,
25 they were not considered here. However, since public transport time is an essential variable for the estimation of the general cost of public transport, we have developed a methodology based on a zonal approach and using the UTAI. This method has limitations compared to CSA or RAPTOR algorithms. However, an appropriate estimation of the UTAI matrix and a suitable calibration of the module with the real transport times leads to satisfactory results. The $Cost_{PT}$ variable represents the daily cost of the transit. The transit penalties can be represented by the
30 frequency of service of public transport.

**4.3 Assignment**

The transport demand previously generated by the ABTD module (MOIRAI) produces travel matrices that only supply information on the origins and destinations of the flows. The next step is to project on our grid the paths
35 taken by the agents, in order to further provide spatialized pollutant emissions from transport. For this purpose, we only take into account the flows related to private vehicle use.
There are 3 ways to deal with traffic assignment. One is the microscopic approach, which considers the traffic at the scale of each vehicle, as proposed by models like VISSIM (Gomes and May, 2004), AIMSUN (J. Barcelo, J.L. Ferrer, 1989) and PARAMICS (Cameron et al., 1994). A second approach is that of mesoscopic models, which
40 are interested in the evolution of sets of vehicles, as do the CONTRAM (Taylor, 2003) and DYNASMART (Mahmassani et al., 2005) models. Both approaches are not very compatible with the city scale we focus on. Indeed, although there may be an added value to running instantaneous emission models like PHEM (Rexeis et al., 2013) and MOVES (U.S. Environmental Protection Agency, 2013), obtaining input traffic data that describes every vehicle acceleration and deceleration cycles is quite challenging, and their consideration requires high
45 computational time. Both constraints make this microscopic approach somehow precarious. We thus have to rely on a macroscopic description of the traffic, in the form of a stream and using global variables such as vehicle flow and average speed on each section of a traffic axis, like what is done in the DAVISUM (Broquereau L., 1999) and TransCAD (Caliper Corporation, 2010) models. As most of these transport models are not open source, we opted for the development of our own traffic assignment model inside the OLYMPUS platform: HERMES.
50 HERMES is a macroscopic traffic module that works with average speed values for the vehicle flows, thus ignoring the dynamics of traffic within a road. This approach is compatible with our simulation scale. It is also compatible with the most common methods of estimating traffic-related combustion emissions, which rely on emission factors per driving cycle, each cycle being characterized by an environment (city, highway, etc..), and by a mean speed per strand.
55

There are main 4 stages in the allocation of agents to the road network in the HERMES module (See Figure 7.a).

 (a) Definition of the road graph
 (b) OD shortest path
60 (c) Goods and inter-regional transport modelling

(d)  Speed on link computation

First, the road network is extracted from GIS road data and transformed into a graph that records the connections
between the different road sections, thus creating a set of edges and nodes (intersections) using the graph theory
(Bondy and Murty, 1982). The speed limit is the main attributes of edges.

Second, HERMES proceeds to the computation of the shortest path for each trip by solving the Dijkstra algorithm
(Dijkstra, 1959). For each trip, the module identifies the nodes of the graph that are closest to the geo-referenced
O and D points. To choose the shortest path from the algorithm outputs, HERMES uses the time spent on a link
as a weighting.

In a third step, the integration of the regional traffic flow – including the goods and different patterns of inter-

regional transportation – is made. This complimentary step is necessary because the MOIRAI travel demand only
considers personal trips for agents living in the city. Inter-regional transportation, heavy-duty vehicle (HDV)
transportation and light commercial vehicle (LCV) transportation are thus not taken into account. This is why we
developed an approach that extrapolates the flow of goods and interregional transportation trips from a reference
ratio "passenger car / total fleet" and the HDV and LCV traffic from known ratios inferred at the urban scale.

Indeed, surveys on fleet composition are available for many cities. They are often based on transport organizations
like TFL in London.

Finally, HERMES proceeds to the integration of network congestion in its evaluation of mobility. Road congestion
alters speed on the road network as shown in figure 7.c. The approach is based on (UK Department of Transport,
1997) and can be represent as follows:

$$
S_{link} = \begin{cases} S_0 \ , \ if \ F_i < 50 \\ S_0 - \frac{S_0 - S_c}{F_c - F_0} \times (F_i - F_0), if \ F_0 < \ F_i < \ F_c \\ \frac{S_c}{\left[1 + \left(\frac{S_c}{8 \times d}\right) \times \left(\frac{F_i}{F_c} - 1\right)\right]} \ , if \ F_i > F_c \end{cases} , \qquad (21)
$$

where the speed on the link $S_{link}$ depends on the link flow $F_i$ and length $d$. $S_0$ is the free-flow speed, $S_c$ is the

congested speed, $F_c$ the link flow capacity.

This is one of the approaches suggested by (Ortuzar and Willumsen, 2011) to attempt to represent empirical
congestion. One limitation of this methodology is the consideration of the impact of signaling. Other congestion

functions like the one presented by (Akçelik, 1991) deal better with delays at the junctions. However, the approach
we chose was shown to produce a satisfactory estimation of traffic flows on the main roads. On the other hand,
this method requires knowledge of the location of traffic lights. For street scale studies, (Akçelik, 1991) method
adds a certain realism to the modeling of the traffic. At the city level, the approach developed in the assignment
module generates good estimations of the road network saturation.

**5 Building energy demand**

Figure 8 presents the flowchart of HESTIA, the OLYMPUS module in charge of simulating the building energy
demand. HESTIA uses the type of housing, the living area of the household and their average annual energy
consumption as input parameters. The main task of this module is to spatialize the energy demand in the territory.
(Swan and Ugursal, 2009) have proposed a review of models and methodologies for simulating the energy demand

of buildings. In this frame, both Top-Down and Bottom-Up approaches rely on the econometric, statistic and
engineering aspects of the energy demand. They are mainly developed to achieve a better understanding of the
efficiency and cost of energy policies. Due to its global approach, the Top-Down method lacks flexibility to create
scenarios involving a change in methodology. On the other hand, some of the input parameters considered in a
Bottom-Up approach go beyond what is feasible on a regional scale. They sinclude detailed data by type of

building (structural properties, equipment, usage) as well as individual parameters such as the orientation of
buildings in relation to the sun. In OLYMPUS, the combustion emissions modeling is carried out in two steps by
the HESTIA module. It lists combustion activities for residential, institutional and commercial heating, as well as
for hot water and cooking. The process is similar to Top-Down approaches, but the implementation of Bottom-Up
factors related to energy efficiency or household characteristics makes it possible to envisage the implementation

of energy scenarios.



The generation of the energy demand of the residential sector is done by modeling the energy demand of each
       household. It is a function of the size of the household, the size of the dwelling, the type of dwelling, the age of
       the dwelling and it also depends on a thermal efficiency coefficient. To generate the energy demand of the
       residential sector, HESTIA uses population density. The first step is to determine the ratio "collective to individual
       dwelling" as a function of population density and zone type, using GAIA outputs. This ratio is clearly dependent
on the country in question, and on local data such as building height or urban density.

       The calculation must therefore be specific to the area of interest. In HESTIA, household dwelling distribution
       (house/apartment) is formulated as follows:

$$P_{DW\ type}(x) = \begin{cases} P_{DW\ urb}(x), \ if \ zone = Urban \\ P_{DW\ sub}(x), \ if \ zone = Suburban, \\ P_{DW\ rur}(x), \ if \ zone = Rural \end{cases} \quad (22)$$

       We assume that CENTER and URBAN areas include a majority of buildings whereas SUBURBAN areas are the
       place where a larger part of individual houses are built. First, HESTIA begins by calculating the size of the
       dwelling ($DW_{zone}$), based on a reference size value ($Surf_{HT}$) for different type of dwelling which depends on
each specific zone ($\gamma_{UDI}$) and takes into account the number of agents ($n$) living in the housing:

       $$DW_{zone} = Surf_{HT} \times \gamma_{UDI} \times n \qquad (23)$$

       The energy used for heating and boiling water is defined by the distribution of the energy-mix, which is an
exogenous parameter referred in the model as:

       $$P_{energy} = \begin{cases} P(energy_1), \ if \ "House" \\ P(energy_2), \ if \ "Apartment"' \end{cases} \qquad (24)$$

       Then, HESTIA calculates the energy consumption per household, $EN_{cons}$, considering the size of the dwelling,
$DW_{zone}$, the unit energy consumption per household (ECU) and the size of the household ($hhsz$) :

       $$EN_{cons}(hh) = DW_{zone} \times ECU \times hhsz \qquad (25)$$

       Finally, the module applies a climatic correction to the energy consumption in order to estimate the under/over
consumption of energy due to the cold/hot climate. The degree-day (DD) is the parameter allowing to quantify
       this correction as a function of everyday temperature in the considered year, compared with a reference year (Jones
       and Harp, 1982).

       The calculation of the energy demand of the tertiary, institutional and commercial sectors is similar to that shown
above, though it is based on an annual energy consumption per employee ($ECUw$). Also, the spatialization of
       emissions is derived from the location of job centers and from their respective capacities (employment data by
       zone). Thus, the employee energy demand can be defined as:

       $$ENW_{cons}(employee) = ECUw \times n_{worker} \qquad (26)$$

       A climate correction is also added to the consumption of this sector.

## 6 Emissions

The calculation of pollutant emissions from both road transportation and building energy consumption is the role
       of the VULCAN module, which constitutes the last step of OLYMPUS. There, the quantification of pollutant
       emissions is based on methodologies recommended by the European environment agency (EEA) guidebook
       (European Environment Agency, 2013) for air pollutants and Green House Gases (GHG) emissions. They rely on
       the use of emissions factors, which may depend on the type of fuel, but also on the age and combustion technology
of engines and stoves. The VULCAN flowchart is shown in figure 9.a.





## 6.1 Road emissions

Emissions from road transport – labeled as mobile emissions in the inventory – are calculated along linear road sections where the traffic properties at a given time are homogeneous (driving cycle, average speed). As for passenger vehicles, the flow of traffic is derived from the travel matrices of the assignment module. From a quantitative point of view, emission factors depending on traffic characteristics are applied to each road section to obtain quantities of pollutants emitted into the atmosphere per unit of time. In literature, there are three main databases that offer exhaust emission factors. These are HBEFA (Keller et al., 2017), COPERT (Ntziachristos et al., 2009) and MOBILE6 (US EPA, 2003). They differ to the extent that some rely on instantaneous speeds, while others consider average driving speeds, or apply to specific driving cycles such as standard highway traffic. The methodology we developed for the emissions module VULCAN is based on the EEA recommendations, that is, it uses COPERT's emission factors based on the average speed of a vehicle during a standard driving cycle (see figure 9.b, c, d). In order to be exhaustive in the counting of traffic-related emissions, we have added mechanic particle emissions from different forms of friction and abrasion during driving, as well as VOC evaporation from vehicle tanks.

One critical step in the road transport emissions modeling process is to determine the composition of the vehicle fleet, which can be derived from national composition data as an exogenous data. In the assignment module, the affectation of specific emission factors depends on the fleet properties (age, cylinder, type of fuel). In VULCAN, the agent car properties are defined using a conditional probability law. One second important step is the addition of cold start emissions. This allows to take into account the effect of over-emission resulting from the poor performance of a vehicle starting and then running with a low-temperature engine. These supplementary emissions are also calculated from the EEA methodology (European Environment Agency, 2013).

Then, to obtain total exhaust emissions, VULCAN first calculates hot emissions factors for the stable engine regimes:

$$E_{hot} = N \times M \times \varepsilon_{hot} \qquad (27)$$

where N is the number of car on a link, M is the length of the link and $\varepsilon_{hot}$ is the emission factor.

Second, Vulcan calculates cold start emissions using an over-emission factor applied to a fraction of the distance traveled by each vehicle. This factor can be defined as

$$E_{cold} = \beta \times N \times M \times \varepsilon_{hot} \times \left( \frac{\varepsilon_{cold}}{hot} - 1 \right) \qquad (28)$$

where ß is the mean fraction of the total distance that is traveled with a cold engine, and $\frac{\varepsilon_{cold}}{hot}$ the ratio cold/hot.

The EEA offers several levels of refinement of calculations, called Tier, the use of which depends on the information available at the input of the calculation. Tier 1 methods are based on a simple linear relationship between activity data and emission factors, representing typical or averaged process conditions, which tend to be technology independent. More advanced Tier 2 methods are available for key categories, allowing in particular to apply country-specific emission factors that rely on process conditions, fuel qualities or abatement technologies (European Environment Agency, 2013). OLYMPUS uses each time the highest level of detail accessible. All emissions are then computed as follow:

$$E_{tier\ i} = N \times M \times \varepsilon_{tier\ i} \qquad (29)$$

where M is the number of travelled kilometers.

For instance, the calculation of LCV, HDV and 2-wheels emissions is based on the EEA tier 2 method. This methodology is used because of the excessive uncertainty on the fret fleet. The number of vehicle N is generated by HERMES using standard ratios of fleet composition. The emissions are calculated for CORINAIR pollutants (NOx, VOCs, PM) and for $CO_2$.

As mentioned above, emissions related to tire and brake wear are added to exhaust emissions, according to the two following equations (European Environment Agency, 2013):



$E_{tire} = N \times M \times \varepsilon_{TSP} \times f_s \times S_s(V)$      (30)
$E_{wear} = N \times M \times \varepsilon_{TSP} \times f_s$      (31)
where

$\varepsilon_{TSP}$ is the Total Suspended Particle (TSP) mass emission factor for vehicles in category j [g/km], $f_s$ is the mass fraction of TSP that can be attributed to particle size of class i and $S_s(V)$ is the correction factor for a mean vehicle
travelling at speed V.

Finally, the VULCAN module considers gasoline evaporation, using an aggregated still based on (European Environment Agency, 2013) :

$E_{evap} = \sum N \times \varepsilon_{evap} \times 365,$      (32)

where $\varepsilon_{evap}$ is the evaporation emission factor depending on ambient temperature.

### 6.2 Building emissions

Building emissions are based on the EEA guidebook for small combustion emissions (European Environment Agency, 2013). This part of the VULCAN modules considers emissions from residential heating (fireplace, stoves, cookers, small boilers), as well as institutional and commercial heating. Thus, small combustion emissions from the agricultural sector are not considered.

The calculation of residential and tertiary emissions is based on the EEA methodology and the emissions factors a based on (Pfeiffer et al., 2000) and (Kubica et al., 2007):

$E_{building} = \sum_{fuel}(\varepsilon_{fuel} \times EN_{cons}(hh))$      (33)

It is important to note that the composition and the age of the fleet are two crucial parameters affecting building
emissions. It has been found that improvement of combustion technologies has a massive impact on pollutant emissions over the years. However, due to a lack of information in literature, these parameters remain difficult to precisely estimate. For these reasons, when applying OLYMPUS on a territory, the hypotheses that we will be able to propose for the partition and the spatial distribution of the heating system technologies will be a determining point of the realism of the simulation.

## 7 Application to the greater Paris

The OLYMPUS model was first implemented on the Paris region. This choice is explained by the fact that the Ile-de-France region is critically exposed to urban spread, anthropogenic pollutant release and climate change, and that all the challenges of sustainable development are at stake. Undeniably, the Paris region is based on a large megacity urban structure, with high density of housing and an expanding peripheral urban area, clearly posing the
problem of mobility, traffic congestion and modal share. Moreover, the quality and availability of the input data make it possible to ensure robustness and reliability to the simulations undertaken. The choice of input data, working hypotheses and configurations selected for the OLYMPUS platform are presented below. The results from the emission calculations are then discussed.

### 7.1 Configuration

The simulations were carried out in 2009 due to an important database for this year (surveys, censuses, inventories). The simulation domain is the region Ile de France (Greater Paris). It is a monocentric urban area with a population density of 21,000 inhabitants / km² in the city center, and a decreasing density of the inner suburb and the outer suburb which are predominantly rural (Figure 10).
The population of the territory is more than 11 million inhabitants. In terms of transport infrastructure, Paris is the
city with the best public transport network. Individual mobility is 3.87 trips per person per day on average, with 41 million trips being completed each day in the region. The majority of travel does not include trips to the city center(70%): the majority of trips in Ile-de-France are short (4.4 km on average) and close to home.





The computing space unit used in OLYMPUS is the TAZ. Here, it is derived from the French National Institute of Statistics (INSEE). INSEE has set up a specific division of the territory called IRIS which includes a large number of inhabitants. For our domain, this choice leads to the constitution of 1300 TAZ. Figure 11 illustrates the IRIS border definition used by OLYMPUS.

Modeling anthropogenic emissions from the combustion of individuals requires a large database whose main
parameters and their sources are described in Table 1.

To generate the synthetic population in GAIA, we have used aggregated data from the city census, mainly derived from INSEE. They include the distribution of the population on the territory by age and sex, the number of households by IRIS and the average distribution of households by type of households (alone, couples, family, single-parent family). Mobility calculations in MOIRAI rely on several types of data, mainly surveys or national
statistical databases. First, the public transport accessibility of the domain was carried out on the basis of the density of public transport networks as provided by the regional transportation agency STIF. As for the attractiveness of the city subareas, the generation of attractive WORK zones is established from INSEE census data providing the number of jobs per municipality. The average distance at which agents can be lured by a professional activity is then derived from the city's overall transportation survey (STIF, 2012). The total mobility
of the agents is also conditioned by the total number of trips in a day, which is weighted according to the number of agent's priorities. Here, the mean daily number of trips was derived from local surveys, and we hypothesized that trips other than those related to occupational mobility have the same average distance, which is not the case in reality. For the category OTHER, the interest of agents for a given activity results from two main parameters: the number of households in the immediate vicinity of an activity, and the estimated average travel distance to
reach this activity. Once these values determined, the determining parameter for the realization of the activity in THEMIS will be the estimated travel duration inside the Paris region. Here, this parameter was derived from the calibration of the THEMIS module through an online application based on GTFS data from all regional transport agencies. It allows the constitution of a matrix of average transport times between the different classes of urbanized areas (UTAI). In the end, the combination of the transport network data and population density allows the
calculation of the accessibility of any activity area.

The vehicle fleet used in HERMES dates back to 2009 and is based on the (Carteret et al., 2015) survey. It includes passenger cars, LCVs and HDVs. This study is based on video observation to characterize a fleet of vehicles and fleets are compared to the global transport survey. The regional fleet of stoves and fireplaces was not estimated, so it was not included. We hypothesized here that individual heating modes, including wood heating, were mainly
from individual dwellings.

The energy demand on the territory was estimated from ARENE (Environmental and New Energies Regional Agency) survey data providing the unit consumption of households in Ile de France, but also from information on the average surface area of each household's dwelling, by type of household and as a function of the living area, as provided by CEREN (Center of Economic Research on Energy). The consumption modeling of the tertiary
sector was carried out on the basis of annual consumption per employee of the tertiary sector. (CEREN, 2015)

**7.2 Results and discussion**

Figure 11 shows the results of the modeling of the synthetic population, obtained by running probability functions from aggregated census data. It also shows that we obtain a realistic representation of household density variations in the territory. With regard to the characteristics of the population (Figure 11), we note that OLYMPUS faithfully
grasps the age distribution of the inhabitants of the Greater Paris, compared with the INSEE census data. However, it should be noted that OLYMPUS underestimates the senior population by about a factor of 2 - or even more for the last age group of the agents - and underestimates the infant population by 24%. However, these are age groups that are associated with low mobility, and with regard to the older age group, it accounts for a small share of total agents. Finally, the gender parity and the employment rate correspond to the census population. And these two
parameters are perfectly equal because they were taken as constraints. Based on the low percentage distribution error of the working population, we consider that the model generates an acceptable synthetic population for transport modeling. The other attributes of the agents also correspond with the forcing data, including the gender distribution, the unemployment rate, the distribution of the type of household and the average household size.

Because OLYMPUS relies on Bayesian statistics to generate a synthetic population, it is necessary - to get results
close to reality - to have a large database which offers specific information on the distribution of the characteristics of the agents of the population as a starting point. Thus, thanks to the transcription of stochastic variables, the synthetic population has great similarities with the population studied. Nevertheless, this approach produces limited variability in socio-economic parameters within the distribution, offering a simulated population rather close to the average characteristics of the actual population. In this simulation, we limited ourselves to the use of
a 3-level UDI and the division of TAZ into 1300 zones. It will be interesting to test the sensitivity of the distribution



of characters in the population to the increase in spatial variability and to the use of a larger number of indices and more of TAZ.

Agent mobility modeling was carried out using home-work origin-destination (OD) trips matrices generated by MORAI, on the basis of surveys employment data in the Greater Paris.

Figures 12.a, b are illustrations of the mobility of the synthetic population from OLYMPUS. Figures 12.b represent the saturation of the road network in terms of volume over capacities (VOC). The network in the city center and in the inner suburbs has an important VOC that is related to the monocentric nature of the megacity. This centrality of mobility is also noticed in figure 12.a, which represents the trajectories of the territory's movements with a strong orientation towards the heart of the megacity.

This result was compared with the mobility indicators from transport surveys. Table 2 shows the comparison of simulated and survey-based data on the average number and length of trips per day. The simulated data is very realistic, with only a 4% difference for the total number of trips per day and per agent in Ile-de-France, and a 6% overestimation for the average trip length, compared with average transport survey values. The total number of trips and the transportation modes are very close to reality. We observe here the ability of the model to faithfully

reproduce the distribution of the regional displacement demand.

Figure 12.c presents a map of the energy demand simulated over the Greater Paris. The results show a fairly logical positive dependence to total population, with a maximum demand in the center of the agglomeration. The final energy consumption the greater Paris is 303 TWh in 2009 according to (ARENE, 2013). And the residential and

tertiary sector represents more than 50% of these consumptions (including transport energy demand). The modeling of the regional energy demand (HESTIA) is quite satisfactory, the difference observed compared to the energy demand of the region is + 9.6%.

Emission modeling from both road transport and building heating was carried out by calculating linear and surface

emissions of atmospheric pollutants and greenhouse gases for each road section and consumption unit, followed by spatialization on the regular grid. The results, illustrated in Figure 12.d for nitrogen oxides (NOx) – a family of gaseous species emitted during combustion processes, show a very good coherence with the emitting structure in Île-de-France (major roads, types and density of housing by zone). Total OLYMPUS emissions are then compared to 2 reference emission inventories: that of the air quality network AIRPARIF, and that of the European network

EMEP. For this purpose, for each inventory, we extracted the activity sectors corresponding to the emissions calculated by OLYMPUS. The comparison is presented in the form of histograms in Figure 13, for NOx and for 2 size sections of particulate matter: $PM_{2.5}$ and $PM_{10}$. It should be noted that their methods of computing the emissions differ: AIRPARIF develops bottom-up approaches from local data gathering, while EMEP inventories arise from national emission totals per species, that are spatially disaggregated using top-down approaches. Furthermore, the

comparison with the EMEP inventory cannot be carried out in detail due to the lack of information on sub-sectors of activity in the EMEP data.

OLYMPUS emissions, although slightly lower than AIRPARIF emissions, present, for each pollutant, total values that are very satisfactory and speciation by activity (or vehicle) that reproduces the variability of the emissions

calculated by AIRPARIF. The only sector that falls outside this rule is residential combustion with a factor 2 of underestimation of the value of the AIRPARIF inventory for $PM_{10}$, but it is recalled that could not rely on local data and equipment technology is a major determinant of particulate emissions related to heating.

For nitrogen oxides (NOx) and the particulate matter ($PM_{2.5}$ and $PM_{10}$), the differences with the regional inventory are approximately 20% for road transport. Moreover, AIRPARIF has its own residential heating modeling hypotheses including emission factories that do not rely on local data as well as the fleet of chimneys, stoves, etc. The relationships between the sub-sectors are very similar between OLYMPUS and the AIRPARIF inventory. The discrepancies with the EMEP inventory are greater, which can be explained by the fact that the EMEP

approach is top-down.

The differences observed with AIRPARIF estimates of road traffic emissions are approximately 20% in most cases. Previous studies (Timmermans et al., 2013) have estimated that this corresponds to the expected gap between inventories operating on different modeling assumptions, whether it is the choices on the cold start

fraction, the fuel evaporation emissions modelling results from an aggregated methodology and especially the differences in the composition of the engine fleet. this makes the uncertainty on these emissions important. Based on these considerations, the estimate of emissions by OLYMPUS does not present a discrepancy with the AIRPARIF estimates that could give rise to fears of bias in the procedure. Nevertheless, from the quantitative




point of view, there remains a part of the total road traffic which is poorly estimated by OLYMPUS, since the model calculates the transport of goods on the basis of the track occupancy rates and does not take into account the interregional mobility, the city being considered here as a closed system. This largely explains the underestimation of traffic at the borders of the territory and in the most remote areas of the urban center. In addition, the issue of congestion is one that should not be overlooked in this type of model, because it affects the

decision of the agents and it contributes to an increase in emissions. At present, congestion is handled by the representation of speed classes on the main axes, but a more dynamic management of this process is desirable, for example by establishing an iterative process between congestion and choice of agents, and by the refinement of the representation of the speed according to the rate of occupation of the tracks. This is where the next developments of the platform will focus.

To go further in analyzing the results of OLYMPUS, we must consider the question of the choice of agents. Mobility modeling is based primarily on the theory of random utility in which each agent in the synthetic population is considered to make rational choices to select their mode of transport. This is a notion used by economists that obviously has several limitations. Principally, because imposing a complete rationality of decisions requires that agents have full access to information in order to make the most rational decision, which is

unrealistic. It is more likely that different agents will have access to partial and different sets of information when making decisions. Secondly, the time given to the agent to make his decision influences the final choice, which is not taken into account in this approach. Finally, the maximum utility of an action may not be the same for each person since it depends on the preferences and weights given to each of the various elements that compose the utility function. One must even consider that rationality is not the guide of all behaviors, which can sometimes

result from a habit, an impulse, etc. However, numerous economic studies have shown that this approach allows reliable predictions of typical behaviors of people within a group. It would be interesting in this context to test the response of the agent's mobility to the weighting of the utility function, or even to an increase of the variability of its expression. In the end, as for the number of total travels, the structure of the trips and the use of the modes of transport, the simulated results are quite close to reality, which validates the representation in OLYMPUS of the

average number of trips by agent, as well as the division of areas into attractiveness classes and the use of the utility function for modal choices.
However, it would be interesting to test the implementation of additional attractiveness classes that may change the spatial distribution of non-commuting mobility. In addition, the sequencing of activities that influence the temporal variability of activities is a parameter easily perfectible via the addition of a specific schedule according

to the characteristics of the agents of the synthetic population.

The emissions modeled by OLYMPUS are underestimated for the residential and tertiary sector. OLYMPUS overestimates energy consumption, which may explain the model's pollutant emissions for combustion in the residential and tertiary sectors. For the residential and tertiary sectors, there is no precise survey about combustion

technologies. This point explains the very large variability of the estimates of combustion emissions in the emission inventories. Thus, when we compare the EMEP and AIRPARIF estimates, their differences are of the same magnitude as the one we have with AIRPARIF. Our results are therefore in a range of values consistent with the estimates of the reference inventories.

Thus, comparison with other inventories shows us that most of the differences between our calculations and the reference inventories are of an order of magnitude consistent with the use of different working hypotheses, on parameters insufficiently constrained by surveys or censuses. Comparing results with existing databases validates the ability of the OLYMPUS platform to produce combustion and road traffic emissions from a population's activity. After evaluation, and considering the main improvements to come such as freight transport and

congestion, our modeling approach appears relevant for carrying out impact analyzes of transport and energy policies in a given territory.

## 8 Conclusions and perspectives

The OLYMPUS modeling platform has been developed to meet the need for the development of a tool that links the urban diagnostics provided by the different disciplinary models, in order to produce analyses of the effects of

urban policies on pollutant emissions, air quality and population exposure. OLYMPUS is a model providing emissions from road transport and energy consumption in the buildings by simulating the activities of a population. It is based on the description of a synthetic population made of agents, with their own characteristics and socio-economic parameters, and it was built from state-of-the-art algorithms and methods for determining urban mobility. In particular, it relies on the production of utility functions to determine the activity of agents within the

territory. It was designed to use



The results obtained with this platform show a quite good understanding of the emission resulting from individual and collective activities in the Paris region, though we had to make rude approximations concerning congestion, the transport of goods and the inter-regional mobility. The lack of constraint data on heating systems also induces uncertainty in the combustion emissions from the residential sector. Improvements on these issues have to be considered in future work.

Refinements could also be implemented in the representation of the choices of the agents. It would be interesting, subsequently, to introduce additional socio-economic segregation parameters (e.g. household income) and their impact on mobility choices. This could allow us to highlight social discrimination in our emission analysis. It may also be interesting to set up feedback loops between the modules of OLYMPUS in order to simulate their interactions and to develop a multi-agent model. For instance, we could couple the discrete choice module with

the displacement assignment module in order to make a feedback to integrate a realistic network congestion into the modal choice. One of our main perspective will also be to implement the model on other cities, so that we can test the transposability of our results, and have a broader vision of decision support in urban planning.

Finally, because it relies on the evaluation of pollutant emissions from the activity of its synthetic population, in connection with the nature and the functioning of the simulated territory, OLYMPUS can produce diagnoses of durability for differentiated situations: widespread cities, compact cities, cities oriented along transport corridors, etc. In this framework, the question of the realistic nature of the input data is no longer relevant, and OLYMPUS can produce innovative results on the emissions of pollutants and greenhouse gases as well as on the levers of energy consumption (urban development, behavior and awareness of populations, public transport offer, etc). This

is a very big step forward in the area of urban decision support.

**Code availability**
OLYMPUS is published as an integrated model of pollutant and greenhouse gas emissions. The source code can be obtained from the LISA website at [http://www.lisa.u-pec.fr/~aelessa/OLP](http://www.lisa.u-pec.fr/~aelessa/OLP) or upon request to the authors. The

version presented here corresponds to OLYMPUS v1.0. Some improvements will be made and OLYMPUS 1.0 will be updated for the latest version of the code.

**Acknowledgements**

This work was performed using HPC resources from GENCI-CCRT (Grant 2017- t2015017232). This work also received support from the French National Agency for Research (ANR-14-CE22-0013) and the Ile-de-France

region. We also acknowledge Airparif and the Labex urban futurs.

**Appendix A: Nomenclature**

**ABTD**   Activity-based travel demand
**GAIA**   Synthetic population generator module
**GIS**   Geographic information system
**HBT**   Home-Based Tour
**HDV**   Heavy duty vehicle
**HERMES** Trip assignment module
**HESTIA** Buildings energy demand module
**LCV**   Light commercial vehicle
**MHBT**  Multiple Home-Based Tour
**MOIRAI** Population agent mobility generator module
**NHBT** Non-Home-Based Tour
**OD** Origin-Destination
**OLYMPUS**   Integrated emissions model
**PC**   Passenger car
**PMF**   Probability Mass Functions
**TAZ**   travel analysis zones
**THEMIS**   Activity based travel demand preprocessing module
**UDI**   Urban density index
**UTAI**  Urban Transport Accessibility Index
**VULCAN**   Emissions module




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



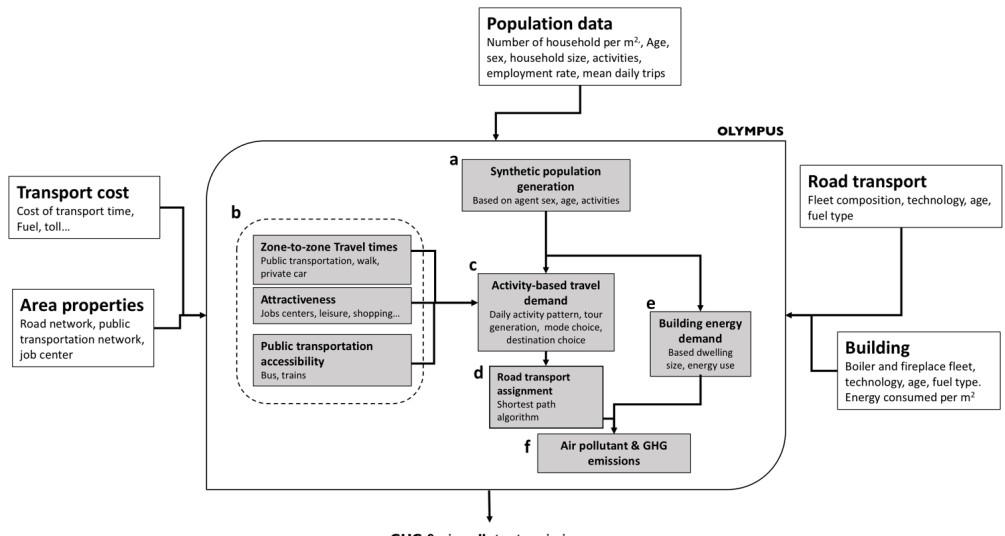

**Figure 1:** Flow chart showing the OLYMPUS emissions operating system, as well as its main modules (**a**) The synthetic population generation module (GAIA) (**b**) The generator of the transport time matrix, transportation accessibility indices and attractiveness of areas (THEMIS) (**c**) The transport demand module based on the activity of the synthetic population, and the modal choice in terms of transport (MOIRAI) (**d**) The module for assigning the travel demand on the road network (HERMES) (**e**) The module for the generation of energy demand at the regional level (**f**) The module for the calculation of greenhouse gases and air pollutant emissions based on emission factors.





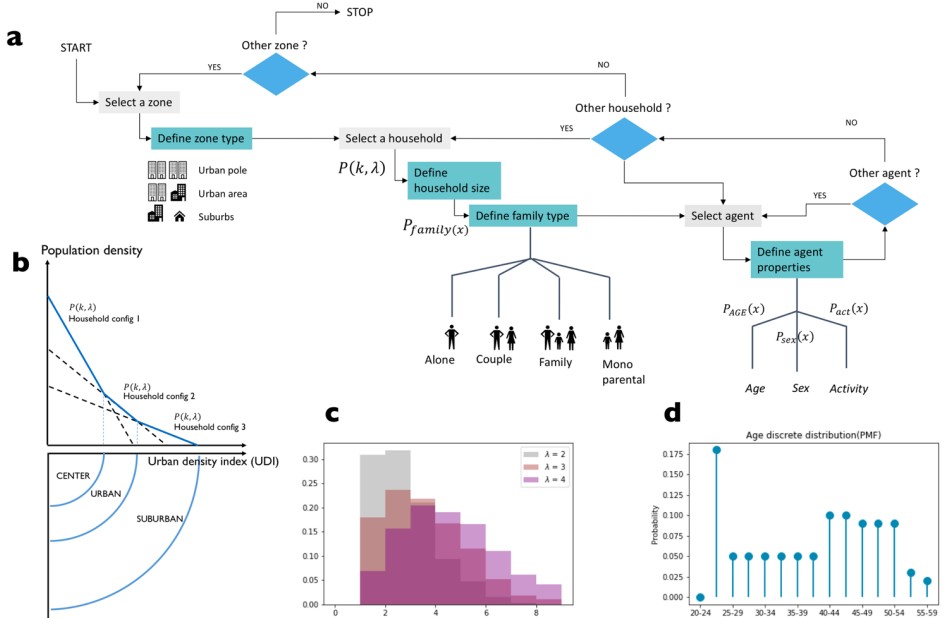

**Figure 2:** (a) synthetic population generator GAIA model operating flow chart. (b) Schematic representation of the urban density index (IDU). (c) Example of the household size probability distribution according to a truncated Poisson distribution. (d) Example of a representation of the distribution of the probability of mass function (PMF) of the age of a living agent alone.

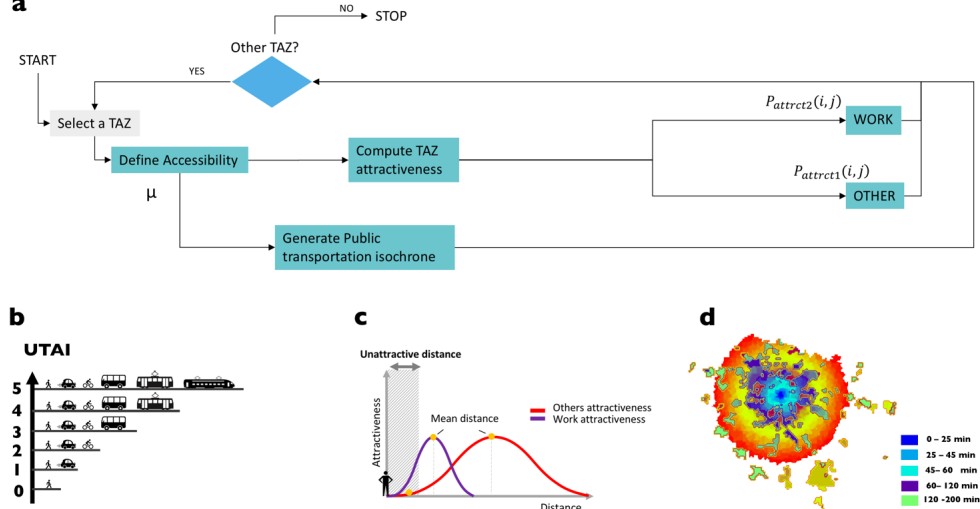

**Figure 3:** (a) The transport time matrix generator, transportation accessibility indices in common and attractiveness of zones of displacements (THEMIS) flow diagram. (b) Schematic representation of UTAI. (c) Schematic representation of the attractiveness of an activity towards an individual as a function of distance. (d) Example of isochronous transit curves from the center of Paris



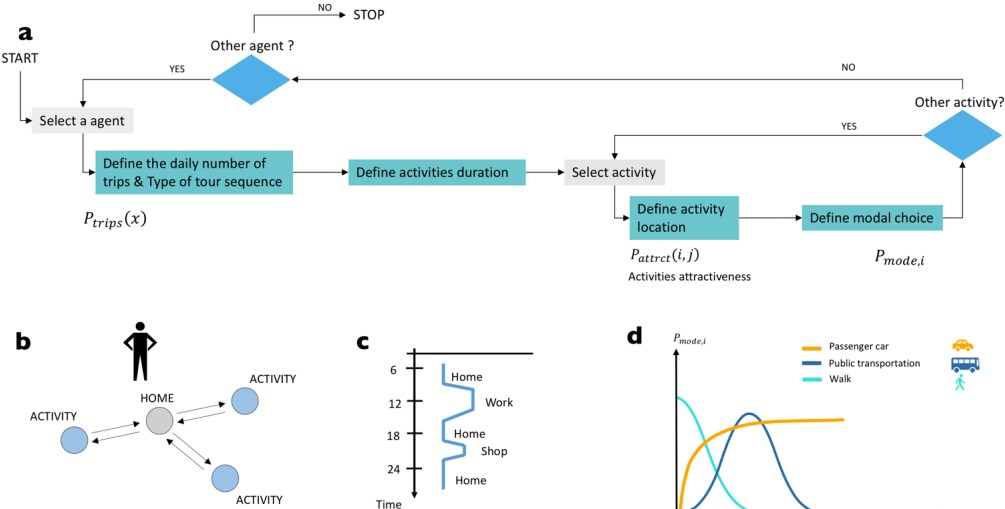

**Figure 4:** (**a**) The transport demand module MOIRAI operating flow diagram. (**b**) Representation of a circuit of activities of an agent of the synthetic population. (**c**) Representation of the timetable of an agent of the synthetic population. (**d**) representation of the probability of favoring a mode of transport according to the cost of transport time

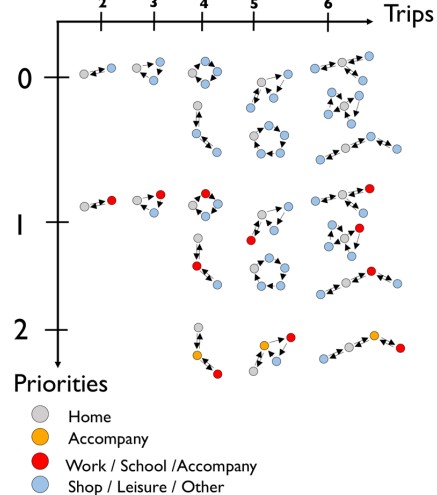

**Figure 5:** The transport demand module MOIRAI activities circuit based on agent priorities and daily number of trips (*p*)





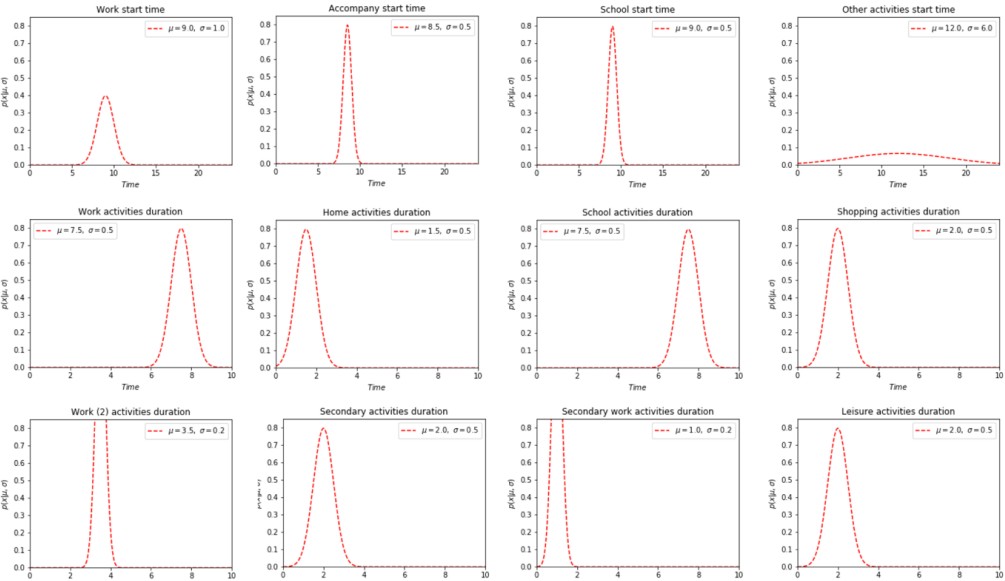

**Figure 6:** Distribution of the activities start time and individual time spent on an activity

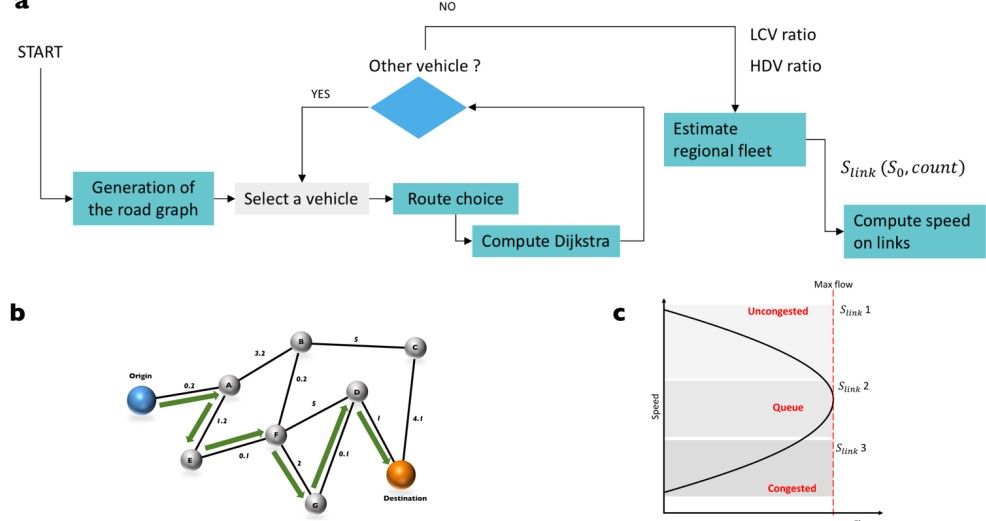

**Figure 7**: (**a**) operating diagram of the assignment of the transport demand on road network (HERMES). (**b**) representation of the calculation of the shortest path based on the speeds of road sections. (**c**) Speed flow curve of the MOIRAI module based on 3 levels of road saturation



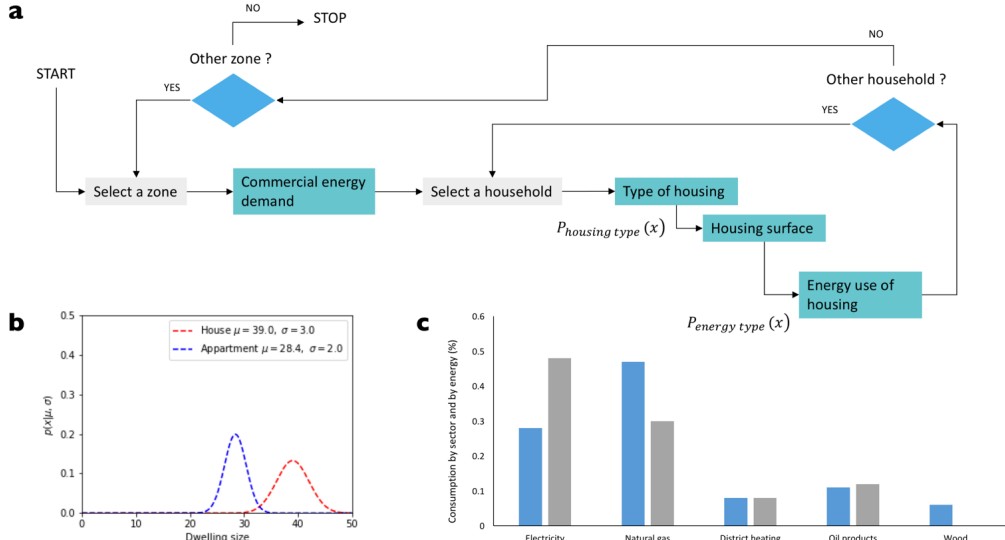

**Figure 8:** (**a**) Energy demand at the regional level generator (HESTIA) flow diagram, (**b**) Example of dwelling size distribution, (**c**) probability mass function of the type of energy consumed for different types of dwellings.

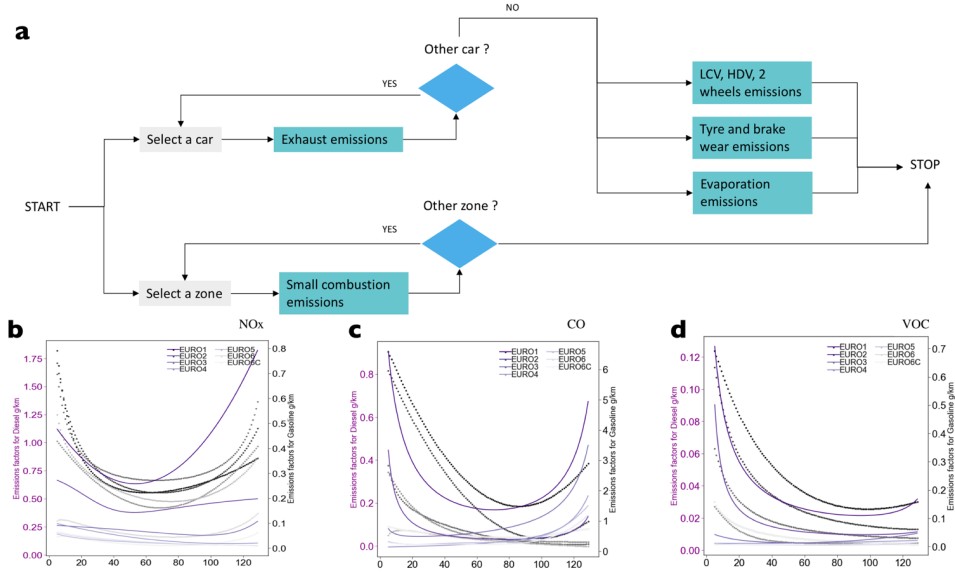

**Figure 9:** (**a**) Greenhouse gases and air pollutant emissions module (VULCAN) flow diagram, (**b**), (**c**), (**d**) represent NOx, CO and VOC emissions factors from diesel and gasoline passenger cars.



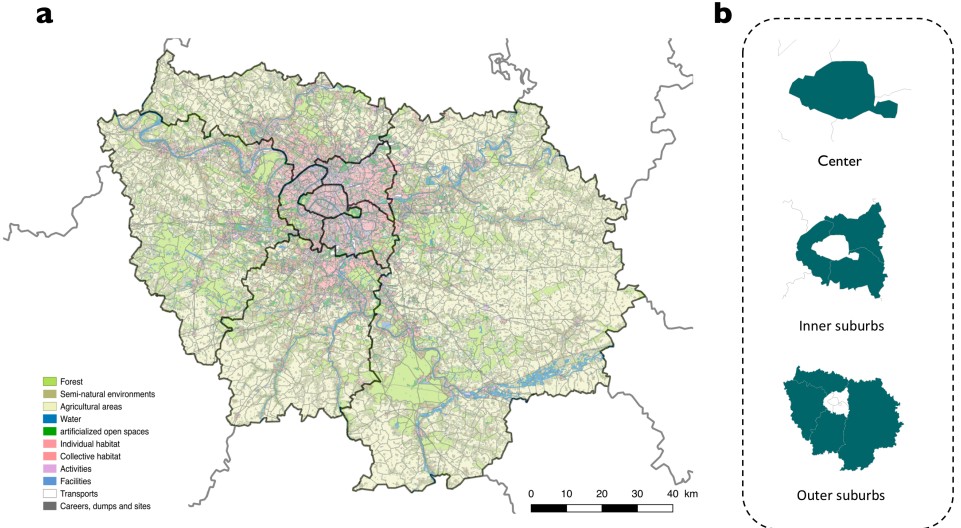

**Figure 10**: (**a**) Representation of the Ile de France region (Greater Paris) and the land use. (**b**) Representation of the Ile de France subdivision.

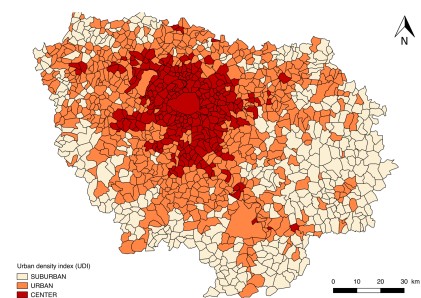

**Figure 11:** TAZ subdivision and Urban density index(UDI) of the greater Paris.



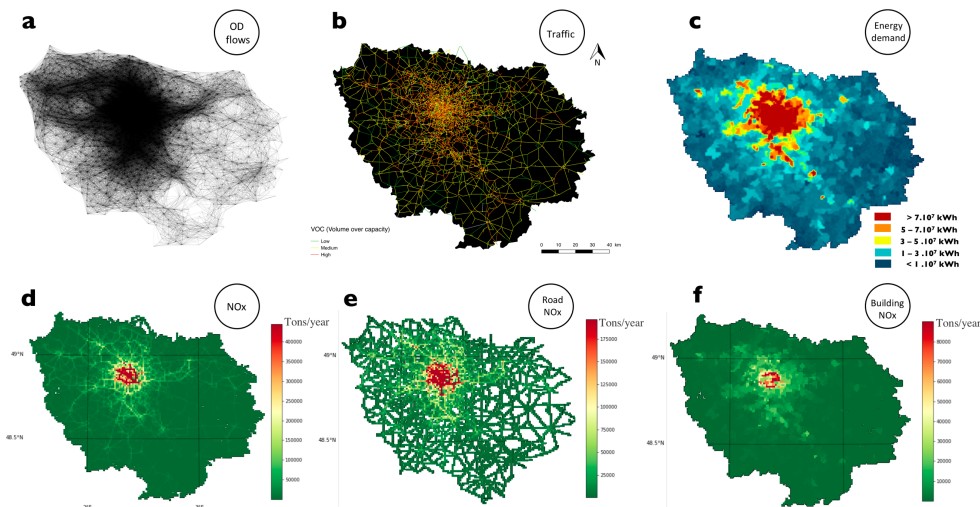

**Figure 12:** (a) representation of all origin-destination flows generated by MOIRAI motion request module. (b) Representation of the daily road traffic in the Greater Paris in terms of volume over capacity (VOC), (d) Nitrogen oxide emissions in Ile de France from road transport and residential / tertiary sector (OLYMPUS), (e) Focus on emissions from road transport, (f) Focus on emissions from the residential / tertiary sector.

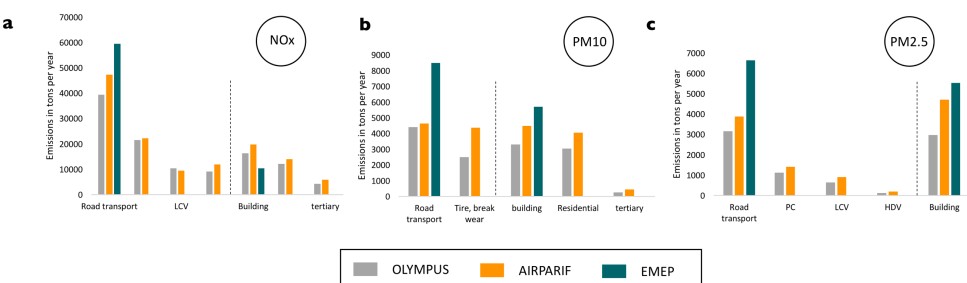

**Figure 13:** Emissions comparison with local and regional inventories (a) nitrogen oxides, (b) particulate matter with a diameter of 10 μm or less and (c) fine particles with a diameter of 2.5 μm or less.

| Module | Inputs | Sources | Description |
|---|---|---|---|
| Synthetic population | | | |
| | Number of household, household sex, age, Employment rate, household density | INSEE | These data are mainly derived from the census of the regional population |
| Travel demand preprocessor) | | | |
| | Transit stations | Ile de France | These databases are spatialized in GIS format. |
| | Job center | INSEE | |
| Activity based travel demand | | | |
| | Number of daily trips Mean transit travel time | STIF, DRIEA STIF | These data are mainly derived from surveys, |



| | | | |
|---|---|---|---|
| | Car cost | OMNIL | including the Global Transport Survey (EGT), the household-displacement survey (EMD), but also national statistics |
| | Public transportation ticket price | | |
| Road assignment | | | |
| | Road network | LVMT | The main road network |
| Building energy demand | | | |
| | Energy mix | ARENE | Greater Paris regional energy agency |
| | Energy use per m² | CEREN | surveys data |
| Building and transport emissions | | | |
| | Car fleet | Carteret et al., 2015 | Video fleet observation studies |

**Table 1: OLYMPUS parametrization for the Greater Paris simulation**

| | OLYMPUS | OMNIL | RD |
|---|---|---|---|
| **Average number of trips per day** | 4.05 | 3.87 | 4.6% |
| **Average length of a trip(km)** | 4.7 | 4.4 | 6.9% |
| **Total number of trips (millions)** | 41 | 41 | 1% |
| **Motorized individual trips** | 40.4% | 39.5% | + 0.9% |
| **walking, cycling** | 41.9% | 40.3% | + 1.6% |
| **Trips by public transportation** | 17.7% | 20.1% | -2.4% |

**Table 2**: Comparison of mobility with global transport surveys.