# Peer review of "OLYMPUS v1.0: Development of an integrated air pollutant and GHG urban emissions model - Methodology and calibration over the greater Paris"

_Geoscientific Model Development, 2018_

## Referee Comment (RC1) · Anonymous Referee #1 · 26 Aug 2018

Review OLYMPUS v1.0: Development of an integrated air pollutant and GHG urban emissions model – Methodology and calibration of the greater Paris

This manuscript proposes an environmental decision-support tool which couples a population, road transport, building energy and emission model to estimate the spatial distribution of pollutant emissions across a city with an application to Greater Paris. Having such a spatially explicit model is valuable for intervention design in case emission inventories are not available. Having said this, the developed model is relevant to advance air pollution mitigation measures and future work in this field, in particular for air pollution from transport. The authors propose a model of good scientific quality and my main comments are related to the necessity of introducing clarity to the presentation and a discussion of limitations.

In particular the introduction needs development since the argument of the model development and its contribution, scale and scope are not quite clear and the literature review not comprehensive enough. First, how does the model consider exposure? In my understanding OLYMPUS only estimates emissions, not exposure. The link between city configuration, emissions and exposure is not clear in the argument and references are missing. Further, the literature review or argument for emissions form building energy consumption is missing. This comment links to the contribution of the model, which is not stated explicitly in the manuscript. I suggest to mention this clearly. There are other models which consider emissions from transport and building energy consumption, what is the specific contribution of OLYMPUS? What can the model help to decide?

The authors argue that OLYMPUS "was developed with the aim of taking into account connections between the types of urban organisations, regulatory constraints, energy consumption behaviours and pollutant emissions" (line 40). Please be more specific (and consistent): how do you define urban organisations (or city configuration (l. 8), built environment (l. 44), morphology (l.56))? Is it simply population distribution? Please clarify and make the underlying assumptions explicit. Which "regulatory constraints" does OLYMPUS consider? This is in my opinion not discussed in the article. I suggest to be more specific, also on the scale of which OLYMPUS considers all these model components. How do you "heighten the role of the built environment" (l.43), which "political and economic forcing" (l.43) are considered in the model? This is not clear and in my opinion misleading as population distribution is the only factor defining the built environment and I don't recognize any political forces in the model. "[…] in the exposure of people to atmospheric pollutants" (l.43) – does the model have an exposure component which is not described in the manuscript?

Which pollutants are considered in the model and why these? Which pollutants are considered in the transport model and which in the energy model? The authors might want to be careful (and consistent) with the terminology at city scale: air pollutants, greenhouse gases, atmospheric pollutants etc. I assume the model only treats primary pollutants?

In general, the model considers much more detail for emissions from road transport than building energy consumption. How do the authors explain/justify this or discuss this as limitations?

The authors state that the household family type distribution etc is different between centre and suburb etc. How do you justify this assumption and how is it defined and implemented? Exogenously? What might be implications of this assumption? The authors might want to add this in the discussion.

It is mentioned that the approach produced a "satisfactory estimation" (l.36) and "generates good estimates" (l. 39). Further, "are very satisfactory" (l.44). How do you justify this?

There is the assumption that CENTER and URBAN areas include "the majority of building whereas suburban areas are the place where a larger part of individual houses are built" (l.17). How do you implement this assumption in the model, how do you justify it and how is it translated in the model? There are many assumption that it makes it difficult to understand how the model goes from population density to housing type, dwelling size etc.

How does the climate influence the transport model?

How does the calculation for hot and cold start emissions vary per pollutant type?

The conclusion needs development and further discussion on how future improvements could look like. Especially the authors state that one aim of the model is to analyse various policy options; yet, it only comes briefly in the conclusion (last paragraph). The authors might want to elaborate on this and discuss how this could be considered in the model. Further, model limitations need to be discussed further. For instance, how do you treat uncertainty?

Is the validation of the model done based on the same assumption as the model? That is, for instance, are both driven by the underling population density assumption and therefore might yield similar results?

The scale of the different models is not clear throughout the manuscript. Please make this more explicit.

The authors might want to add a discussion of global vs local emissions, also given that they refer to air pollution, atmospheric pollution, greenhouse gas emissions and even exposure. At which scale is your model relevant and why?

I suggest elaborating on the overall model workflow and rationale in 2.1 about the main characteristics of the model and how the different sub-models play together. This could help the reader understanding the big picture before diving into the specifics of each module.

Overall, it is a very interesting model. However, the description needs to be clearer, the argument for it strengthened and presented in such a way that the reader can understand well underlying assumptions, the rationale of choices made and limitations of the approach.

Detailed comments

P.3, l. 23: Does the ABTD consider work and leisure trips?

The paragraph starting in l. 36 (P.3) is not clear, there is confusion on the technological specifications.

P.3, l. 28: "dwelling size of the household"?

L. 14: what does this mean: "[…] we proceeded to the implementation of a spatial component in the distribution…" Please clarify.

L. 45: what is the meaning of mu?

There is no such format as a GIS format for data. This needs to be reviewed

Reference missing for the Huff random probability approach

The parameter "M" is defined twice: as the length of a link and as the number of km travelled

The same is true for VOC: Volume over capacities and emission type

What exactly is the definition of "tertiary emissions"?

What do you mean by "the age of the fleet" (l.29) in the building energy model? Please clarify this entire paragraph

Title and throughout the paper: It's "Greater Paris" not the greater Paris

How do you justify that Paris has "the best public transport network"? Further, there are references missing in this entire paragraph.

P.17: the last sentence is cuff-off.

P. 2: "individual mobility in the exposure of individuals"?

P. 34: What do you mean by "reference methodologies"?

References are in many cases not put correctly (inside brackets rather than in text)

References need to be provided for all model and data sources mentioned and/or used in the article

P. 16: does the model over- or underestimate? (l.56)

"numerous economic studies have shown[..]" please provide example references

Finally, the manuscript needs to be thoroughly English proof-read

---

## Referee Comment (RC2) · Anonymous Referee #2 · 27 Aug 2018

The paper addresses a relevant scientific modelling question, that of the estimation of atmospheric emissions in a urban area. The model is correctly developed and seems to present novelty; however the authors should more clearly identify what distinguishes OLYMPUS from other similar models. The manuscript should be reconsidered after revisions. It needs to be revised for its English since the language is sometimes incorrect and not always clear.Besides the English revision, my major reason of concern is as follows. In section 8, page 17 lines 53-55 is stated "The OLYMPUS modeling platform has been developed to meet the need for the development of a tool that links the urban

diagnostics provided by the different disciplinary models, in order to produce analyses of the effects of urban policies on pollutant emissions, air quality and population exposure." The issues of exposure, air quality and urban policies are mentioned often in the literature review, yet OLYMPUS does not address these issues. In fact as stated in the beginning of section 2 "The objective of this model is to estimate the pollutant emissions linked with energy-consuming urban activities", and that is factual and correct. Exposure estimation requires combining pollutants concentrations (air quality) with time. In turn, to have pollutant concentrations some kind of an air quality model is required. Also the pollutants covered in a emission model are not necessarily the same as the ones covered in exposure assessments, where secondary, short-lived, pollutants are highly relevant. Going back to the statement, more specifically the part where the authors mention that OLYMPUS can be used "to produce analyses of the effects of urban policies on pollutant emissions", the work presented does not allow verifying the ability of the model to do that, in fact it doesn't even address urban policies. How will the model "react" to a change in urban policies? how will the authors change the inputs to reflect different urban policies? how will the authors address urban configuration? what exactly is urban configuration? These are all questions that remain unanswered, since the model was not applied to alternative urban policies. The authors should be very clear in the manuscript about the scope of the tool and the case-study presented. Any further considerations must be either accompanied by further modelling applications or removed.

---

## Author Comment (AC1) · 23 Oct 2018

**Author's response to Anonymous Referee #1**

*Q1: How does the model consider exposure? In my understanding OLYMPUS only estimates emissions, not exposure.*

It was an abusive formulation to say that the model was intended to calculate exposure. In fact, the model provides elements that will be used in our modelling platform to improve the representation of urban population's exposure.

Indeed, the OLYMPUS model has been developed to fit into a modeling platform consisting of a set of disciplinary models. It was designed to produce a new generation of emission scenarios for Air Quality Models (AQMs), but also to simulate a set of new parameters that will be needed for the calculation of the exposure in the next steps of the modeling platform. These last parameters are (i) a synthetic population characterized in particular by place of residence and age and (ii) loops of individual mobility. After the AQM simulation, individual urban travels provided by OLYMPUS are intended to be confronted with the 4D (x,y,z,t) concentration fields of our air quality model CHIMERE* in order to link spatial mobility and concentrations at the hourly time step, and to create a space-time exposure budget. This is a totally innovative element in the field of air quality research. Without these parameters simulated by OLYMPUS, the modeling of exposure could only be "classical" (map crossing concentrations with population density).

The calculation of exposure was therefore mentioned because it is part of our scientific objectives, and that OLYMPUS was also thought for this feature. This is one of the great innovations of the OLYMPUS platform, even if it is not the subject of a presentation here since it is a question of describing how OLYMPUS works.

***We have better specified the link between OLYMPUS and the exposure calculation in the article.***

*CHIMERE is an air quality model (AQM) labeled as a "national tool" that is used to provide air quality forecasts in the regulatory framework of air quality monitoring, but also to conduct research on air pollution in many research laboratories in Europe.

*Q2: The link between city configuration, emissions and exposure is not clear in the argument and references are missing*

The paragraph has been rewritten to be more detailed and better referenced on the subject.

*Q3: The literature review or argument for emissions from building energy consumption is missing*

The emissions from building energy consumption literature are based on the EEA guidebook for small combustion emissions (European Environment Agency, 2013) as referred in the manuscript. The emissions factors a based on (Pfeiffer et al., 2000) and (Kubica et al., 2007). These references were carefully all included in the text.

*Q4: This comment links to the contribution of the model, which is not stated explicitly in the manuscript. There are other models which consider emissions from transport and building energy consumption, what is the specific contribution of OLYMPUS?*

• Compared with traffic models, OLYMPUS is not original because it is based on a classic approach and on known & robust equations (although it is a strength to rely on a formulation already tested).

• However, there are 2 specific contributions of OLYMPUS.

  ✓ The first one is to couple several disciplines with the final objective of air quality modeling. Indeed, OLYMPUS is dedicated to produce elements necessary for the modeling of AQ

5   scenarios, and not only to the urban traffic management. This is why - as mentioned above - it is included in an urban modeling platform. It is forced by a model of economy, land use and urban growth that provides urban scenarios (mainly described by land use, transportation network and population density). These scenario may consider densification or urban spread policies, as well as improved public transportation system, or city green belts… OLYMPUS

10  includes the outputs such as land use and transport supply to calculate the transport demand and then the city shape-dependent emissions, which will be provided to the CHIMERE AQM. This provides a **new generation of urban scenarios** for air quality issues.

✓   The second one is to provide a **new form of environmental decision support**, through an approach based on individual decision. Indeed, to estimate the modal share, OLYMPUS mostly

15  relies on the decision of each agent of its synthetic population (utility approach). OLYMPUS is thus able to take into account the possible "changes of practices" in mobility. The same may be considered for energy consumption, regarding the choice of heating mode. In the field of air quality research, this is totally new. This does not prevent that OLYMPUS also support classic emission scenarios. OLYMPUS is indeed able to consider a fleet of vehicles and boilers

20  corresponding to a particular future scenario: it can therefore take into account the emission policies based on improved technologies.

The data produced by OLYMPUS allows much more advanced scenarios than usual air quality models usually do (they only consider a regulatory percentage of emission reduction). OLYMPUS simulates the scenario elements needed to provide decision support that takes into account the

25  collective appropriation of environmental policies as well as the impact of urban forms on transport demand and mobility, which is absolutely not the case in the literature today. To our knowledge, OLYMPUS is the most comprehensive tool for conducting AQ / energy consumption scenarios inside an urban modeling platform.

30  *Q6: What can the model help to decide? I suggest to mention this clearly.*
OLYMPUS can produce, for the CHIMERE model, emission scenarios that estimate the changes in car and energy use (whether because of an incentive policy or an evolution of urban form) and their effects on energy consumption (climate diagnosis) and on air quality via concentration calculations in CHIMERE (environmental diagnosis).

35  For example, for the transport sector, the model can simulate the impact of transport cost, or improved transport supply on modal choice and on associated emissions. For the residential sector, the model can simulate the impact of thermal renovation on energy consumption. The model can also be set on car fleet composition, stoves composition, housing type composition…

OLYMPUS can therefore help to understand the link between urban form, transport supply and

40  mobility and consumption practices. The scenarios conducted with OLYMPUS make it possible to predict the impact of sprawl or urban densification on road emissions, but also the extent of the behavioral changes required to change the state of play and improve air quality.

45  *Q7: How do you define urban organisations (or city configuration (l. 8)), built environment (l. 44), morphology (l.56))? Is it simply population distribution?*
The city **morphology** (or the urban area morphology) is the pattern of its occupation of space as measured by the density and the degree of hierarchization of the different clusters that compose it. We may also call it the urban form.

5      Indeed, we consider the city with a systemic approach (this is why we use urban system for larger urban area including several cities such as the Paris area). Our definition of **urban organization** is the way in which individual activities are organized in the built space. The urban organization is the way in which the urban form, the distribution of employment areas and the transport supply impose spatial interactions between individuals. It is one global parameter, even if it is the result of individual actions

10     conducted without care of the collective. We are interested almost exclusively here in the "mobility" aspect of the spatial interactions between city and individuals. In the model, the parameters designing urban configuration are population distribution, transport infrastructures, buildings, job centers in the city. It may be observed through the UDI and UTAI index mapping.

       Urban organization is the result of population interacting with the **city configuration**, that refers both

15     to the spatial configuration of the building area (the so-called **built environment**) and to the spatial configuration of the services and networks.

       All definitions are based on those of Bretagnolle et al. (2010). Here is a quote about urban organization:
       *"The organization of urban systems can be described on three main levels [including] the meso-level [that] corresponds to the city itself (when really defined as a consistent geographical entity). (…). This organization is shaped by interactions operating*

20     *on different spatial and temporal scales of observation. We can visualize a city as the envelope for the daily activities of its inhabitants and the buildings hosting them. (…). In order to correctly identify a city as a consistent geographical entity, and considering that its spatial expansion as well as its in situ development are part of its growth, we define it as a "daily urban system" (this concept (…) allows for frequent social interactions which usually take place within one day). While the maximum spatial development of cities has continuously been constrained by this typical one-hour time-budget, the speed of means of*

25     *transportation available has enabled a multiplication by a factor ten of the average commuting distance between places of work or urban services and places of residence, in the last two centuries. (…) A behavioral parameter defined for spatial interactions at the individual level is reflected in the organization of the urban entity at a higher level".*

       ***Anne Bretagnolle, Denise Pumain, Céline Vacchiani-Marcuzzo. The organisation of urban systems. D. Lane, D. Pumain, S. Van der Leeuw, G. West. Complexity perspective in innovation and social***

30     ***change, Springer, pp.197-220, 2009.***

       ***This has been clarified in the text, and the use of these terms has been made more rigorous.***

       **Q8:** *Which "regulatory constraints" does OLYMPUS consider?*
       As mentioned above, OLYMPUS can integrate all the regulations concerning the technological

35     improvements of combustions: penetration of the best technologies for the stoves and evolution of the road fleet. They may also include energy requirements for new buildings or for the renovation of certain neighborhoods.

       But public policies taken into account in our platform can also include recommendations on the urban planning scheme (these are taken into account in the economy and land use model and transferred to

40     OLYMPUS via land use and housing density parameters). The also may plan the development of the public transport offer (which will be directly taken into account in OLYMPUS via transport times for the modal share simulation).

       ***All these clarifications that we make here have been introduced in the text to better clarify the role of OLYMPUS, its contribution, its specificities and the limits of its fields of action.***

45     **Q9:** *How do you "heighten the role of the built environment" (l.43)?*
       It is a vocabulary mistake, we meant "highlight the role of the built environment" (the built environment being understood as the built city configuration described above). Indeed, the OLYMPUS functionalities described above allow to link the built city configuration to environmental city parameters.

  ***This has been clarified in the text.***

Part of the economic and political forcing are taken into account upstream of OLYMPUS, in the model of economy and land use: regional master plans regulating urban development and growth, urban mobility plans, energy costs... OLYMPUS is able to indirectly integrate these economic and political
10    forcing through its insertion into the modeling platform, downstream of the LUTI.

However, OLYMPUS can also produce a "mobility" or "energy consumption" diagnosis from financial incentive policies or environmental policies. Firstly because of its ability to estimate the modal share (car / public transport) thanks to a utility approach for each agent : it can consider the impact on modal choice and urban congestion (and therefore on pollutant emissions) of a policy of lowering / increasing
15    the costs of public transport, strengthening the supply of public transport, creating urban tolls, taxing fossil fuels… A political forcing like an urban toll can be added in the model by adding a penalty on the cost of using the car to the city center, in the utility function. Second, by evaluating the gain of a proactive policy on mobility, for example electric: OLYMPUS allows linking the share of electric vehicles, the type of population concerned (urban / suburban / rural), the expected emission reduction
20    and the improvement of air quality.

*Q11: "[...] in the exposure of people to atmospheric pollutants" (l.43) – does the model have an exposure component which is not described in the manuscript?*
No, see answer to Q1.

25    ***This sentence has been clarified.***

*Q12: Which pollutants are considered in the model and why these? Which pollutants are considered in the transport model and which in the energy model? The authors might want to be careful (and consistent) with the terminology at city scale: air pollutants, greenhouse gases, atmospheric pollutants etc. I assume the model only
30    treats primary pollutants?*
We are not sure what you call "the transport model" and "the energy model" because all emissions are processed inside one module: VULCAN. Of course, these calculations are differentiated within the module since they use different equations and different emission factor databases. However, OLYMPUS considers the emission of the following primary pollutants, whether emitted by transport,
35    building heating or both: $NO$, $NO_2$, Non-Methane VOC, $CO$, $CO_2$, $PM_{2.5}$, $PM_{10}$, $SO_2$ and $CH_4$, according to the procedures followed by the VULCAN module. A speciation profile for NMVOCs and NOx is applied to emissions, depending on the type of activity (heating, trafic) and distinguishing even the type of car (fuel, technology) (European Environment Agency, 2013).

OLYMPUS is an emission model. Thus, it does not calculate pollutant dispersion and transformation,
40    but only emissions. As such, there is no reason to consider secondary pollutants in its calculations. The primary pollutants mentioned above are the main input expected by chemistry-transport models. The use of these emissions in the CHIMERE model will thus make it possible to conduct a classic air quality modeling (that is to say, to calculate the dispersion and the physical and chemical transformations of the emitted species, in order to produce concentration fields of all the primary and secondary species
45    of atmospheric interest). Thus, the entire representation of atmospheric chemistry is possible with pollutants treated with OLYMPUS.

*Q13: In general, the model considers much more detail for emissions from road transport than building energy consumption. How do the authors explain/justify this or discuss this as limitations?*

5 - If we look at what is done upstream of VULCAN, when calculating the "traffic" or "heating" activity. There is indeed a difference in the number of modules dedicated to each type of activity. This is explained by the fact that for the traffic it is necessary to identify the origin and the destination of each path, the choice of the mode, the spatialization of the route, the congestion on the roads ... We must therefore represent the whole decision-making process and logic that lead to the elaboration of the

10 individual travels and simulate each lever of mobility. The emissions obviously depend very strongly on these parameters. None of this happens in the "building heating" activity, which is linked to the living area and the type of appliance. There is no spatial variability of the activity during the day, so no trajectory calculation and no choice to simulate. We have therefore simply identified the places of emission from population density data, applied a speciation of the heating modes corresponding to

15 the surveys, and calculated the energy needs according to the statistical properties (size, thermal insulation, etc.) of the housings. There is no reason to consider that it induces limitations for the representation of the heating activity.

- If we look at the assumptions used in VULCAN, indeed, there are far fewer factors that must be considered for emissions from boilers and stove than for cars. On the one hand there is no speed effect

20 that changes emissions, nor emission fraction with cold or hot engine. The variations of activity according to the season (external T) are taken into account. However, it is clear that we do not take into account elements that have an effect on the emissions (power of the equipment, effect of aging with identical technology, heating practices) because there is no technical data on the evolution of the emissions with the obsolescence, and very little data on the impact of the use of the equipment. As

25 much as it is important to use the data characterizing the types of engines and the uses for the traffic, as much imposing uses and coefficients of obsolescence for the heating is impossible. Using default choices would be absolutely unverifiable. In the same way, the data relating to the distribution of the technologies and types of energy used (wood, gas, electricity...) are partial and very little detailed. Having access to the data of a fleet of boiler stoves (for small combustion) is not easy. In the case of

30 the Paris area these data were not available. This is a limitation of the model, but not that of OLYMPUS, of all models that aim to simulate emissions related to building heating over a region. It may be interesting in the future to implement in a next version of OLYMPUS a more complex methodology that would allow you to choose more details about equipment and practices, but it would be up to the user to produce the required input data.

35 *Q14: The authors state that the household family type distribution etc is different between centre and suburb etc. How do you justify this assumption and how is it defined and implemented? Exogenously? What might be implications of this assumption? The authors might want to add this in the discussion.*
The household family type distribution is different between Center-Suburb. Our hypothesis is based on different studies (Pisman et al., 2011; Thomas et al., 2015). In those studies, the majority of single

40 people live in the city center. This is an exogenous parameter taken into account in the model. People living alone are mainly students and seniors. They have different behaviors (mobility, heating, boiling), which implies different emissions.

The question of the uncertainty induced by each input parameter of a model is recurrent. It can be treated by forcing or comparing this modeled parameter with field data, or by sensitivity studies to

45 this parameter. We chose this first path. For the case of Paris we used the INSEE census as a forcing parameter.

https://www.apur.org/sites/default/files/documents/taille_moyenne_menages_ile_de_france.pdf.

5    If we want to understand the impact of the distribution of household family types on a territory, we could change this "household distribution" parameter. These tests could be carried out with the model, in order to improve the knowledge on how a distribution of households in a territory can affect energy consumption. However, these test would not directly question the relevance of our household distribution, since it is forced by survey data.

*Q15: It is mentioned that the approach produced a "satisfactory estimation" (l.36) and "generates good estimates" (l. 39). Further, "are very satisfactory" (l.44). How do you justify this?*

That's right, these expressions are associated with a qualitative assessment that is not clear to the reader. So we worked to make clearer the quantitative elements of the comparison and we explained

15    by more precise sentences what could be understood by "satisfactory" for example.

The validation of the model was carried out by comparing its emissions with those of the Paris Air Quality Monitoring Agency (AIRPARIF), considered as a reference. The comparison shows a difference of less than 20% of the total emissions but also for most of the subsectors of activity and for the main species of interest.

20    This result is satisfactory for two reasons:

1) First because the typical range of uncertainties for road transport and small combustion is estimated between 50% and 200% (European Environment Agency, 2013) and we have less than 20% difference with the local bottom-up reference inventory. https://www.eea.europa.eu/publications/emep-eea-guidebook-2016/part-a-general-guidance-chapters/5-uncertainties-2016 )

2) Second because the approaches used for the calculations in OLYMPUS are very different from that of AIRPARIF. Although we use the same COPERT IV methodology for emission factors, OLYMPUS uses individual travel modeling while AIRPARIF relies on the traffic surveys of the year. And it has been shown that differences in the methodology of road emission inventories lead to a minimum of 20%

30    difference in total emissions, which is higher than the gap between OLYMPUS outputs and AIRPARIF data.

*Q16: There is the assumption that CENTER and URBAN areas include "the majority of building whereas suburban areas are the place where a larger part of individual houses are built" (l.17). How do you implement this*

35    *assumption in the model, how do you justify it and how is it translated in the model? There are many assumption that it makes it difficult to understand how the model goes from population density to housing type, dwelling size etc.*

If we try to reproduce a real situation (and therefore we are not in a scenario where these properties can be postulates), the model relies on external data to provide the properties of the households: it is

40    the distribution of the type of housing type by urban density index (UDI) and average dwelling size for different UDIs.

  ✓  For each household the model defines the housing type based on the location of the household and the UDI

  ✓  The dwelling of the housing based on the type of housing and the UDI.

45    The simulation of those parameters is based on a probability mass function.

*We have clarified this in the text.*

*Q17: How does the climate influence the transport model?*

5  The ambient temperature is an important parameter in the cold start engine emissions for the passenger cars as referred in EEA guidebook (European Environment Agency, 2013). It is also important for the energy demand in the buildings, but this is not linked with transport.

*Q18: How does the calculation for hot and cold start emissions vary per pollutant type?*

10  It is based on COPERT IV. Here is a description of the COPERT IV calculations we used at the highest level of refinement (called Tier 3, corresponding to vehicle with km and mean travelling speed available per mode and vehicle technology). Tier 2 and Tier 1 are more simple calculations derived from this procedure, for less informed emission vehicles. This text is extracted from the EMEP/EEA air pollutant emission inventory guidebook 2016.

15  *About emissions during transient thermal engine operation (termed 'cold-start' emissions).*

*It should be noted that, in this context, the word 'engine' is used as shorthand for 'engine and any exhaust aftertreatment devices'. The distinction between emissions during the 'hot' stabilised phase and the transient 'warming-up' phase is necessary because of the substantial difference in vehicle emission performance during these two conditions. (…)*

20   ✓ *They occur for all vehicle categories, but emission factors are only available, or can be reasonably estimated, for petrol, diesel and LPG cars and — assuming that these vehicles behave like passenger cars — light commercial vehicles, so that only these categories are covered by the methodology. Moreover, they are not considered to be a function of vehicle age.*

   ✓ *A relevant factor, corresponding to the ratio of cold over hot emissions, is applied to the fraction of kilometres driven with a cold engine. This factor varies from country to country. Driving behaviour (varying trip lengths) and climatic*
25  *conditions affect the time required to warm up the engine and/or the catalyst, and hence the fraction of a trip driven with a cold engine.*

The cold/hot emission quotient eCOLD/eHOT result from complex calculations. The calculation algorithm strongly depends on Euro technology, on the ambient temperature and on the pollutant being considered. There is not simple summary about them as there are several reports and updates
30  about these algorithms. It was specified in the text.

*Q19: how do you treat uncertainty?*

The stake for the model is more to appreciate its relevance on the final data with respect to a reference value, than to give a value of uncertainty. Indeed, uncertainty has a variety of possible causes (the
35  impacts of the uncertainty / error on the input data, the effect of simplified algorithms, the assumptions of statistical distribution of some parameters and the use of equations of rational choice, among others) and it doesn't make sense. Furthermore, the calculation of error propagation in models is not appropriate. It gives final uncertainties far too high, which are not consistent with the differences observed between the model outputs and the reference data.

40  For OLYMPUS, the method chosen for the evaluation of the uncertainties is to extract the intermediate (from each OLYMPUS module) and final (OLYMPUS emissions) output data and compare them to the reference datasets (surveys, inventories, measures...) when they exist. This was done for every module, and it is presented at the end of the article. The differences with the reference values allow us to estimate the uncertainty (in the statistical sense of the term) on our final value. Finally, the
45  multiplication of study cases and periods of study make it possible to better evaluate the mode of operation of the model. This is a classic approach and it is also the one we chose for OLYMPUS. Two sites of study (a polycentric urban zone and a coastal zone) are being simulated. Their presentation is beyond the scope of this article, which presents the structure of OLYMPUS.

5    However, evaluation is not sufficiently investigated in the current version of the paper. We took care to better explain the way in which the relevance of OLYMPUS outputs was assessed.

*Q20: Is the validation of the model done based on the same assumption as the model? That is, for instance, are both driven by the underling population density assumption and therefore might yield similar results?*

10   No, we don't use the same values for forcing and evaluation.

*Q21: The conclusion needs development and further discussion on how future improvements could look like. Especially the authors state that one aim of the model is to analyse various policy options; yet, it only comes briefly in the conclusion (last paragraph). The authors might want to elaborate on this and discuss how this could be considered in the model. Further, model limitations need to be discussed.*

15   These points have been more deeply discussed. The necessary improvements of the model were identified either by the fact that we didn't consider certain parameters, by the lack of robustness of an approach, or by the wish to develop new functionalities. In all cases, they rely on research leads, for which we do not always have yet a suitable methodology. For OLYMPUS, the objective is to decompartmentalize urban environmental issues, and it is a field in which the procedures are not easily 20   transferable from one discipline to another.

As far as policy options are concerned, we have identified a number of them, the purpose and scope of which have been detailed, but whose details are also outside the scope of the article.

*Q21: The scale of the different models is not clear throughout the manuscript. Please make this more explicit. // At which scale is your model relevant and why?*

25   The model is designed to produce emissions at the scale of an urban area, which can be a city and its suburbs, or even an entire region. In the case of Paris, the area is about 150 km wide. It's the same scale for all modules in OLYMPUS. Nous considérons en effet qu'une ville et ses environs constituent un système lié par les activités journalières, et c'est l'échelle que décrit OLYMPUS.

To calculate the mobility matrices, the modules use the Transport Accessibility Zones (TAZ) as smaller 30   administrative units, and have a linear approach to project the travels on the road network from one TAZ to another. For energy demand calculations they use data at the resolution of a municipality. Once emissions are calculated, everything is then reprojected on a regular kilometric mesh. OLYMPUS produces emissions at a 1x1km² resolution, which is the highest resolution usable in the CHIMERE AQM.

*Q22: The authors might want to add a discussion of global vs local emissions, also given that they refer to air pollution, atmospheric pollution, greenhouse gas emissions and even exposure*

The link with the large scale is ensured by the fact that the CHIMERE simulation is forced to its limits by a simulation of larger scale (nesting one-way). Therefore pollutants emitted in the central zone 40   (focus zone) can interact with pollutants imported from larger scales. In the CHIMERE simulation, pollutants and GHG emitted in the focus zone cannot alter the larger scale as there is no two-way nesting. In air quality issues, the area of highest spatial resolution (where the emission processes are represented with the most detailed approach) is generally the one on which we evaluate the environmental impacts.

45   The interest of calculating some GHG gases is only to diagnose the climate impact of a scenario.

*Q23: I suggest elaborating on the overall model workflow and rationale in 2.1 about the main characteristics of the model and how the different sub-models play together. This could help the reader understanding the big picture before diving into the specifics of each module.*

5 Figure 1 summarizes quite well the different stages of emission modeling. Especially when the article will be formatted, it will be easier to understand the different processes.

*Reference -*

[Figure]

**Figure 1:** Flow chart showing the OLYMPUS emissions operating system, as well as its main modules (**a**) The synthetic population generation module (GAIA) (**b**) The generator of the transport time matrix, transportation accessibility indices and attractiveness of areas (THEMIS) (**c**) The transport demand module based on the activity of the synthetic population, and the modal choice in terms of transport (MOIRAI) (**d**) The module for assigning the travel demand on the road network (HERMES) (**e**) The module for the generation of energy demand at the regional level (**f**) The module for the calculation of greenhouse gases and air pollutant emissions based on emission factors.

*P.3, l. 23: Does the ABTD consider work and leisure trips?*
Yes

*The paragraph starting in l. 36 (P.3) is not clear, there is confusion on the technological specifications.*
*P.3, l. 28: "dwelling size of the household"?*
OK, see changes in the manuscript

*L. 14: what does this mean: "[...] we proceeded to the implementation of a spatial component in the distribution..." Please clarify.*
it was clarified in the manuscript

*L. 45: what is the meaning of mu?*
Mu represent the household and transit density.

*This needs to be reviewed  Reference missing for the Huff random probability approach*
Huff reference is at P7. L.25

*The parameter "M" is defined twice: as the length of a link and as the number of km travelled. The same is true for VOC: Volume over capacities and emission type*
Indeed. We changed the name.

*What exactly is the definition of "tertiary emissions"?*
They are emission from the tertiary sector, which produces services. The sector is composed of activities related to trade and services provided to businesses and individuals, but also to information

5      and communication as well as financial activities. Finally, it also concerns non-market activities such as public administration, education, human health and social action. Tertiary is defined in the text and often changed to Commercial/institutional in the manuscript.

*What do you mean by "the age of the fleet" (l.29) in the building energy model? Please clarify this entire paragraph*

10      The age of the fleet refers to the age characteristics of the heating equipment. The whole paragraph was checked to be clearer.

*Title and throughout the paper: It's "Greater Paris" not the greater Paris*
That was changed in the manuscript

*How do you justify that Paris has "the best public transport network"? Further, there are references missing in this entire paragraph.*

15      This sentence referred to a field survey from the Institute for Transportation and Development Policy, it was specified and referenced.

*P.17: the last sentence is cuff-off.*
We reread the whole article. This sentence has been removed.

20      *P. 2: "individual mobility in the exposure of individuals"?*
It was unclear, changes were made in the manuscript

*P. 34: What do you mean by "reference methodologies"?*
We used COPER methods for road traffic. These are methods whose technical development is ensured by the European Environment Agency (EEA), as part of the activities of the "European Topic Center on

25      Air and Climate Change" and since 2007, under the aegis of the Joint Research Center of the European Commission, which is the scientific coordinator of the COPERT program. In this sense, they can be considered as reference methodologies. This was better explained in the manuscript.

*References are in many cases not put correctly (inside brackets rather than in text)*

30      OK, see changes in the manuscript

*P. 16: does the model over- or underestimate?*
The model underestimates. This discussion is now broader.

35      *(l.56) "numerous economic studies have shown[..]" please provide example references*
We have added references.

*Finally, the manuscript needs to be thoroughly English proof-read (2)*
The text has been carefully re-read to simplify the sentences and make them clearer and better understood.

40      ## Author's response to Anonymous Referee #2

Some questions are identical between AR#1 and AR#2 so some answers are reported twice.

**Q1:** *the authors should more clearly identify what distinguishes OLYMPUS from other similar models.*

45      Compared to traffic models, OLYMPUS is not original because it is based on a classic approach and on known / robust equations (although it is a strength to rely on a formulation already tested).

However, there are 2 specific contributions of OLYMPUS.

5 ✓ The first one is to couple several disciplines with the final objective of air quality modeling. Indeed, OLYMPUS is dedicated to produce elements necessary for the modeling of AQ scenarios, and not only to the urban traffic management. This is why - as mentioned above - it is included in an urban modeling platform. It is forced by a model of economy, land use and urban growth that provides urban scenarios (mainly described by land use, transportation

10 network and population density). These scenario may consider densification or urban spread policies, as well as improved public transportation system, or city green belts… OLYMPUS includes the outputs such as land use and transport supply to calculate the transport demand and then the city shape-dependent emissions, which will be provided to the CHIMERE AQM. This provides a **new generation of urban scenarios** for air quality issues.

15 ✓ The second one is to provide a **new form of environmental decision support**, through an approach based on individual decision. Indeed, to estimate the modal share, OLYMPUS mostly relies on the decision of each agent of its synthetic population (utility approach). OLYMPUS is thus able to take into account the possible "changes of practices" in mobility. The same may be considered for energy consumption, regarding the choice of heating mode. In the field of

20 air quality research, this is totally new. This does not prevent that OLYMPUS also support classic emission scenarios. OLYMPUS is indeed able to consider a fleet of vehicles and boilers corresponding to a particular future scenario: it can therefore take into account the emission policies based on improved technologies.

The data produced by OLYMPUS allows much more advanced scenarios than usual air quality
25 models usually do (they only consider a regulatory percentage of emission reduction). OLYMPUS simulates the scenario elements needed to provide decision support that takes into account the collective appropriation of environmental policies as well as the impact of urban forms on transport demand and mobility, which is absolutely not the case in the literature today. To our knowledge, OLYMPUS is the most comprehensive tool for conducting AQ / energy consumption
30 scenarios inside an urban modeling platform.

*Q2: My major reason of concern is as follows. In section 8, page 17 lines 53-55 is stated "The OLYMPUS modeling platform has been developed to meet the need for the development of a tool that links the urban diagnostics provided by the different disciplinary models, in order to produce analyses of the effects of urban policies on*
35 *pollutant emissions, air quality and population exposure." The issues of exposure, air quality and urban policies are mentioned often in the literature review, yet OLYMPUS does not address these issues. In fact as stated in the beginning of section 2 "The objective of this model is to estimate the pollutant emissions linked with energy-consuming urban activities", and that is factual and correct. Exposure estimation requires combining pollutants concentrations (air quality) with time.*

40 Yes, this was not clear.

Indeed, the OLYMPUS model has been developed to fit into a modeling platform consisting of a set of disciplinary models. It was designed to produce a new generation of emission scenarios for Air Quality Models (AQMs), but also to simulate a set of new parameters that will be needed for the calculation of the exposure in the next steps of the modeling platform. These last parameters are (i) a synthetic
45 population characterized in particular by place of residence and age and (ii) loops of individual mobility. After the AQM simulation, individual urban travels provided by OLYMPUS are intended to be confronted with the 4D (x,y,z,t) concentration fields of our air quality model CHIMERE* in order to link spatial mobility and concentrations at the hourly time step, and to create a space-time exposure budget. This is a totally innovative element in the field of air quality research. Without those data, we
50 would not be able to address innovative environmental and management policy issues in the platform,

5     or to treat exposure in a finer way than what has been done so far in the literature (i.e. map crossing concentrations with population density).

However, it is worth noting that OLYMPUS still produces diagnoses on energy consumption and builds pollutant emission inventories, characterizes congestion, simulates modal choice for all individuals in the city and calculates passenger car mobility in different hypothetical situations. Which in itself is an

10     environmental diagnosis in the broad sense. The abovementioned situations may include urban planning, urban fabric, transport policies or any socio-economic context, which would be simulated by a LUTI model upstream of OLYMPUS. Those situations would be transferred to OLYMPUS through land use, population density and transport network data.

OLYMPUS is therefore a tool that produces many new generation data needed to address innovating

15     issues of air quality and to improve exposure. In the previous version of the paper we merged the final objectives of OLYMPUS and the actual OLYMPUS tasks. We corrected this point.

*Q3: OLYMPUS can be used "to produce analyses of the effects of urban policies on pollutant emissions", the work presented does not allow verifying the ability of the model to do that, in fact it doesn't even address urban policies.*

20     *How will the model "react" to a change in urban policies? How will the authors change the inputs to reflect different urban policies?*

OLYMPUS can integrate all the regulations concerning the technological improvements of combustions: penetration of the best technologies for the stoves and evolution of the road fleet. They may also include energy requirements for new buildings or for the renovation of certain

25     neighborhoods.

But public policies taken into account in our modeling platform can also include recommendations on the urban planning scheme (these are taken into account in the economy and land use model and transferred to OLYMPUS via land use and housing density parameters). The also may plan the development of the public transport offer (which will be directly taken into account in OLYMPUS via

30     transport times for the modal share simulation).

Furthermore, OLYMPUS can produce, for the CHIMERE air quality model, emission scenarios that estimate the changes in car and energy use (whether because of an incentive policy or an evolution of urban form) and their effects on energy consumption (climate diagnosis) and on air quality via concentration calculations in CHIMERE (environmental diagnosis). For example, for the transport

35     sector, the model can simulate the impact of transport cost, or improved transport supply on modal choice and on associated emissions. For the residential sector, the model can simulate the impact of thermal renovation on energy consumption. The model can also be set on car fleet composition, stoves composition, housing type composition…

OLYMPUS can therefore help to understand the link between urban form, transport supply and

40     mobility and consumption practices. The scenarios conducted with OLYMPUS make it possible to predict the impact of sprawl or urban densification on road emissions, but also the extent of the behavioral changes required to change the state of play and improve air quality.

In this article we highlighted the ability of the model to reproduce a real situation to evaluate the

45     robustness of the model. But the OLYMPUS model can be used on different urban development scenarios. For example, it is possible to modulate the following parameters in the model:

✓ Transport infrastructure (add / remove transit stations, road links)
✓ Tolls

5     ✓  Parking
      ✓  Income
      ✓  Time cost
      ✓  Carpooling
      ✓  Change fleet composition (diesel, gasoline, electric cars)
10    ✓  Energy efficiency of buildings
      ✓  Stoves fleet
      ✓  Boiler fleet

*All these clarifications that we make here have been introduced in the text to better clarify the role*
15  *of OLYMPUS, its contribution, its specificities and the limits of its fields of action.*

*Q4: how will the authors address urban configuration?  what exactly is urban configuration?*
The city **morphology** (or the urban area morphology) is the pattern of its occupation of space as measured by the density and the degree of hierarchization of the different clusters that compose it. We may also call it the urban form.

20  Indeed, we consider the city with a systemic approach (this is why we use urban system for larger urban area including several cities such as the Paris area). Our definition of **urban organization** is the way in which individual activities are organized in the built space. The urban organization is the way in which the urban form, the distribution of employment areas and the transport supply impose spatial interactions between individuals. It is one global parameter, even if it is the result of individual actions
25  conducted without care of the collective. We are interested almost exclusively here in the "mobility" aspect of the spatial interactions between city and individuals. In the model, the parameters designing urban configuration are population distribution, transport infrastructures, buildings, job centers in the city. It may be observed through the UDI and UTAI index mapping.

Urban organization is the result of population interacting with the **city configuration**, that refers both
30  to the spatial configuration of the building area (the so-called **built environment**) and to the spatial configuration of the services and networks.

All definitions are based on those of Bretagnolle et al. (2010). Here is a quote about urban organization:
*"The organization of urban systems can be described on three main levels [including] the meso-level [that] corresponds to the city itself (when really defined as a consistent geographical entity). (…). This organization is shaped by interactions operating*
35  *on different spatial and temporal scales of observation. We can visualize a city as the envelope for the daily activities of its inhabitants and the buildings hosting them. (…). In order to correctly identify a city as a consistent geographical entity, and considering that its spatial expansion as well as its in situ development are part of its growth, we define it as a "daily urban system" (this concept (…) allows for frequent social interactions which usually take place within one day). While the maximum spatial development of cities has continuously been constrained by this typical one-hour time-budget, the speed of means of*
40  *transportation available has enabled a multiplication by a factor ten of the average commuting distance between places of work or urban services and places of residence, in the last two centuries. (…) A behavioral parameter defined for spatial interactions at the individual level is reflected in the organization of the urban entity at a higher level".*

*Anne Bretagnolle, Denise Pumain, Céline Vacchiani-Marcuzzo. The organisation of urban systems. D. Lane, D. Pumain, S. Van der Leeuw, G. West. Complexity perspective in innovation and social*
45  *change, Springer, pp.197-220, 2009.*

*This has been clarified in the text, and the use of these terms has been made more rigorous.*

*Q5: These are all questions that remain unanswered, since the model was not applied to alternative urban policies. The authors should be very clear in the manuscript about the scope of the tool and the case-study presented. Any further considerations must be either accompanied by further modelling applications or removed.*

Yes, the objective is not to discuss at length simulations of public policies that are not presented here. The goal is to describe the structure and operation of a model. However, the scenario simulations

considered for OLYMPUS will be mentioned in the conclusion in order to clarify the scope of the tool, which was designed to produce emission scenarios of a new type in air pollution research. This is less troublesome now because it is accompanied by a better explanation of the tasks performed by the model and the use of its output data for air quality modeling and exposure issues.

*Q6: It needs to be revised for its English since the language is sometimes incorrect and not always clear*

The text has been carefully re-read to simplify the sentences and make them clearer and better understood.

**OLYMPUS v1.0: Development of an integrated air pollutant and GHG urban emissions model - Methodology and calibration over Greater Paris**

Arthur Elessa Etuman[1], Isabelle Coll[1]

[1]Laboratoire Interuniversitaire des Systèmes Atmosphériques (LISA), UMR CNRS 7583, Université Paris Est Créteil et Université Paris Diderot, Institut Pierre Simon Laplace (IPSL), Créteil, France

*Correspondence to*: Arthur Elessa Etuman (Arthur.elessa-etuman@lisa.u-pec.fr)

**Abstract.** Air pollutants and greenhouse gases have many effects on health, the economy, urban climate and atmospheric environment. At the city level, the transport and heating sectors contribute significantly to air pollution. In order to quantify the impact of urban policies on anthropogenic air pollutants, the main processes leading to emissions need to be understood: they principally include mobility for work and leisure as well as household behavior, themselves impacted by a variety of social parameters.

In this context, the Olympus modeling platform has been designed for environmental decision support. It generates a synthetic population of individuals and defines the mobility of each individual in the city through an activity-based approach of travel demand. The model then spatializes road traffic by taking into account congestion on the road network. It also includes a module that estimates the energy demand of the territory by calculating the unit energy consumption of households and the commercial/institutional sector. Finally, the emissions associated with all the modeled activities are calculated using the COPERT emission factors for the traffic, and the European Environmental Agency (EEA) methodology for heating-related combustions. The comparison of emissions with AIRPARIF's regional inventory shows discrepancies that are consistent with differences in assumptions and input data, mainly in the sense of underestimation. The methodological choices, as well as the potential ways of improvement, including the refinement of traffic congestion modeling and of the transport of goods, are discussed.

**1 Introduction**

As the world's population grows, the share of the population living in urban areas also increases (United Nations, 2014). These areas can be described as hubs of activity with a substantial density of individuals, buildings, transport networks and employment centers. All the human activities associated with these metropolises induce a large local consumption of fossil energy and natural resources, favoring the concentration of a great variety of nuisances (noise, stress, pollution). Among the most emitting activities induced by the city, one can find - according to the IPCC nomenclature (IPCC, 1996) - energy consumption, industrial processes, use of solvents and agriculture. However, at the city level, anthropogenic emissions are mainly the result of the combustion of road transportation fuels, as well as residential, commercial and institutional heating and boiling, which account for more than half of total urban emissions (International Energy Agency, 2016). In Europe in particular, some cities are associated with massive use of passenger cars (and sometimes even diesel fuel) which further increases their potential for the emission of air pollutants. In these areas, road transport and the production of electricity and heat represent more than 60% of the anthropogenic emissions of nitrogen oxides (NOx), particles smaller than 2.5μm ($PM_{2.5}$) and Non-Methane Volatile Organic Compounds (NMVOCs) (International Energy Agency, 2016). Quantitatively, although sulfur oxide (SOx) emissions have declined since the 1990s, NOx and particulate matter

(PM) emissions continue to increase in Asia and show no clear downward trend in Europe (Amann et al., 2013; Klimont, 2017; Miyazaki et al., 2016). As a result, even though exposure to short-lived peaks is decreasing, the exposure of the population to chronic pollution is still high in European urban areas (EEA, 2015), and 94% of exceedances of the short-term limit value for $PM_{10}$ have been observed in urban or suburban areas (EEA, 2016). Air pollution has serious consequences for human health. Recent estimates confirm the considerable burden of diseases associated with air pollution in urban areas, which results in pulmonary and cardiovascular diseases, cancer, but also certain types of diabetes in adults or by an attack on the neuronal development of very young populations (World Health Organization, 2013). From an economic perspective, it leads to high health care costs and to a significant drop in productivity for businesses. At the same time, the societal question related to the degradation of air quality arises. According to a survey carried out between 2007 and 2015 on behalf of the European Commission (European Commission, 2010), there are 9 European Union capitals among the 20 cities with the lowest rate of people satisfied with the quality of the urban air, with the greatest decrease in the satisfaction index being observed in Greater Paris.

Given the systemic nature of urban areas, it became clear that we could no longer ignore the links between urban morphology, individuals, energy consumption and pollutant emissions when dealing with environmental urban issues (Le Néchet, 2010) - urban morphology (or urban form) being defined here as the patterns of space occupation by a metropolis, measured by the density and degree of hierarchization of the different urban cores. Indeed, the IPCC has recently recognized the impact of 4 variables linked with urban morphology (density, mixed land use, connectivity and accessibility) on energy consumption, climate and air quality issues. The effects of these 4 variables are expressed through the elasticity of the number of kilometers traveled, a parameter called "Vehicle km traveled - VKT" (Seto K. C. et al., 2014). First, urban density - which reflects the spatial distribution of population, employment, housing or transport structures - impacts mobility choices through the deployment and sustainability of the local transport supply. Mixed land use (estimated by local employment to household ratios or households to services ratios, for example) also determines the morphology of the city, since the reduction in land use diversity reinforces the centrality of activities, and shapes the population mobility for all trip purposes. Connectivity corresponds to the spatial structure and density of roads and pedestrian ways: in particular, it has been shown to promote walking. Finally, accessibility - defined as access to jobs, housing and services - can help reduce VKT, particularly for professional mobility (commuting). The way in which the urban form, the distribution of employment areas and the transport supply impose spatial interactions between individuals can be identified as the urban organization (Bretagnolle et al., 2010). Such an organization appears clearly dependent on the cost of energy. And when the relations between urban form and daily mobility are questioned, they invariably lead to classic issues in the literature (Melia et al., 2011; Le Néchet, 2010; Schindler and Caruso, 2014): spread urban forms would be the most energy-consuming structures, while a strong hierarchy between urban centers with an increase in central compactness would help reduce the distance to jobs and the use of the car. Moreover, dense urban forms, unlike spread urban forms, allow a more efficient use of energy through the use of dense networks (heating, electricity, gas). However, it seems that dense urban structures also tend to reduce the share of local trips undertaken by sustainable modes, due to increased metropolitan integration.

Models that aim to predict air quality in a given geographic area (called Chemistry Transport Models – CTMs or Air Quality Models – AQMs) require a set of input data that includes an anthropogenic emission inventory. This

type of input characterizes the intensity, the composition and the spatial and temporal distribution of pollutant releases by human activities. Emission inventories for a given situation can be obtained either through a top-down (using national aggregated information and indicators to spatialize the emissions) or a bottom-up (collecting local information from specific activities - e.g. road traffic count data - to generate a high-resolution inventory) process. Conventionally, regulatory abatement coefficients are applied to current emissions to produce prospective emission inventories, to account for both technological developments and the effects of a constant re-evaluation of emission standards. The emission scenarios approach traditionally uses these modified inventories to simulate air quality over a given time horizon. However, considering the above mentioned findings, prospective emissions calculations need to be rethought to take into account all the parameters affecting the urban organization and produce a more comprehensive calculation of energy consumption in the urban area. In particular, the models providing prospective emission scenarios to AQMs should be able to predict the effects of urban planning and individual practices on mobility and energy demand. Only by integrating emission scenarios of this nature into air quality models can the levers of urban air quality and sustainability be identified. Finally, it is also important to go beyond the quantification of future pollutant emissions and the mapping of air quality obtained through AQMs, and consider exposure to air pollution, which makes it possible to address the issues of environmental inequalities and health risks. Indeed, the relationship between the individual and urban space is known to be at the origin of a highly differentiated exposure, discriminating places of residence, lifestyles and social categories. But our understanding of this issue remains uncomplete, and additional research that integrates the theory and practice from both air pollution and social epidemiology is expected (O'Neill et al., 2003). In particular, it is essential to change the traditional calculations of exposure to integrate mobility within the urban space, and take into account the evolution of the exposure of individuals during the day (Steinle et al., 2013).

There are still few research projects in the literature that have included a large number of urban components into emission scenarios dedicated to AQMs (Manins, 1995; Marquez and Smith, 1999; Martins, 2012; De Ridder et al., 2008). Prospective modeling research in the 2000s has revealed the determining role of mobility and city configuration (considered as the spatial organization of buildings, services and networks) in the exposure of individuals, but the study focused on academic situations (Borrego et al., 2006). However, over the last decade, social components have progressively been integrated into urban emissions models such as TASHA-MATSIM-MOBILE6.2C (Hao et al., 2010) or TRANUS-TREM (Bandeira, Coelho, S, Tavares, & Borrego, 2011), which are now able to quantify the impact of urban policies on road traffic emissions through car-pooling, transportation fleet technology and individual modal choice. The strength of these models is linked to the implementation of a microsimulation approach based on individual choice, which depends on economic parameters. However, most of the applications focused on road traffic emissions only Hatzopoulou et al. (2008); Hülsmann et al. (2014) whereas in the current context which places particular emphasis on the emerging concept of sustainable cities, it is necessary to take into account all air pollutant emissions related to energy consumption, insofar as they interact with air quality and climate change. In particular, there is a need to also take into account small combustion emissions (both residential and commercial) and their related policies to go further in the realism of the urban scenarios, and to address the issue of air quality levers in a more holistic manner.

OLYMPUS is an emission model designed to produce a new generation of emission scenarios for Air Quality Models (AQMs) at the scale of an urban area. It aims to meet the need described above to produce emission

scenarios that integrate the interactions between the geographical aspects of the city, its population, the organization of buildings and urban networks, in order to produce a more comprehensive environmental decision support. It has been developed to integrate into a platform of disciplinary urban models connected in series. The platform provides data on urban morphology, localization of activity centers and organization of transport networks corresponding to an urban planning scenario, or more broadly to public policies. OLYMPUS uses this data to produce a transport and energy demand diagnosis in the study area, which takes into account the main parameters influencing the urban organization (urban morphology, population density, services and networks), based on the simulation of individual behaviors related to mobility and energy consumption. Finally, these diagnoses are used to produce a pollutant and greenhouse gas emission inventory corresponding to the simulated scenario, and resulting from a systemic representation of urban areas that highlight the role of urban configuration, urban planning, individual choices and political forcing in the sustainability of cities. The use of this new generation inventory in the CHIMERE air quality model (Menut et al., 2013), located at the end of the chain in our modeling platform, will make it possible to predict the air quality associated with the emission scenario produced by OLYMPUS, and to provide a new form of decision support on the relationship between urban forms, population and air quality. In addition, OLYMPUS simulates the individual mobility data that will be needed to improve the calculation of population exposure to pollutants in the final stages of the modeling platform. After the air quality simulation, individual urban travels provided by OLYMPUS will be processed with the AQM simulated concentration fields in order to create a space-time exposure budget for all individuals. Although this is beyond the scope of this article, improvement of exposure in our modelling platform is one of the great innovations brought by the development of the OLYMPUS model.

In this paper, the operation and main features of the OLYMPUS model are described. The different modules will be presented individually. An application on Greater Paris will be presented in the last section. The model results for this case study will be presented, evaluated and discussed.

**2 OLYMPUS model overview**

The main objective of the OLYMPUS model is to estimate the pollutant emissions resulting from energy-consuming activities at the scale of an urban system, considered as the area that groups the daily activities of its inhabitants and the buildings hosting them (Bretagnolle et al., 2010), to produce innovative emission scenarios for AQMs as part of environmental decision support. The first specific contribution of OLYMPUS is to process data from multiple disciplines, but dedicated to serving air quality modeling. The second specificity of OLYMPUS is to provide new forms of decision-making support for the environment, thanks to an emissions calculation approach that integrates individual behaviors. Indeed, OLYMPUS relies mainly on the decision of each agent of a synthetic population to estimate the modal share. It is thus able to take into account the possible "changes of practices" related to mobility. The same can be considered for domestic heating practices. Finally, the emission data produced by OLYMPUS allow us to build much more advanced emission scenarios than the air quality models have simulated so far (taking into account only a regulatory factor for emission abatement). From this point of view, OLYMPUS is a fairly comprehensive and innovative tool.

OLYMPUS is integrated into an urban modeling platform connecting in series several urban models. It has been designed to collect city-specific input data such as morphology, population distribution and employment centers, road transport networks and public transport as well as climate variables that affect emissions. Climate data are

provided by a meteorological model. In the current situation, land use, population and urban services can be obtained from surveys. When simulating a public policy scenario, these data can be provided by the outputs of the NEDUM-2D model (http://www.rgte.centre-cired.fr/Rubrique-de-services/Archive-Equipe/Vincent-Viguie/article/NEDUM-2D-model) included in the platform and simulating an urban organization corresponding to an economic, environmental and urban planning scenario on a horizon given time. The first step in OLYMPUS is to simulate a synthetic population, its properties (age, type of household ...) and its spatial distribution, in order to describe, count and then spatialize individual activities within the area. Then, OLYMPUS uses activity-based emission factors to produce a spatially-based emission inventory for Non-Methane Volatile Organic Compounds (NMVOCs), nitrogen oxides ($NO_x$), carbon oxides (CO and $CO_2$), $SO_2$ and primary particles. The general structure of OLYMPUS is detailed below. At each stage of operation, the calculation methods will be precisely described.

**2.1 Main characteristics**

In its current version, OLYMPUS models the main pollutant emissions linked with energy consumption, namely road transport and combustion processes from domestic activities and building heating in the tertiary sector. This last sector is composed of activities related to trade and services, but also to information and communication as well as finance. It also includes public administration, education, human health and social action. It will also be referred to as the commercial/institutional sector. As shown in Figure 1, OLYMPUS is composed of 6 calculation modules, supporting 4 main tasks.

The first task of the model is to create a synthetic population to which a set of properties will be assigned. This synthetic population is designed to be representative of the population living in the territory concerned and is characterized by the age, gender and main activity of the agent as well as his belonging to a household. The creation of this **synthetic population** is based on the reconstitution of surveys in the GAIA module (element (a) of the flowchart in Figure 1).

The second task of OLYMPUS is to provide a transportation database, built taking into account the lifestyles of individuals. This database is obtained from successive diagnoses on the generation of individual trips - zonal attractiveness, spatial and temporal distribution of activities, transport supply and choice of routes. In the OLYMPUS modeling process, this task is based on 3 modules.
- A first module - THEMIS, (b) - defines the **accessibility** and **attractiveness** of the different administrative units of the city, as well as the average time travels between them.
- An **Activity-Based Travel Demand** (ABTD) module called MOIRAI (c) computes the daily activity patterns of all agents. It also describes their daily mobility in time and space.
- An **assignment** module called HERMES (d) provides spatialized daily trips by computing the shortest path between the origin and the destination of a trip (OD matrices).

In parallel, OLYMPUS is leading the third task of calculating the **energy demand of buildings**. To this end, the HESTIA (e) module calculates the average household energy consumption per square meter, the size of dwellings, and the city energy mix in order to produce a spatialized energy demand for buildings, including a specific climatic correction on the simulated period.

In the fourth task, OLYMPUS generates **emissions** from both road transport and small combustion heating systems. These emissions are calculated using reference methodologies such as COPERT IV for road traffic, whose development ensured by the European Environment Agency (EEA) in the frame of institutional and research activities about air quality and climate. For buildings, emission calculation methodologies are also taken from the EEA guidebook (European Environment Agency, 2013). The computation of pollutant emissions is carried out by the VULCAN module (f).

All running OLYMPUS scripts are shell-based, Python 2.7-programmed, and C-compiled for faster execution speed. For network graphs and spatialized data analysis, NetworkX (Hagberg et al., 2008), as well as GeoPandas and NetCDF libraries were included in the Python interpreter. Due to the large number of computation loops, the model applies data parallelism that consists of partitioning the data with a Multithreading approach.

**3 The synthetic population generator (GAIA)**

The GAIA Synthetic Population Generator is the first OLYMPUS module to be run. It allows the generation of a synthetic population representative of a given urban area. The synthetic population generator uses mainly urban census data to assign each agent in this population an age, gender and main activity, as well as socio-economic parameters such as possession of a driver's license. The module distributes this synthetic population over the modeled territory, through an urban zoning based on household densities in the urban area - an exogenous variable provided to the model. In the end, we obtain a synthetic population based on census data or demographic scenarios, with an individual description of its agents which is the specific contribution of GAIA.

There are several statistical techniques for estimating the characteristics of a population in a restricted area, as reported in Rahman (2017). The most common method is the Iterative Proportional Fitting (IPF) procedure (Deming & Stephan, 1940, Baggerly, & McKay, 1996; Müller K., & Axhausen, K.W. 2010), which generates an adjusted matrix of the survey data used to constrain the global synthetic population patterns, based on the minimization of chi2, a method for estimating unobserved quantities from marginal numbers. The algorithm must be fed with the total population data and with subtotals for each of the property types, using both aggregated and disaggregated data. Conditional probabilities are also part of the methodologies used to create a synthetic population. This approach is based on Bayesian statistics and relies on a representative sample of population, in which the discrete conditional probabilities governing every characteristic (e.g. age) are identified. Then, a unique value of this characteristic is assigned to each agent of the population using a random distribution that follows the identified probability law. This approach makes it possible to create (on the basis of a representative sample) a database that distinguishes each individual (disaggregated data) as well as each household and dwelling by assigning their own characteristics. These two methods differ in how to generate a built-in population but the results are recognized as quite comparable, although one of the main interests of the IPF procedure is its ability to generate greater variability than conditional probabilities in the population. Despite this, the use of the IPFP in a region such as the Ile-de-France would require considerable work to structure the input data. For this reason, we decided to use the conditional probability approach, which has been widely used in the field of transport demand modeling (Antoni et al., 2010; Banos et al., 2010; Mathis et al., 2008). In order to mitigate a possible lack of variability, we implemented a spatial component in the distribution of the socio-economic characteristics of agents.

The implementation of GAIA takes p                                    gure 2.

- The determination of the u                                    index (**UDI**) and divided into 3
  classes: the urban pole (CE                                    suburbs (SUBURBAN) (Figure
  2.b)
- For each household in the                                    etic population by defining the
  household size and the prop                                    lities (Figure 2)

**3.1 Urban structure and population**

The prerequisites for population gen                                    the classification of urban areas
on the basis of an urban density in                                    urban area on a scale of 1 to 3
(SUBURBAN – URBAN – CENTE
The UDI index is defined in Eq. (1).                                    lataset into 3 large sets of urban
density by applying a linear division                                    reas to the very dense areas. It is
based solely on real population dens                                    g to the logarithm of population
density.

$$
UDI(z) = \begin{cases}
CENTER, & if\ \log\left(1 + \frac{n_{hh}}{A}\right) > \propto_2 \\
URBAN, & if\ \propto_1 < \log\left(1 + \frac{n_{hh}}{A}\right) < \propto_2 \quad (1) \\
SUBURB, & if\ \log\left(1 + \frac{n_{hh}}{A}\right) < \propto_1
\end{cases}
$$

Where $\propto_1$ and $\propto_2$ are key classification values depending on the logarithm of households density, $n_{hh}$ the number of households, z a specific area of the domain and A is the surface area.

Figure 2.b shows a schematic representation of the UDI as well as the population-specific attributes for each UDI value. Such discrimination of the properties is necessary for the realism of our output data because population density affects the urban landscape (buildings and houses), the location of activities and the structure of households. It is assumed here that the distribution of household types is different between urban centers, surrounding urban areas and suburbs (Pisman et al., 2011; Thomas et al., 2015), and that the distribution of buildings and single-family houses will vary between these different areas.   Finally, we have added a variation in household structure with distance from the urban center, according to Hulchanski, 2010. These hypotheses offer greater variability in the spatial distribution of agents than a simple conditional probability distribution.

**3.2 Generation of a synthetic population**

The generation of the population depends on Probability Mass Functions (PMFs) that rely on census data, as shown in Figure 2.d which represent the age-specific PMF of an agent living alone. In each zone and for each household, GAIA uses a discrete probability distribution to:

(a) define the number of agents in the household
(b) characterize the type of family
(c) define gender, age and main activity of agents

Eq. (2) predicts the number of agents in the household, according to the type of zone (CENTER, URBAN, SUBURBAN), which is recalled that they differ in the density of population, and in the distribution of the types of housing. The probability of having n agents in the household is based on conditional probabilities and defined by a truncated Poisson distribution:

$$P_{hhs}(n|\lambda, UDI) = e^{-\lambda}\frac{\lambda^n}{n!} \quad (2)$$

where $\lambda$ is the average household size, $n$ is the number of agents in the household and $n \in A$, with A = [1,7] and $A \in \mathbb{N}$. Figure 2.c is an example of a household size probability distribution based on a truncated Poisson's law.

Eq. (3) is used to define the type of household among the 4 family classes which are Single (male/female); Couple - No children; Couple with children; Single parent family (male/female). The selection of the family type is also based on conditional probabilities ($P_{FAM}$) and follows

$$P_{FAM}(n) = \begin{cases} \text{"Single"}, & if\ n = 1 \\ P_{1F}(n), & if\ n = 2\ and\ \in\ R_{1F} \\ P_{2F}(n), & if\ n \geq 3\ and\ \in\ R_{2F} \end{cases} \quad (3)$$

where $R_{nF}$ is the family type defined for the n persons in the household

$$R_{1F} = \{\text{"Couple"}, single\ parent\ family\text{"}\}$$

$$R_{2F} = \{\text{"Family"}, single\ parent\ family\text{"}\}$$

and $P_{1F}(n), P_{2F}(n)$ correspond to weighted functions based on survey data.

Eq. (4) allows to estimate the attributes of the agents (age, gender, main activity). The gender of each agent is defined by a conditional probability (while the gender of its partner is opposite), such as:

$$P_{gdr}(\eta) = \begin{cases} P_{sex}(), & if\ householder\ and \in\ R_{gdr} \\ P_{sex}(), & if\ Child\ and \in\ R_{gdr} \\ P_{sex2}(), & if\ Partner\ and \in\ R_{gdr} \end{cases} \quad (4)$$

where $\eta$ represents the situation of the agent in the household, $R_{GDR}$ is the sample space and consists of 2 elements {"male", "female"}, $P_{sex1}()$ corresponds to a weighted function based on the census data and $P_{sex2}()$ is conditioned by the gender of the householder.

The age of an agent depends on the type of household - still based on conditional probabilities – and is linked to specific sample spaces: for householders, for couples (age difference less than 20 years), and for children. There are 20 age classes with a 5-year division.

$$P_{AGE}(\eta) = \begin{cases} P_{A\_1}(n), & if\ householder\ and \in R_{householder} \\ P_{A\_2}(n), & if\ Child\ and \in R_{child} \\ P_{A\_3}(n), & if\ Partner\ and \in R_{householder} \end{cases} \quad (5)$$

where $P_{A\_1}(n), P_{A\_2}(n)$ and $P_{A\_3}(n)$ are probability mass functions based on census data. $R_{householder}$, $R_{child}$ and $R_{partner}$ are age types defined for n people in the household with

$$R_{head} \quad = [20;70] \text{ and } R_{head} \in \mathbb{N}$$
$$R_{child} \quad = [0;20] \text{ and } R_{child} \in \mathbb{N}$$
$$R_{partner} = [20;70] \text{ and } R_{partner} \in \mathbb{N}$$

The principal activity of the agent depends on its age and on the unemployment rate. Agents under 18 are educated and agents over 65 are retired. Other agents may be employed, unemployed or studying.

$$P_{ACT}(age) = \begin{cases} P_{ACT1}(age), if \ 20 < age < 30 \ and \ \in R_{act1} \\ P_{ACT2}(age), if \ 30 < age < 65 \ and \ \in R_{act2} \\ \text{"School"}, if \ f \ age < 18 \\ \text{"Retired"}, if \ age > 65 \end{cases} \quad (6)$$

where $P_{ACT1}(age)$ and $P_{ACT2}(age)$ represent the mass probability functions giving the probability of having as main activity one of the activities defined in $R_{act1}$ or $R_{act2}$

$$R_{act1} = \{\text{"Active"}, "School"\}$$
$$R_{act2} = \{\text{"Active"}, "Unemployed"\}$$

**4 Road transportation generation**

To simulate the transportation demand according to the activities of the population, OLYMPUS requires a large number of external data. The spatial distribution of employment centers is a first key parameter: the calculation is based on a spatialized file containing the number of jobs per zone, usually in a Geographic Information System (GIS) format. To simulate ABTD, OLYMPUS uses socio-economic data at a disaggregated level for each area, which this time can be provided by the GAIA synthetic population generator.

Transport networks are a second external parameter needed to calculate mobility. The road network includes urban and non-urban highways as well as major traffic lanes, and information on no-load speeds. The public transport network includes all related stations, also in GIS format. All of this data will be analyzed at the finest accessible spatial scale, called the Transport Analysis Zone (**TAZ**), which can be a district, a sub-district, a municipality, or any other division of the city. The resolution of the data foreshadows the refinement of the mobility of the population.

The modeling of urban road transport is organized in 3 stages:
(1) Determination of the attractiveness and accessibility of the different zones that make up the domain.
(2) Restitution of the agent travels on the basis of the realization of the programmed activities.
(3) Assignment of motorized trips on the road network.

**4.1 TAZ accessibility and attractiveness (THEMIS)**

The operating diagram of THEMIS is presented in Figure 3.a. The main steps are:
- Definition of accessibility
- Computation of attractiveness

One of the key forcing data of the ABTD Model (ABTDM) is the accessibility of the TAZ, which provides the basis for mobility choices. Accessibility is calculated from all the activities considered useful for the agents within a given radius, its value also taking into account the public transport service in this area. For this purpose, THEMIS analyzes the population density, road network and public transport network of the TAZ. The result is an index with five levels of accessibility, accounting for public and individual transport to the area, and called the Urban Transport Accessibility Index (**UTAI**).

This flag is used to set the access mode shares. As shown in Figure 3.b, an area with a UTAI$_{MIN}$ index will only be accessible by walking (WALK) while an area characterized by a UTAI$_{MAX}$ index will be well serviced with a wide choice of transport infrastructure. The definition of the 5 UTAI classes depends on the value of μ, defined by the following equation:

$$\boldsymbol{UTAI}(zone) = \begin{cases} 1, if \ \mu < \propto_1 \\ 2, if \ \propto_1 < \mu < \propto_2 \\ 3, if \ \propto_1 < \mu \ < \propto_3 \ (7) \\ 4, if \ \propto_1 < \mu \ < \propto_4 \\ 5, if \ \mu \ > \propto_4 \end{cases}$$

with

$$\mu = \log\left(1 + \frac{n_{hh} \times n_{st}}{A}\right)$$

where $\propto_1, \propto_2, \propto_3$ and $\propto_4$ are key classification values depending on the logarithm of household density and public transport density. $n_{hh}$ is the number of households per TAZ, $n_{st}$ is the number of public transportation stations in the TAZ and $A$ is the area of the TAZ. The UTAI index (see correspondence in Figure 3.b) thus helps to design the use of the city from its transport infrastructure and to define a realistic travel time for public transport, as represented on the isochronous curve of public transport in Paris (Figure 3.c).

The attractiveness of activities is an important parameter that shapes the agenda of agents. It can be defined as the ability of a zone to get an agent to carry out a given activity on the site, respecting the average length of the trip associated with this type of activity, as illustrated in Figure 3.c. The model assumes that there are 2 types of activities: WORK and OTHER. The complete list of the OTHER activities taken into account in the ABTD model are:

- HOME
- SCHOOL
- SHOPPING
- SECONDARY
- ACCOMPAGNYING
- VISIT
- LEISURE

The main parameter that differentiates our two types of activities is the average duration of the journey, which varies considerably, the average distance traveled for commuting (home-work) being longer than that of other types of activities. The distance to the TAZ is an important variable in the estimation of its attractiveness for an agent. The attraction potential of WORK depends on the number of jobs in the TAZ. For OTHER activities, their

attractiveness depends on the population density of TAZ. However, some activities like visiting a friend or going on vacation are still underestimated by the ABTD model. The computation of attractiveness is based on Huff's theory of the gravity model (Huff, 1964). It is based on the definition of an activity weight and works by analogy with Newton's law of gravity. The probability of conducting any activity at a specific location is therefore defined as follows:

$$\boldsymbol{P_\Omega(i,j)} = \frac{\boldsymbol{\Omega}(i,j,d)}{\sum_j \boldsymbol{\Omega}(i,j,d)} \tag{8}$$

where the attractiveness $\boldsymbol{\Omega}$ is defined by:

$$\boldsymbol{\Omega}(i,j,d) = \sum_i Act \times \sum_j Act \times \left( \frac{1}{\sigma\sqrt{2\pi}} e^{-\frac{1}{2}\left(\frac{|x - \overline{d}|}{\sigma}\right)} \right) \tag{9}$$

$$\text{and} \quad \sigma = \frac{\overline{d}}{2}$$

In this equation, $d_{mean}$ represents the average distance to reach the *Act* activity, while $i$ and $j$ are respectively the indexes of the origin and destination zones.

In the end, the attractiveness parameter is highly dependent on the city's structure and the travel practices of the inhabitants. Kwan (2003) found that few peoples act to minimize their commuting to work by relocating their home or workplaces. Given this, mobility surveys provided by some countries can be used to force average travel distances and make them more realistic.

**4.2 Activity-based travel demand (MOIRAI)**

The ABTD module MOIRAI simulates the mobility choices for each agent in the synthetic population during the day. One of the challenges of the module is to represent mobility in the most realistic way possible, taking into account the social constraints of each agent in space and time. Several ABTDM exist in the literature. Malayath and Verma (2013) proposed a review of existing models and their uses. Based on this review, we decided to use random utility theory to simulate the choice of individuals in MOIRAI. In this theory, a stochastic approach makes it possible to take into account the rationality of agents' decisions. That is, the decision is described as the choice of the agent to do what is most useful to him, depending on the opportunities available to him. In this process, utility is generally divided into 2 components, one describing the observed practices and the other describing the random component. In the theory of random utility, the main hypothesis on which the choice is based is that the maximization of utility influences the decisions of the agent. MOIRAI relies on the use of the MultiNomial Logit (MNL) model (McFadden, 1973) in which it is considered that the random components of utility are independent and identically distributed (IID) and follow a Gumbel distribution:

$$P_{mode,i} = \frac{e^{-Utility_{mode,i}}}{\sum_{i=0}^{n} e^{-Utility_{mode,i}}} \tag{10}$$

where $Utility_{mode,i}$ represent the utility function of a transportation mode $i$.

MOIRAI is implemented in 3 main stages common to many ABTD models (Castiglione et al., 2015):

(2.1) Generate the daily activities of the agent (number of trips & tour sequence)

(2.2) Agent Schedule Management (duration of activities)

(2.3) Identify the type of transportation used for each trip (location and modal choice)

These steps are described in the MOIRAI operating diagram in Figure 4.a.

**4.2.1 Generation of daily activities**

An important step in modeling an agent's schedule is the estimation of the number of trips the agent makes during the day, as shown in Figure 4.b. A first step in the simulation of the agent's agenda is therefore to define his priorities. The obligation to conduct an activity - such as going to work or accompanying children for care or education - defines the priority of an officer. Once these priorities are set, an agent can perform optional activities such as shopping, visiting a friend or going to the movies. There are 3 priorities in the model. **Work - School - Accompanying** (bring a child under 10 to school). The model can combine the priorities of the WORK and ACCOMPANYING activities. The number of trips per day ($p$) performed by an agent depends on his priorities ($x$) and is based on the following discrete probability distribution:

$$P_{trips}(x) = \begin{cases} P_{trips\ act1}(x), if\ priorities = 1 \\ P_{trips\ act2}(x), if\ priorities = 2 \\ P_{trips\ other}(x), if\ priorities = 0 \\ P_{trips\ retired}(x), if\ agent\ age > 65 \end{cases} \quad (11)$$

where $P_{trips\ act1}(x)$, $P_{trips\ act2}(x)$, $P_{trips\ other}(x)$, $P_{trips\ retired}(x)$ represent the probability of performing p daily trips according to the x priorities of the agent and with $x, p \in \mathbb{N}$.

The probability $P_{trips}(x)$ is derived from a specification of the number of trips per day based on age and priorities. This number of daily trips varies from country to country. It can be provided by local surveys or estimated from an aggregated survey database. We used information from household travel surveys, which indicates that the mobility of children and the elderly is lower than that of the population aged 20-60 years.

After determining the number of activities of the agent and establishing an order of priority, MOIRAI defines the sequence of trips. Sequences can start from home (Home-Based Sequence, HBS), include multiple home returns (Multiple Home-Based Sequence, MHBS) or be fully performed without returning home (Non Home-Based Sequence, NHABS). Figure 5 shows the different types of sequencing modeled in OLYMPUS. Depending on the number of trips, it is possible to create a single home circuit or several circuits, one centered on the place of residence and others on the activities. The probability of making one or more circuits depends on the number of daily trips.

After generating the agent schedule, the module locates each activity. Depending on the TAZ in which the agent is located, the model estimates where the agent will have the highest probability of carrying out the planned activities. This is done according to the attractiveness of the TAZ calculated by THEMIS and by using Huff's random probability approach for choosing the place of activity. For the location of the WORK activity, we use the $\boldsymbol{P_{\Omega}(i,j)}$ probability of attractiveness of the employment center. For OTHER activities, the $\boldsymbol{P_{\Omega}(i,j)}$ probability of attractiveness is based on population density.

The last step in the generation of daily activities is the insertion of the time of realization of the activities, which requires the attribution of a duration to each of the actions carried out by the agent. As shown in Figure 4.c. MOIRAI calculates this parameter using conditional probabilities with a time step of 1 hour. The module assigns

a random value to the start time, based on the agent's priorities. If the sum of the activities exceeds 24 hours, the simulation is restarted.

For the first activity, the start time is calculated as follows:

$$ACT_1^{st}(Act) = \begin{cases} st_{escort}(x), & if\ Act = "escort" \\ st_{school}(x), & if\ Act = "School" \\ st_{work}(x), & if\ Act = "Work" \\ st_{other}(x), & if\ Act = "Other" \end{cases} \quad (12)$$

where $st_{escort}(x)$, $st_{school}(x)$, $st_{work}(x)$ and $st_{other}(x)$ represent the start time of the first activity of the day. The start time is based on a normal distribution as shown in Figure 6.

$$ACT_{Other}^{st}(Act) = itime + Dur(Act) \quad (13)$$

where

$$itime(i) = \sum_0^{i-1} Dur_{Act}(j) + ACT_1^{st}(Act) \quad (14)$$

and *Dur* represent the duration of activities, defined as a random variable in a truncated interval over a time range. The distribution of start time and duration of activities in OLYMPUS is presented in Figure 6.

**4.2.2 Modal split**

The modal choice is clearly a critical parameter for calculating pollutant emissions from urban mobility. In OLYMPUS, travel modes include WALK (walking and cycling), PC (passenger car and two-wheeled car) and PT (public transport including metro, bus, tram and train). The objective is to define the probability of using a specific mode of transport according to their utility for a given route and agent. The modal choice is obtained from the expression of the utility function for each transportation mode ($U_{mod,i}$). The value of utility comes from a cost calculation including the generalized cost of transport (including the time budget ($t_{budget}$), the perception $a$ of mode $i$ and the monetary cost ($m_{cost}$)). In this calculation, the travel time can be penalized by different elements of the trip such as tolls, congestion, parking problems, etc. included in the variable $P_{mod}$, such as:

$$U_{mod,i} = m_{cost} + a * t_{budget} + P_{mod} \quad (15)$$

The utility function of the WALK mode mainly depends on the time cost of the trip. The WALK mode average speed, $Speed_{WALK\ mean}$, is defined to be 3.6 km/h. Thus:

$$U_{mod,WALK} = \frac{OD_{distance}}{Speed_{WALK\ mean}} + P_{WALK} \quad (16)$$

where $P_{WALK}$ represent walk penalties and the distance between the origin and the destination activity, $OD_{distance}$, is based on the Great-circle distance calculation,

$$OD_d = Arc\ cos\ (sin\ \varphi_A \times sin\ \varphi_B + cos\ \varphi_A \times cos\ \varphi_B \times d\lambda) \quad (17)$$

where A and B respectively denote the origin and destination points, $\varphi_A$, $\varphi_B$, $\lambda_A$ and $\lambda_B$ represent their latitudes and longitudes, and $d\lambda = \lambda_B - \lambda_A$.

For the individual passenger car mode (PC), the utility function is defined as follows:

$$U_{mod,PC} = \frac{OD_{distance}}{Speed_{PC\ mean}} + Cost_{CAR} + P_{PC} \qquad (18)$$

By default, the average speed of PC mode - $Speed_{PC\ mean}$ - is set to 22.6 km/h in urban areas. This value is based on Hickman et al. (1999) and represents the average driving speed in urban areas recorded during the MEET project. The $OD_{distance}$ is also based on the Great-circle distance equation (Eq. (17)).

The $Cost_{CAR}$ variable represents the average kilometric cost of the use of the private car. All penalties are coded as additional monetary costs: these include tolls, parking tickets, congestion and taxes as well as additional penalties for short-distance trips. All of them can be summed in the calculation of PC mode utility. The time cost for the agent is calculated from the shortest path. This step of OLYMPUS requires a considerable computation time because of the large number of agents.

With regard to the utility function of public transport (PT) from one TAZ to another, we use the following equation:

$$U_{mod,PT} = t_{PT} + Cost_{PT} + P_{PT} \qquad (19)$$

In this equation, the travel time with PT mode is $Time_{PT}$. It depends both on the accessibility of the destination area and the average distance between origin and destination points. The average travel time from one TAZ to another includes the duration of walking, waiting and traveling. It is calculated using a linear regression based on time zone transport survey data and is therefore based on realistic values.

$$t_{PT}(d,z) = \alpha_{i,j} \times a \times d + b + W_{Time}(UTAI) \qquad (20)$$

where $\alpha_{i,j}$ represents the average transit time between 2 zones and $a$ and $b$ are the linear regression coefficients. $W_{Time}$ is the UTAI-dependent waiting time.

The $t_{PT}$ parameter is usually calculated using General Transit Feed Specification (GTFS) data - if available for the city, and computed using either the Connection Scan Algorithm (Dibbelt et al., 2013) or the RAPTOR algorithm (Delling et al., 2012). The limitation of these methods is the huge computing time required. As a consequence, they were not chosen here. Since public transport time is an essential variable for the estimation of the general cost of public transport, we have developed a methodology based on a zonal approach and using the UTAI. This method has limitations with respect to CSA or RAPTOR algorithms. However, we consider that a realistic simulation of the UTAI matrix and an appropriate calibration of the module with real transport times can lead to satisfactory results. The $Cost_{PT}$ variable represents the daily cost of transit. Transit penalties may be well represented by the frequency of transit service.

**4.3 Assignment**

The transport demand previously generated by the ABTD module MOIRAI generates travel matrices providing information only about the origins and destinations of the flows. The next step is to project on our modeling grid the paths taken by the agents, in order to provide spatialized pollutant emissions from transport. For this purpose, we only take into account flows related to private vehicle use.

There are 3 ways to handle traffic assignment. One is the microscopic approach, which manages the traffic at the scale of each vehicle, as proposed by models such as VISSIM (Gomes and May, 2004), AIMSUN (J. Barcelo, J.L.

Ferrer, 1989) and PARAMICS (Cameron et al., 1994). A second approach is that of mesoscopic models, which are interested in the evolution of sets of vehicles, like the models CONTRAM (Taylor, 2003) and DYNASMART (Mahmassani et al., 2005). Both approaches are not very compatible with the scale of the city, on which we focus. Indeed, although the use of instant-emission models like PHEM (Rexeis et al., 2013) and MOVES (U.S. Environmental Protection Agency, 2013) can provide added value, obtaining input traffic data describing each cycle of acceleration and deceleration of the vehicle is quite difficult, and their consideration requires high computational time. These constraints make the microscopic approach somewhat precarious. We must therefore rely on a macroscopic description of the traffic, in the form of a vehicle flow and using global variables such as the average speed on each section of a traffic axis, as in DAVISUM (Broquereau L., 1999) and TransCAD (Caliper Corporation, 2010) models. As most of these transport models are not open source, we opted for the development of our own traffic assignment model within the OLYMPUS ensemble: HERMES. HERMES is a macroscopic traffic module that works with average speed values for vehicle flows, ignoring the dynamics of traffic on a road. This approach is compatible with our simulation scale. It is also compatible with the most common methods of estimating traffic-related combustion emissions, which rely on emission factors per driving cycle, each cycle being characterized by an environment (city, highway, etc..), and by a mean speed per strand.

The HERMES module consists of are main 4 stages in the assignment of agents to the road network (See Figure 7.a).

(3.1) Definition of the road graph. The road network is extracted from GIS road data and transformed into a graph that records the connections between different road sections, thus creating a set of edges and nodes (intersections) using the graph theory (Bondy and Murty, 1982). The speed limit is the main attributes of edges.

(3.2) OD shortest path. HERMES computes the shortest path for each trip by solving Dijkstra's algorithm (Dijkstra, 1959). For each trip, the module identifies the nodes of the graph closest to the georeferenced O and D points. To choose the shortest path among the algorithm outputs, HERMES uses the time spent on a link as weighting.

(3.3) Goods and inter-regional transport modelling. In a third step, the integration of the regional traffic flow – including the goods and the different patterns of inter-regional transportation – is achieved. This additional step is necessary because the MOIRAI travel request only takes into account the personal trips of agents living in the city. Inter-regional transport, heavy-duty vehicles (HDVs) and light commercial vehicles (LCVs) are therefore not taken into account. This is why we developed an approach that extrapolates the transportation of goods and interregional transportation trips based on a reference ratio "passenger car / total fleet" and using ratios between HDV and LCV in urban areas. Indeed, fleet composition surveys are available for many cities. They are often based on transport organizations like TFL in London.

(3.4) Speed on link computation. Finally, HERMES integrates network congestion in its assessment of mobility. Road congestion alters speed on the road network as shown in Figure 7.c. The approach is based on the UK Department of Transport (1997) and can be represented as follows:

$$S_{link} = \begin{cases} S_0 \,, \; if \; F_i < 50 \\ S_0 - \frac{S_0 - S_c}{F_c - F_0} \times (F_i - F_0), if \; F_0 < F_i < F_c \\ \frac{S_c}{\left[1 + (\frac{S_c}{8 \times d}) \times (\frac{F_i}{F_c} - 1)\right]} \,, if \; F_i > F_c \end{cases}, \qquad (21)$$

where the speed on the road link $S_{link}$ depends on the flow $F_i$ and length $d$. $S_0$ is the free-flow speed, $S_c$ is the congested speed, $F_c$ the road link flow capacity.

This is one of the approaches suggested by Ortuzar and Willumsen, 2011 to attempt to represent empirical congestion. One of the limitations of this methodology is the consideration of the impact of signaling. Other congestion functions such as that presented by Akçelik, 1991 make it possible to better manage delays at junctions. On the other hand, this method requires knowledge of the location of the traffic lights. For street-level studies, the Akçelik, 1991 method adds a certain realism to traffic modeling. However, it has been shown that the approach we have chosen produces a satisfactory estimate of the traffic flows and road network saturation on the main roads.

**5 Building energy demand**

Figure 8 shows the organizational chart of HESTIA, the OLYMPUS module responsible for simulating the energy demand of the building. HESTIA uses the type of dwelling, the living space of the household and its average annual energy consumption as input parameters. The main task of this module is to spatialize the energy demand in the territory.

Swan and Ugursal (2009) proposed a review of models and methodologies for simulating the energy demand of buildings. In this framework, both Top-Down and Bottom-Up approaches are based on econometric, statistical and technical aspects of energy demand. They are mainly developed to better understand the efficiency and cost of energy policies. Because of its global approach, the Top-Down method lacks flexibility to create scenarios involving a change in methodology. On the other hand, some of the input parameters taken into account in a Bottom-Up approach go beyond what is feasible at the regional level. They require detailed data by type of building (structural properties, equipment, use) as well as individual parameters such as the orientation of buildings in relation to the sun. In OLYMPUS, the combustion emissions modeling is carried out in two stages by the HESTIA module. It lists combustion activities for residential, institutional and commercial heating, as well as domestic hot water and cooking. The process is similar to Top-Down approaches, but the implementation of Bottom-Up factors related to energy efficiency or household characteristics makes it possible to consider the implementation of energy scenarios.

The generation of energy demand in the residential sector is achieved by modeling the energy demand of each household. It depends on the size of the household, the size of the dwelling, the type of housing, the age of the dwelling and it also depends on a coefficient of thermal efficiency. To generate the energy demand of the residential sector, HESTIA uses population density. The first step is to determine the ratio of "collective housing to individual housing" according to the population density and type of area, using GAIA outputs. This ratio is clearly dependent on the country and local data such as building heights or urban density. The calculation must therefore be specific to the area of interest. In HESTIA, distribution of household dwellings (house / apartment) is formulated as follows:

$$P_{DW\,type}(x) = \begin{cases} P_{DW\,urb}(x), & if\ zone = Urban \\ P_{DW\,sub}(x), & if\ zone = Suburban \\ P_{DW\,rur}(x), & if\ zone = Rural \end{cases} \quad (22)$$

We assume that the CENTER and URBAN areas comprise a majority of buildings, while the SUBURBAN areas are where most of the individual houses are built. First of all, HESTIA starts by calculating the size of the dwelling ($DW_{zone}$) according to a reference size value for different type of dwellings ($Surf_{HT}$), which depends on each specific zone ($\gamma_{UDI}$) and takes into account the number of agents ($n$) living in the dwelling:

$$DW_{zone} = Surf_{HT} \times \gamma_{UDI} \times n \qquad (23)$$

The energy used to heat and boil water is defined by the distribution of the energy mix, an exogenous parameter in the model:

$$P_{energy} = \begin{cases} P(energy_1), & if \ "House" \\ P(energy_2), & if \ "Apartment"' \end{cases} \qquad (24)$$

Then, HESTIA calculates the energy consumption per household, $EN_{cons}$, taking into account the size of the dwelling, $DW_{zone}$, the unit energy consumption per household (ECU) and the household size ($hhsz$) :

$$EN_{cons}(hh) = DW_{zone} \times ECU \times hhsz \qquad (25)$$

Finally, the module applies a climatic correction to energy consumption in order to estimate the under/over consumption of energy due to the cold/hot climate. The degree-day (DD) is the parameter allowing to quantify this correction as a function of the daily temperature of the considered year, compared to a reference year (Jones and Harp, 1982).

The calculation of energy demand of the tertiary, institutional and commercial sectors is similar to that presented above, although it is based on annual energy consumption per employee ($ECUw$). In addition, the spatialization of emissions is derived from the location of employment centers and from their respective capacities (employment data by zone). Thus, the energy demand of employees can be defined as:

$$ENW_{cons}(employee) = ECUw \times n_{worker} \qquad (26)$$

A climate correction is also added to the consumption of this sector.

**6 Emissions**

The role of the VULCAN module is to calculate pollutant emissions from both road transport and energy consumption of buildings, which is the final step of OLYMPUS. There, the quantification of pollutant emissions is based on methodologies recommended by the European Environment Agency (EEA) guidebook for air pollutants and Greenhouse Gas (GHG) emissions (European Environment Agency, 2013). They rely on the use of emissions factors, which may depend on the type of fuel, but also on the age and combustion technology of the engines and stoves. The organizational chart of VULCAN is shown in Figure 9.a.

**6.1 Road emissions**

Road transport emissions - referred to as mobile emissions in the inventory - are calculated on linear road sections where traffic properties at a given time are homogeneous (driving cycle, average speed). For passenger vehicles, the traffic flows are derived from the travel matrices of the assignment module. From a quantitative point of view, emission factors based on traffic characteristics are applied to each section of road in order to obtain the quantities

of pollutants emitted into the atmosphere per unit of time. In the literature, three main databases provide exhaust emission factors. These are HBEFA (Keller et al., 2017), COPERT (Ntziachristos et al., 2009) and MOBILE6 (US EPA, 2003). They differ in that some depend on instantaneous speeds, while others consider average driving speeds or apply to specific driving cycles such as standard highway traffic. The methodology we developed for the VULCAN Emissions Module is based on the recommendations of the EEA, which uses COPERT emission factors based on the average speed of a vehicle during a standard driving cycle (see Figure 9.b, c, d). To be comprehensive in the counting of traffic-related emissions, we added the mechanical emissions of particles from different forms of friction and abrasion during driving, as well as the evaporation of NMVOC from vehicle tanks.

A critical step in the road transport emissions modeling process is to determine the composition of the fleet, which can be inferred from the national composition data as exogenous data. In the assignment module, the choice of specific emission factors depends on the properties of the fleet (age, cylinder, fuel type), and the properties of the agent's car are defined using a conditional probability law. A second important step is the addition of cold start emissions. This makes it possible to take into account the over-emission effect resulting from the poor performance of a vehicle starting and then running with a low-temperature engine. In order to obtain total exhaust emissions, VULCAN first calculates hot emissions factors for the stable engine regimes:

$$E_{hot} = N \times M \times \varepsilon_{hot} \qquad (27)$$

where N is the number of car on a road link, M is the length of the road link and $\varepsilon_{hot}$ is the emission factor. Then, Vulcan calculates cold start emissions using an over-emission factor applied to a fraction of the distance traveled by each vehicle. This factor can be defined as

$$E_{cold} = \beta \times N \times M \times \varepsilon_{hot} \times \left( \varepsilon_{\frac{cold}{hot}} - 1 \right) \quad (28)$$

where ß is the average fraction of the total distance traveled with a cold engine, and $\varepsilon_{\frac{cold}{hot}}$ the cold/hot ratio. The calculation algorithm of the cold/hot emission quotient strongly depends on Euro technology, on the ambient temperature and on the pollutant being considered (European Environment Agency, 2013).

The EEA offers several levels of refinement of calculations, called Tier, the use of which depends on the information available at the input of the calculation. Tier 1 methods are based on a simple linear relationship between activity data and emission factors, representing typical or averaged process conditions, which tend to be technology independent. More advanced Tier 2 methods are available for key categories, allowing in particular to apply country-specific emission factors which depend on processing conditions, fuel qualities or abatement technologies (European Environment Agency, 2013). OLYMPUS uses each time the highest level of accessible detail. All emissions are then computed as follows:

$$E_{tier\ i} = N \times M \times \varepsilon_{tier\ i} \quad (29)$$

where M is the number of travelled kilometers. For example, the calculation emissions from LCVs, HDVs and two-wheeled vehicles is based on the EEA Tier 2 method. This methodology is used because of the excessive uncertainty on the freight fleet. HERMES generates the number N of vehicles using standard fleet composition ratios. Emissions are calculated for CORINAIR pollutants (NOx, NMVOCs, PM) and for $CO_2$.

As mentioned above, emissions related to tire and brake wear add to exhaust emissions, according to the following two equations (European Environment Agency, 2013):

$$E_{tire} = N \times M \times \varepsilon_{TSP} \times f_s \times S_s(V) \qquad (30)$$

$$E_{wear} = N \times M \times \varepsilon_{TSP} \times f_s \qquad (31)$$

where $\varepsilon_{TSP}$ is the mass emission factor of Total Suspended Particulate matter (TSP) for vehicles in category j [g/km], $f_s$ is the mass fraction of TSP attributable to the particle size of class i and $S_s(V)$ the correction factor for an average vehicle traveling at speed V.

Finally, the VULCAN module considers the evaporation of gasoline, using the following equation (European Environment Agency, 2013) :

$$E_{evap} = \sum N \times \varepsilon_{evap} \times 365, \qquad (32)$$

where $\varepsilon_{evap}$ is the evaporative emission factor depending on the ambient temperature.

**6.2 Building emissions**

Emissions from buildings are based on the EEA guidebook for small combustion emissions (European Environment Agency, 2013). This part of the VULCAN modules takes into account emissions from residential heating (fireplace, stoves, cookers, small boilers), as well as institutional and commercial heating. Small combustion emissions from the agricultural sector are not considered.

The calculation of residential and commercial/institutional emissions is based on the EEA methodology and emissions factors derived from Pfeiffer et al., 2000 and Kubica et al., 2007:

$$E_{building} = \sum_{fuel}(\varepsilon_{fuel} \times EN_{cons}(hh)) \qquad (33)$$

There are several types of heating and boiling equipment to consider in combustion modeling for the residential, commercial and institutional sectors (fireplaces, stoves, cookers, small boilers, space heaters, combined heat and power on a small scale). It is important to note that the composition and the age of the fleet are two crucial parameters affecting the emissions of building, as emission factors vary with equipment and age. It has been found that improved combustion technologies has a significant impact on pollutant emissions over the years. However, due to a lack of information in the literature, these parameters remain difficult to estimate accurately. For these reasons, when applying OLYMPUS on a territory, the hypotheses that we will be able to propose for the partition and the spatial distribution of the technologies of heating systems will be a determining point of the realism of the simulation.

**7 Application to Greater Paris**

We applied the OLYMPUS model to the Paris region. One of the reasons for this choice is that the Île-de-France region is based on a classic monocentric urban structure, with a high housing density in the center and an expanding peripheral urban area, clearly raising the problem of mobility, congestion and modal share. More generally, it is a place of intense emissions of anthropogenic pollutants that generate high annual levels of pollution. In winter, in particular, it is affected by serious problems of exposure to fine particles resulting from the combustion of biomass for domestic heating. Finally, like all the areas, the region is facing the challenges of climate change and low-carbon economy, a challenge for which road traffic control could prove to be a particularly effective lever. In this area, the quality of the available input data would allow for greater robustness and better reliability of the simulations.

We conducted a one-year simulation with OLYMPUS in the Paris region. The input data selection, working assumptions, and configurations selected for the OLYMPUS model are described below. The results of the emissions calculations are then analyzed.

**7.1 Configuration**

The simulations were carried out for the year 2009, for which we had a large database of input data (surveys, censuses, inventories). The simulation domain is the Ile de France region (Greater Paris). It is a monocentric urban area with a population density of 21,000 inhabitants / km² in the city center and a density that decreases radially to the remote suburbs, which are predominantly rural (Figure 10).

The population in this territory is greater than 11 million inhabitants. In terms of transport infrastructure, Paris is considered by the Institute for Transportation and Development Policy as the city with the most efficient network (Marks, 2016). Individual mobility is 3.87 trips per person per day on average, with 41 million trips made each day in the region. The majority of trips (70%) do not include travel to the center of the metropolis: trips in Ile-de-France are mostly short (4.4 km on average) and close to home.

In OLYMPUS, the computing space unit is TAZ. Here, it comes from the National Institute of Statistics (INSEE) which has set up a specific division of the territory called IRIS which gathers between 1 800 and 5 000 inhabitants. For our domain, this choice leads to the constitution of 1300 TAZ. Figure 11 illustrates the division of the region into IRIS, as used by OLYMPUS.

The modeling of anthropogenic combustion emissions resulting from individual activities requires a very large amount of data. The main sources of these data are shown in Table 1. To generate the synthetic population in GAIA, we used aggregated data from the census of the city, mainly derived from INSEE. They include the distribution of the population on the territory by age and gender, the number of households by IRIS and the average distribution of households by type (single, couples, family, single-parent family). Mobility calculations in MOIRAI are based on several types of data, mainly surveys or national statistical databases. First, the accessibility of the area by public transport was carried out on the basis of the density of the public transport networks provided by the STIF regional transport agency. Regarding the attractiveness of urban sub-areas, the creation of attractive WORK zones is based on INSEE census data, which provides the number of jobs per municipality. The average distance at which agents can be attracted to an occupational zone is then deduced from the city's overall transport survey (STIF, 2012). The total mobility of the agents is also conditioned by the total number of trips per day

weighted according to the number of priorities of the agents. Here, the average number of daily trips was estimated from local surveys and we assumed that trips other than those related to commuting were characterized by the same average distance, which is not the case in reality. For the OTHERS category, agents' interest in a given activity results from two main parameters: the number of households in the immediate vicinity of an activity and the estimated average distance traveled to reach this activity. Once these values are set, the determining parameter for carrying out the activity will be the estimated duration of the trip. Here, this parameter was derived from THEMIS, whose calibration had been achieved through an online application based on GTFS data from all regional transport agencies. It allows the constitution of a matrix of average transport times between the different classes of urbanized areas (UTAI). In the end, the combination of transport network data and population density makes it possible to calculate the accessibility of any area of activity.

The fleet of vehicles used in HERMES dates back to 2009 and is based on the Carteret et al., 2015 survey. It includes passenger cars, LCVs and HDVs. This study is based on the use of video observation to characterize a fleet of vehicles, then compared to the global transport survey. The regional fleet of stoves and fireplaces was not estimated, so it was not included. We simply assumed here that individual heating modes, including wood heating, came mainly from individual dwellings. The total energy demand on the territory was estimated from data from ARENE surveys (Regional Agency for the Environment and New Energies) providing unitary consumption of households in Ile de France, but also from information on the average household housing area, by type of household and by place of residence, provided by CEREN (Center for Economic Research on Energy). The consumption modeling of the commercial/institutional sector was carried out on the basis of annual consumption per employee of the commercial/institutional sector (CEREN, 2015).

**7.2 Results and discussion**

Figure 11 shows the results of synthetic population modeling, obtained by performing probability functions from aggregated census data. It also shows that we obtain a realistic representation of the variations in household density in the territory. Regarding the characteristics of the population (Figure 11), note that OLYMPUS accurately captures the age distribution of the inhabitants of Greater Paris, compared to data from the INSEE census. However, it should be noted that OLYMPUS underestimates the elderly population by a factor of about 2 - or more for the oldest age group - and underestimates the child population by 24%. These are age groups associated with low mobility, and for the oldest age group this represents only a small proportion of the total number of agents. Thus, based on the low distribution error (in %) of the labor force, we consider that the model generates an acceptable synthetic population for transport modeling. Finally, the average attributes of the agents used as forcing in an aggregated form are correctly represented in the synthetic population: gender data, unemployment rate, distribution of household types and average household size. Because OLYMPUS relies on Bayesian statistics to generate a synthetic population, it is necessary - to obtain representative results - to initially have a large database containing specific information on the distribution of agent characteristics in the population. Thus, thanks to the transcription of stochastic variables, the synthetic population has great similarities with the population studied. Nevertheless, this approach produces limited variability of socio-economic parameters within the distribution, offering a simulated population too close to the average characteristics of the actual population. In this simulation, we limited ourselves to the use of a 3-level UDI index and the division of the domain into 1300 TAZ. It will be interesting – in order to define the most sensitive components of the urban system modeled with OLYMPUS - to

test the sensitivity of the outputs of the model to the increase of the spatial variability of the properties of the agents, to the use of a wider range of indices and to a greater number of TAZ.

Agent mobility modeling was carried out using the OD matrices of home-work trips generated by MORAI, based on data from the employment survey in Greater Paris. Figures 12.a and b illustrate the mobility of the synthetic population modeled by OLYMPUS. Figure 12.a shows the complete set of routes built by OLYMPUS, while Figure 12.b shows more specifically the saturation of the road network in terms of volume on capacity (VOC). This last map shows very high values of the VOC factor in the city center and in the suburbs close to the center, confirming the monocentric nature of the megalopolis perceived by OLYMPUS. The same centrality of the simulated mobility is found in Figure 12.a, which presents trajectories strongly oriented towards the heart of the megacity. This result was compared to mobility indicators from transport surveys. Table 2 shows the comparison of simulated and survey-based data on average number and length of trips per day. The simulated data are very realistic, with a difference of only 4% for the total number of trips per day and per agent in Ile-de-France, and an overestimate of 6.9% for the average trip length, compared to average values of the transport survey. The total number of trips and the modal shares are very close to reality (less than 3% difference with field data). If we consider that differences of less than 5% between simulated and observed values are satisfactory, and that differences of less than 3% are very satisfactory, then this first comparison work reveals that OLYMPUS simulates from satisfactorily to very satisfactorily the main characteristics of the regional transport demand. Only the simulated average trip length is not completely satisfactory (+6.9%). In order to evaluate the OLYMPUS results in more detail (and to propose a comprehensive correction of the average trip length), future works could include be the establishment of a map of the simulated and observed mobility. This spatialized analysis would take into account the fraction, the length, and the modal share of the trips made between the different zones (center-to-center, center-to-near-suburbs, suburb-to-suburbs journeys ...). Such an evaluation requires a lot of data process, but it could allow both a more refined evaluation of OLYMPUS outputs and a diagnosis of mobility levers in the model.

Figure 12.c shows a map of simulated energy demand in Greater Paris. The results show a fairly logical positive dependence on the total population, with maximum demand at the center of the agglomeration. The total energy consumption of Greater Paris (including energy demand for electricity) is 303 TWh in 2009, according to ARENE, 2013. The energy consumption from the residential, commercial and institutional sectors represents more than 50% of this total. The modeling of the regional energy demand (HESTIA) is consistent with surveys, the difference being around + 9.6%.

Emission modeling from both road transport and building heating was carried out by calculating the linear and surface emissions of air pollutants and greenhouse gases for each road segment and consumption unit. It was followed by a projection of the emissions on a regular grid of kilometric resolution. The results, illustrated in Figure 12.d for nitrogen oxides (NOx) - a family of gaseous species emitted during combustion processes - show a very good consistency with the spatial emitting patterns in Île-de-France (major roads, types and density of housing by zone). The total emissions of OLYMPUS are then compared to two reference emission inventories: the AIRPARIF air quality network inventory, and that of the European network EMEP. To this end, we have extracted for each inventory the activity sectors corresponding to the emissions calculated by OLYMPUS. The comparison is presented as histograms in Figure 13, for NOx and for 2 size sections of particulate matter: $PM_{2.5}$

and $PM_{10}$. It should be noted that the calculation methods differ between the 2 "reference" inventories: AIRPARIF develops bottom-up approaches from local data collection, while EMEP inventories are derived from national totals by species, which are spatially disaggregated using top-down approaches. In addition, the comparison with the EMEP inventory cannot be done in detail due to the lack of information on the subsectors of activity in the EMEP data.

The emissions produced by OLYMPUS, although slightly underestimated compared with the emissions of AIRPARIF, are considered here as very satisfactory. Indeed the OLYMPUS emissions, either total or by vehicle type, show differences of less than 20% with the AIRPARIF values for nitrogen oxides (NOx) and particulate matter ($PM_{2.5}$ and $PM_{10}$). The variations in the AIRPARIF emissions from one subgroup to another (PC, LCVs, HDVs) is also well reproduced by OLYMPUS. This is remarkable considering that OLYMPUS (which constructs mobility matrices using a gravity approach and relying on the choice of individuals) has very few forcing data in common with AIRPARIF (which mainly uses road count data, vehicle sales and registration, fuel consumption surveys…). In particular, although both inventories use the COPERT methodology, other sources of differences exist, notably in the hypotheses about the fleet in circulation. An earlier study by Timmermans et al., 2013 confirms that the observed discrepancies in emissions are consistent with the fact that different approaches are used. Indeed, the authors indicate that the expected gap between emission inventories based on different modeling assumptions (choices on the cold start fraction, fuel evaporation emissions modelling, engine fleet…) is expected to be 20% at the minimum. By contrast, particulate matter emissions related to abrasion seem to be more severely underestimated by OLYMPUS compared with the AIRPARIF database (-30%), but there are currently very few ways to estimate real emission values. Deviations from the EMEP inventory are greater, which can be explained by the fact that the EMEP approach is coarse, and strongly overestimates some emissions compared with the AIRPARIF inventory. Nevertheless, the tendency to underestimate emissions in OLYMPUS may reflect the lack of consideration of specific sources in the model. In particular, the model calculates the transport of goods on the basis of occupancy rates of urban roads and does not take into account inter-regional mobility, the city being considered here as a closed system. This largely explains the underestimation of traffic at the borders of the region and in certain rural areas (not shown here). Taking this source into account is considered as a priority evolution of the model.

In addition, the issue of congestion should not be treated superficially in a model like OLYMPUS, because it affects the decision of the agents along their commute and contributes to an increase in emissions. At present, congestion is managed by the representation of speed classes on the main axes, but a more dynamic management of this process is envisaged. One of the methods identified to address this problem is the establishment of an iterative process between congestion and the choice of agents, and the refinement of the representation of speed as a function of the occupancy rate of the lanes. This is also what future developments of the platform will focus on.

The activity sector that shows the highest differences between the two regional inventories is residential combustion, with OLYMPUS underestimating the value of the AIRPARIF inventory for $PM_{10}$ by a factor of 2. However, we mentioned that we do not have information about local combustion equipment data, and that equipment technology is a determinant factor of emissions related to heating. In particular, wood burning is responsible for more than 90% of particulate matter emissions in this sector. In addition, AIRPARIF has its own

modeling assumptions for residential heating, including industrial heaters not based on local data, as well as an estimate of the number of chimneys, stoves, etc. Furthermore, there is no specific survey on combustion technologies in the commercial and institutional commercial and institutional sectors. This point explains the very large variability of emission estimates in the different inventories. The comparison between inventories can hardly overcome this lack of constraints. We should therefore consider that the range of values given by OLYMPUS is consistent with the estimates of the reference inventories, and our work contributes to the improvement of the evaluation of household combustion emissions in Île-de-France.

Finally, as for the total number of trips, the structure of trips and the modal share, the results simulated by OLYMPUS are quite close to the data used as reference (surveys or models), which validates each of the steps in OLYMPUS: the representation of the average number of trips per agent, the division of urban areas into attractiveness classes and the use of the utility function for modal choices. As a consequence, our modeling approach appears relevant for analyzing the links between city, population, mobility, energy consumption and pollutant emissions. A better consideration of the effects of freight transport and congestion will make analyzes conducted with OLYMPUS even more robust.

Nevertheless, to go further in the analysis of the results of OLYMPUS, we must examine the question of rationality. Mobility modeling is based primarily on the theory of random utility, in which each agent in the synthetic population is considered to make rational choices for its transportation mode. Utility is a notion used in economy, and it obviously has several limitations as described by the economists Becker and Murphy (1988). Firstly, because it requires that agents have full access to information in order to make the most rational decision, which is unrealistic. It is more likely that different agents will have access to partial and different sets of information when making decisions. Secondly, the time given to the agent to make his decision influences the final choice, which is not taken into account in this approach. Finally, the maximum utility of an action may not be the same for each person since it depends on the preferences and weights given to each of the various elements that compose the utility function. Despite this, Wegener (2004) resented an exhaustive list of the main land use transport interaction models, including those using the random utility such as EMME, VISSUM, MATSIM. This gives the "utility" approach more robustness because of the reliable predictions made by these models on typical behaviors of people within a group. To better test the implications of rational choice in OLYMPUS, it would be interesting to test the response of the mobility of the agent to the weighting of the utility function, or even to an increase in the variability of its expression.

**8 Conclusions and perspectives**

In the current environmental context, where the problems of air quality and climate change are exacerbated in cities, it is absolutely necessary to develop integrated urban modeling capable of providing a diagnosis of the effects of urban public policies on energy consumption, pollutant emissions, air quality and exposure of the population. The OLYMPUS model has been developed to meet this need. Indeed, OLYMPUS is an emission model calculating pollutant emissions from anthropogenic combustion in the city (road traffic, building heating, cooking, etc.) and designed to integrate into an urban modeling platform, consisting of a set of disciplinary models. In this platform, OLYMPUS need to be forced by data on morphology and urban services (land use, population

density, transport supply, etc.) that can be derived from surveys or obtained from a model providing urban growth from public policy scenarios (densification or urban sprawl, improvement of public transport, green belts...). Then, OLYMPUS builds a synthetic population corresponding to the urban scenario. In a second step, using an individual decision-based approach, it provides mobility matrices and spatialized energy demand to account for agents' activities in the city. The energy demand is finally used to produce a pollutant and GHG emission inventory at a kilometric scale, for the whole urban area. The data produced by OLYMPUS allows much more advanced scenarios than the usual air quality models (they only consider a regulatory percentage of emission reductions). OLYMPUS simulates the scenario elements needed to provide decision support that takes into account the collective appropriation of environmental policies as well as the impact of urban forms on transport demand and mobility, which is absolutely not the case in the current literature.

OLYMPUS has been implemented in the Paris region. The emissions inventory produced with this model shows a fairly good understanding of the individual and collective activities that consume energy in this area. Although OLYMPUS outputs (pollutant emissions) have been shown to be quite comparable with the AIRPARIF emissions inventory, assumptions about congestion, lack of representation of freight transport and inter-regional mobility may be responsible for an overall underestimation of emissions. The representation of these processes will be improved in the very short term in OLYMPUS.

In the longer term, other evolutions have to be considered for OLYMPUS.
Uncertainties on domestic / residential / commercial heating emissions can be reduced by changing the energy demand setting in OLYMPUS, integrating local parameters such as building height or sunlight. However, a significant improvement could be immediately achieved by using a realistic speciation of energy consumption according to the type of agent of the population. Although these data are not fully available for the Paris region, these improvements should be considered for future works.
Improvements can be included in the representation of agent choices. It would therefore be interesting to introduce additional socio-economic segregation parameters (for example, household income) and their impact on mobility choices. This could allow us to highlight social discrimination in our emissions analysis. It may also be interesting to set up feedback loops between OLYMPUS modules to simulate agent interactions and develop a multi-agent model. For example, the discrete choice module can be coupled to the path assignment module to make a feedback between congestion and modal choice. Finally, among the developments envisaged for the model, it would be also interesting to test the implementation of time-dependent attractiveness classes, which may modify the spatial distribution of mobility outside working hours. In addition, the sequencing of activities that influence the temporal variability of the activities is a parameter that can be improved by adding a specific program according to the characteristics of the agents of the synthetic population.

In addition to OLYMPUS developments, one of our main perspectives is to implement OLYMPUS on other cities with different morphologies, in order to test the transposability of the model, and to have a broader vision of cities' response to political forcing.

**Code availability**

[revised manuscript text omitted]

Le Néchet, F.: Aprroche multiscalaire des liens entre mobilité quotidienne, morphologie et soutenabilité des métropoles européennes. Cas de Paris et de la région Rhin-Ruhr., , 496, 2010.

Ntziachristos, L., Gkatzoflias, D. and Kouridis, C.: COPERT: A European road transport emission inventory model, , (April 2016), doi:10.1007/978-3-540-88351-7, 2009.

O'Neill, M. S., Jerrett, M., Kawachi, I., Levy, J. I., Cohen, A. J., Gouveia, N., Wilkinson, P., Fletcher, T., Cifuentes, L., Schwartz, J., Bateson, T. F., Cann, C., Dockery, D., Gold, D., Laden, F., London, S., Loomis, D., Speizer, F., Van den Eeden, S. and Zanobetti, A.: Health, wealth, and air pollution: Advancing theory and methods, Environ. Health Perspect., 111(16), 1861–1870, doi:10.1289/ehp.6334, 2003.

Ortuzar, J. de D. and Willumsen, L. G.: Modelling Transport, 4th Edition, 2011 John., WILEY, ISBN: 9780470760390., 2011.

Pfeiffer, F., Struschka, M., Baumbach, G., Hagenmaier, H. and Hein, K. R. G.: PCDD/PCDF emissions from small firing systems in households, Chemosphere, 40(2), 225–232, doi:10.1016/S0045-6535(99)00307-0, 2000.

Pisman, A., Allaert, G. and Lombaerde, P.: Urban and suburban lifestyles and residential preferences in a highly urbanized society, , (September 2018), 1–2, doi:10.4000/belgeo.6394, 2011.

Rahman, A.: Estimating small area health-related characteristics of populations: A methodological review, Geospat. Health, 12(1), doi:10.4081/gh.2017.495, 2017.

Rexeis, M., Hausberger, S., Kühlwein, J. and Luz, R.: Update of Emission Factors for EURO 5 and EURO 6 vehicles for the HBEFA Version 3 . 2, Univ. Technol. Graz, Rep. Nr. I-31/2013 Rex-EM-I, 43(316), doi:Report No. I-25/2013/ Rex EM-I 2011/20 679, 2013.

De Ridder, K., Lefebre, F., Adriaensen, S., Arnold, U., Beckroege, W., Bronner, C., Damsgaard, O., Dostal, I., Dufek, J., Hirsch, J., IntPanis, L., Kotek, Z., Ramadier, T., Thierry, A., Vermoote, S., Wania, A. and Weber, C.: Simulating the impact of urban sprawl on air quality and population exposure in the German Ruhr area. Part II: Development and evaluation of an urban growth scenario, Atmos. Environ., 42(30), 7070–7077, doi:10.1016/j.atmosenv.2008.06.044, 2008.

Schindler, M. and Caruso, G.: Urban compactness and the trade-off between air pollution emission and exposure: Lessons from a spatially explicit theoretical model, Comput. Environ. Urban Syst., 45, 13–23, doi:10.1016/j.compenvurbsys.2014.01.004, 2014.

Seto K. C., S. Dhakal, Bigio, A., Blanco, H., Delgado, G. C., Dewar, D., Huang, L., Inaba, A., Kansal, A., Lwasa, S., McMahon, J. E., Müller, D. B., Murakami, J., Nagendra, H. and Ramaswami, A.: Human Settlements, Infrastructure and Spatial Planning., in Climate Change 2014: Mitigation of Climate Change. Contribution of Working Group III to the Fifth Assessment Report of the Intergovernmental Panel on Climate Change, pp. 923–1000., 2014.

Steinle, S., Reis, S. and Eric, C.: Science of the Total Environment Quantifying human exposure to air pollution — Moving from static monitoring to spatio-temporally resolved personal exposure assessment, Sci. Total Environ., 443, 184–193, doi:10.1016/j.scitotenv.2012.10.098, 2013.

STIF: Enquête globale transport: La mobilité en Île-de-France, Paris, France. [online] Available from: http://www.omnil.fr/IMG/pdf/egt2010_enquete_globale_transports_-_2010.pdf, 2012.

Swan, L. G. and Ugursal, V. I.: Modeling of end-use energy consumption in the residential sector: A review of modeling techniques, Renew. Sustain. Energy Rev., 13(8), 1819–1835, doi:10.1016/j.rser.2008.09.033, 2009.

Taylor, N. B.: The CONTRAM Dynamic Traffic Assignment Model, Networks Spat. Econ., 3, 297–322, doi:10.1023/A:1025394201651, 2003.

Thomas, E., Serwicka, I. and Swinney, P.: Urban demographics Why people live where they do., 2015.

Timmermans, R. M. A., Denier van der Gon, H. A. C., Kuenen, J. J. P., Segers, A. J., Honoré, C., Perrussel, O., Builtjes, P. J. H. and Schaap, M.: Quantification of the urban air pollution increment and its dependency on the use of down-scaled and bottom-up city emission inventories, Urban Clim., 6, 44–62, doi:10.1016/j.uclim.2013.10.004, 2013.

U.S. Environmental Protection Agency: Draft Motor Vehicle Emission Simulator (MOVES) 2009: Software Design and Reference Manual, 2013.

UK Department of Transport: Traffic Appraisal Advice, in Design manual for roads and bridges, vol. 12. [online] Available from: http://www.standardsforhighways.co.uk/ha/standards/dmrb/vol12/section2/12s2p1.pdf, 1997.

United Nations: World Urbanization Prospects, United Nations, 12, 32, doi:10.4054/DemRes.2005.12.9, 2014.

US EPA: User 's Guide to Mobile Source Emission Factor Model, , (August), 262 [online] Available from: http://www.epa.gov/otaq/models/mobile6/420r03010.pdf, 2003.

Wegener, M.: Overview of land-use transport models. [online] Available from: http://www.spiekermann-wegener.de/pub/pdf/MW_Handbook_in_Transport.pdf, 2004.

World Health Organization: Review of evidence on health aspects of air pollution – REVIHAAP Project. [online] Available from: http://www.euro.who.int/__data/assets/pdf_file/0004/193108/REVIHAAP-Final-technical-report-final-version.pdf, 2013.

[revised manuscript text omitted]

---

## Author Comment (AC2) · 23 Oct 2018

You can find the final response in attachments. Best regards.

Please also note the supplement to this comment:
https://www.geosci-model-dev-discuss.net/gmd-2018-154/gmd-2018-154-AC2-supplement.pdf

———————————————————

2018.